# On the Power of (Approximate) Reward Models for Inference-Time Scaling: Sequential Monte Carlo and Beyond

**Youheng Zhu** [1]   **Yiping Lu** [1]

## Abstract

Inference-time scaling has recently emerged as a powerful paradigm for improving the reasoning capability of large language models. Among various approaches, *Sequential Monte Carlo (SMC)* has become a particularly important framework, enabling iterative generation, evaluation, rejection, and resampling of intermediate reasoning trajectories. A central component in this process is the *reward model*, which evaluates partial solutions and guides the allocation of computation during inference. However, in practice, true reward models are never available. All deployed systems rely on *approximate reward models*, raising a fundamental question: *Why and when do approximate reward models suffice for effective inference-time scaling?* In this work, we provide a theoretical answer. We identify the *Bellman error* of the approximate reward model as the key quantity governing the effectiveness of SMC-based inference-time scaling. For a reasoning process of length $T$, we show that if the Bellman error of the approximate reward model is bounded by $O(1/T)$, then combining this reward model with SMC reduces the computational complexity of reasoning from exponential in $T$ to polynomial in $T$. This yields an *exponential improvement* in inference efficiency despite using only approximate rewards.

## 1. Introduction

Over the past several years, the reasoning ability of large language models has improved dramatically (Wei et al., 2022). While model scale and training data remain critical factors, a complementary paradigm of inference-time scaling has recently emerged as a powerful mechanism for enhancing reasoning performance without modifying model parameters (Ke et al., 2025; Zhou et al., 2023; Hendrycks et al., 2021; Lightman et al., 2023). By allocating additional computation during inference, such methods enable models to explore, refine, and verify intermediate reasoning steps, yielding substantial gains on complex multi-step tasks (Wei et al., 2022; Wang et al., 2023). Inference-time scaling improves model performance by generating multiple candidate answers along with diverse reasoning paths and selecting higher-quality outputs using a reward model (Liu et al., 2025). This strategy has demonstrated strong empirical success across a wide range of reasoning tasks. However, despite its practical effectiveness, there remains little theoretical understanding of how many reasoning paths are required to achieve a given level of performance, or what principles govern this trade-off. In this work, we take a first step toward addressing this gap by investigating how the reward model contributes to the effectiveness of inference-time scaling and clarifying its role in guiding the selection process. Despite its empirical success, the relationship between the number of reasoning paths, reward model quality, and overall performance remains poorly understood.

Among existing inference-time scaling approaches, Sequential Monte Carlo (SMC) has gained increasing attention as a principled and flexible framework for structured reasoning (Feng et al., 2025; Kim et al., 2025; Zhao et al., 2024; Loula et al., 2025). In SMC-based inference, the model generates multiple candidate reasoning trajectories, evaluates their quality using a reward model, and iteratively performs rejection and resampling to concentrate computation on the most promising paths. This procedure allows the system to trade computation for solution quality in a controlled and theoretically grounded manner. A critical ingredient underlying the success of SMC is the availability of a reward model that can reliably assess the quality of partial reasoning states. In practice, however, such reward models are never exact. They are learned from finite data, noisy human feedback, or heuristic signals, and thus inevitably deviate from the true task objective. This raises a fundamental and largely unexplored question:

Why does inference-time scaling work at all when only

[1]Department of Industrial Engineering and Management Sciences, Northwestern University, Evanston, USA. Correspondence to: Yiping Lu <yiping.lu@northwestern.edu>.

*Proceedings of the $43^{rd}$ International Conference on Machine Learning*, Seoul, South Korea. PMLR 306, 2026. Copyright 2026 by the author(s).

approximate reward models are available?

Intuitively, one might expect small inaccuracies in reward estimation to accumulate over long reasoning chains, causing SMC to amplify errors rather than correct them. Yet empirically, approximate reward models appear sufficient to drive remarkable improvements in reasoning performance. Understanding this phenomenon is essential for both the theoretical foundations of inference-time scaling and the practical design of robust reward models.

## 1.1. Contribution

A unifying perspective of inference-time scaling is *twisting* the base model dynamics to realize reward-tilted sampling. The *optimal* twist is a *lookahead-to-go value function*, but is generally intractable. This motivates our focus on *approximate* value functions and how their approximation quality (via Bellman error) governs the computational efficiency of sampling. The main algorithms studied in the main text are shown in Table 1. Particularly, our contribution includes:

- **Lower bounds: sub-exponential complexity needs small Bellman error.** In Section 4, we prove two exponential-in-$T$ information-theoretic lower bounds for inference-time sampling: one *without* intermediate guidance (Theorem 4.1), and one *with* access to an approximate reward model (Corollary 4.2). In particular, escaping exponential dependence on the reasoning horizon requires a *small Bellman error* regime (e.g., $\varepsilon = O(1/T)$).
- **First particle & time complexity bounds for SMC in inference-time scaling.** Sequential Monte Carlo (SMC) is a core inference-time scaling primitive with strong empirical success. We provide in Section 5 the *first* end-to-end guarantees that SMC achieves TV accuracy $\delta_{\mathrm{TV}}$ with an explicit number of particles $N$ and total runtime polynomial in $(T, \delta_{\mathrm{TV}}^{-1})$ under our Bellman-error Assumption 3.2. Moreover, the theory and analysis is discussed on a flexible choice of proposals and particle transition dynamics in the Appendices, providing useful guidance for future algorithm designs.
- **Complexity of single-particle guided SMC with Metropolis-Hastings correction.** We study a widely used naive guidance primitive by formalizing it as a *single-particle* guided SMC update, which alone cannot achieve arbitrarily small TV error unless the reward model is perfect (Theorem 4.3). To quantify the computational knob needed for arbitrary accuracy, we augment it with a Metropolis–Hastings correction and prove geometric TV contraction on a high-probability event, implying that $O\big(\log(\delta_{\mathrm{TV}}^{-1})\big)$ MH steps suffice to reach TV accuracy $\delta_{\mathrm{TV}}$. This matches the qualitative geometric mixing behavior in (Rohatgi et al., 2025) while imposing no extra information requirement for the algorithm.

*Table 1.* Comparison of time complexity between main algorithms in the main text under Assumptions 3.1 and 3.2. Results gathered from Corollary 5.2 and Theorem 6.1. We consider sentence level reasoning algorithms.

| METHOD | TIME COMPLEXITY |
|---|---|
| BELLMAN ERROR ASSUMPTION 3.2 ($\varepsilon = O(1/T)$) | |
| SP-GSMC+MH | $\tilde{O}\big(LT^3 \log(\delta^{-1}) \log(\delta_{\mathrm{TV}}^{-1})\big)$ |
| SMC (NAIVE) | $O\big(L^6 T^2 \delta_{\mathrm{TV}}^{-1}\big)$ |
| VGB$^\dagger$ | $\tilde{O}\big(LT^3 \log(\delta^{-1}) \log(\delta_{\mathrm{TV}}^{-1})\big)$ |

$^\dagger$VGB refers to (Rohatgi et al., 2025). This algorithm requires special knowledge of $L(1 + \varepsilon)$ for the choice of hyperparameter. This is because they use rejection sampling (Algorithm 3) which requires the rejection ratio to be greater than the coverage between distributions; see Remark B.2.

**Remark:** single-particle guided SMC (SP-gSMC) without correction cannot produce samples with arbitrary accuracy, unless the reward model is perfect (Theorem 4.3).

## 2. Preliminaries

In this section, we introduce the main tools used in the analysis of this paper. We also refer the readers to Appendix A for a full list of notations used throughout the paper.

### 2.1. Metropolis–Hastings

Metropolis–Hastings (MH) provides a generic way to sample from a target distribution when we can only evaluate it up to a normalizing constant. In this paper, MH will be used as a correction step that turns an easily-sampled proposal into a Markov chain whose stationary distribution is the desired (reward-tilted) target. Let $\tilde{\pi}$ be a target distribution on a state space E. We assume $\tilde{\pi}$ is only available through an unnormalized score $\tilde{\pi}'$ such that $\tilde{\pi}(x) \propto \tilde{\pi}'(x)$ (the normalizing constant is unknown). Let $q(x, \cdot)$ be a proposal distribution from which we can sample efficiently. Given the current state $x \in \mathsf{E}$, draw a proposal $y \sim q(x, \cdot)$ and accept it with probability

$$\alpha(x, y) := 1 \wedge \frac{\tilde{\pi}'(y)\, q(y, x)}{\tilde{\pi}'(x)\, q(x, y)}.$$

If accepted, set $X_{n+1} = y$; otherwise set $X_{n+1} = x$. Equivalently, the MH transition kernel can be written as

$$K(x, dy) := q(x, dy)\, \alpha(x, y) + \\ \Big(1 - \int q(x, dy')\, \alpha(x, y')\Big)\, \delta_x(dy),$$

where $\delta_x$ denotes the Dirac measure at $x$. By construction, $K$ is $\tilde{\pi}$-reversible and hence induces a Markov chain which admits $\tilde{\pi}$ as an invariant distribution.

## 2.2. Feynman-Kac Model for Sequential Monte Carlo

Sequential Monte Carlo (SMC) methods are a class of simulation-based algorithms that approximate sequences of probability distributions using a collection of weighted particles evolved through resampling an propagating.

A Feynman–Kac model describes a sequence of probability distributions obtained by iteratively *reweighting by potential functions*, *normalizing*, and *propagating via Markov kernels*. Formally, it defines a (discrete) flow of probability measures $\{\eta_t\}_{t\geq 0}$, where $\eta_t \in \mathcal{P}(E_t)$ is a distribution on a (possibly time-varying) state space $E_t$, usually called the Feynman-Kac flow.

From a dual perspective, a Feynman–Kac model characterizes the evolution of distributions via a sequence of operators. These operators can be viewed either as acting on measures or, equivalently, through their adjoint action on test functions. Concretely, an operator $P$ on measures is uniquely specified by its adjoint $P$ on test functions through the duality pairing $\langle \mu, f \rangle := \int f\, d\mu$, i.e., $\langle \mu P, f \rangle = \langle \mu, Pf \rangle$. Thus for each $t \in [T]$, let $M_t$ be a Markov kernel from $E_{t-1}$ to $E_t$, written as $M_t(x, dy)$, and let it act on test functions $f : E_t \to \mathbb{R}$ by $M_t(f)(x) := \int f(y) M_t(x, dy)$. Let $G_{t-1} : E_{t-1} \to [0, \infty)$ be the potential function. Define the unnormalized Feynman–Kac operator acting on test functions by

$$Q_t(f)(x) := G_{t-1}(x)\, M_t(f)(x), \quad Q_{p,n} := Q_{p+1}\cdots Q_n.$$

By duality, the corresponding action of $Q_t$ on measures is defined via $\langle \mu Q_t, f \rangle = \langle \mu, Q_t f \rangle$, so $Q_t$ can be interpreted as an unnormalized "propagate–reweight" update on distributions. Starting from an initial distribution $\eta_0 \in \mathcal{P}(E_0)$, define the unnormalized flow $\gamma_t \in \mathcal{P}(E_t)$

$$\gamma_t := \gamma_{t-1} Q_t, \qquad \eta_t := \frac{\gamma_t}{\gamma_t(1)}.$$

Intuitively, the recursion $\gamma_t = \gamma_{t-1} Q_t$ corresponds to a *reweight–propagate* evolution of measures: first, $\gamma_{t-1}$ is tilted on $E_{t-1}$ by the potential $G_{t-1}$, and then the reweighted mass is pushed forward to $E_t$ by the Markov kernel $M_t$, yielding an unnormalized measure. Equivalently, $\gamma_t(dy) = \int \gamma_{t-1}(dx)\, G_{t-1}(x)\, M_t(x, dy)$. The normalized flow $\eta_t = \gamma_t / \gamma_t(1)$ is the corresponding probability law after this reweighting and propagation. Building upon this, one may also consider the normalized semigroup, i.e. incorporating the normalizing step into a nonlinear operator

$$\Phi_{p,n}(\mu)(f) := \frac{\mu(Q_{p,n}(f))}{\mu(Q_{p,n}(1))}, \qquad (0 \leq p \leq n \leq T),$$

so that we have $\eta_t = \Phi_t(\eta_{t-1})$, where $\Phi_t := \Phi_{t-1,t}$.

Now we describe a SMC algorithm using the nonlinear FK flow $\eta_{p+1} = \Phi_{p+1}(\eta_p)$. Namely, for each $p \in \{0, \dots, T-$

---

**Algorithm 1** SMC for a FK model with $(G_t, M_t)$

1: **Input:** horizon $T$, #particles $N$, initial law $\eta_1$ on $E_1$, potentials $\{G_{t-1} : E_{t-1} \to \mathbb{R}_+\}_{t=2}^T$, kernels $\{M_t : E_{t-1} \to \mathcal{P}(E_t)\}_{t=2}^T$.
2: Sample $X_1^i \sim \eta_1$ independently for $i = 1, \dots, N$.
3: **for** $t = 2, 3, \dots, T$ **do**
4: $\quad w_{t-1}^j \leftarrow G_{t-1}(X_{t-1}^j), \quad \bar{w}_{t-1}^j \leftarrow \frac{w_{t-1}^j}{\sum_{\ell=1}^N w_{t-1}^\ell}.$
5: $\quad$ Resample $A_{t-1}^i \sim \text{Cat}(\bar{w}_{t-1}^{1:N})$ i.i.d. for $i \in [N]$.
6: $\quad$ Draw $X_t^i \sim M_t(X_{t-1}^{A_{t-1}^i}, \cdot)$ independently for $i \in [N]$.
7: **end for**
8: Draw $I \sim \text{Unif}(\{1, \dots, N\})$ and **Output** $X_T^I$.

---

$1\}$ and each $\eta \in \mathcal{P}(E_p)$, let $K_{p+1,\eta}$ be a Markov kernel from $E_p$ to $E_{p+1}$ such that

$$\eta\, K_{p+1,\eta} = \Phi_{p+1}(\eta). \tag{1}$$

SMC simulate the Feyman-Kac flow via particles $\{X_p^i\}_{i=1}^N \subset E_p$ and their empirical measure $\eta_p^N := \frac{1}{N} \sum_{i=1}^N \delta_{X_p^i}$, the corresponding interacting particle update samples the next generation by the product kernel

$$(X_{p+1}^1, \dots, X_{p+1}^N) \sim \prod_{i=1}^N K_{p+1,\eta_p^N}(X_p^i, dx_{p+1}^i), \tag{2}$$

conditionally independently given $\mathcal{F}_p^N$, with $\mathcal{F}_p^N := \sigma(X_q^i : 0 \leq q \leq p,\ 1 \leq i \leq N)$ be the smallest $\sigma$-algebra that makes all particles up to time $p$ measurable.

Equation (1) generally admits many choices of $K_{p+1,\eta}$. One simple choice is an $x$-independent kernel

$$K_{p+1,\eta}(x, dy) = \Phi_{p+1}(\eta)(dy),$$

in which case particles are drawn i.i.d. from the current FK target $\Phi_{p+1}(\eta_p^N)$, and the process can be described in Algorithm 1. Intuitively, from the formulation of SMC sampler, it can be viewed as a particle-based *discrete simulation* of this reweighting-and-normalization step of Feynman-Kac flow: it replaces the (intractable) weighted measure update by resampling on finitely many particles, followed by propagation.

## 3. Problem Setting

In this paper, we model inference-time reasoning as *reward-tilted sampling* over reasoning trajectories. Let $\mathcal{S}_0$ be the prompt space, and let $\mathcal{S}_t$ be the reasoning state space (token/sentence) at step $t$. We write $\mathcal{S}_{1:t} := \mathcal{S}_{1:t-1} \times \mathcal{S}_t$ (with $\mathcal{S}_{1:0} := \{\varnothing\}$) for the space of prefixes, so a trajectory $s_{1:T} \in \mathcal{S}_{1:T}$ is generated from a variable-length prompt $s_0 \in \mathcal{S}_0$ by an autoregressive base model $\pi_{\text{ref}}(s_{1:T}|s_0) =$

$\prod_{t=1}^{T} \pi_{\text{ref}}(s_t|s_0, s_{1:t-1})$. Following (Zhao et al., 2024), we treat inference-time reasoning as sampling from an energy-tilted posterior over reasoning trajectories, with $\pi_{\text{ref}}$ as the prior. The induced target distribution is given by

$$\tilde{\pi}(s_{1:T}|s_0) \propto \pi_{\text{ref}}(s_{1:T}|s_0)\, \phi(s_{1:T}),$$

where $\pi_{\text{ref}}$ is the pretrained model's intrinsic prior over natural-language reasoning paths, and $\phi(s)$ is a task-dependent utility (or energy) function (e.g. human preference model or verifiable rewards) that biases inference toward high-quality solutions. This formulation is often referred to as *test-time alignment*. For any $t < T$, we also define the intermediate reasoning distribution

$$\tilde{\pi}_t(s_{1:t}|s_0) \ \propto \ \pi_{\text{ref}}(s_{1:t}|s_0)\, \hat{V}(s_0, s_{1:t}),$$

where $\hat{V}(s_0, s_{1:t})$ evaluates the quality of partial reasoning prefixes and $\hat{V}(s_0, s_{1:T}) = \phi(s_{1:T})$. Our goal is to fit a Feynman-Kac model to this sequence of intermediate distributions, then our previously introduced Algorithm 1 can be applied directly to this language model reasoning setting.

Before getting into details of the Feynman-Kac model and its induced SMC process, we briefly show the general procedure of SMC used in language model reasoning: Suppose the samples are drawn from a proposal distribution $\pi_{\text{p}}$, we maintain $N$ particles that approximate $\tilde{\pi}_{t-1}$ and iteratively update them to target $\tilde{\pi}_t$ via two steps:

- **Resampling:** given prefixes $s_{1:t-1}^{1:N}$, compute weights $w_{t-1}^{1:N}$, normalize $\bar{w}_{t-1}^{j} := \frac{w_{t-1}^{j}}{\sum_{\ell=1}^{N} w_{t-1}^{\ell}}$, sample $A_{t-1}^{i} \sim \text{Cat}(\bar{w}_{t-1}^{1:N})$ i.i.d., and set $\tilde{s}_{1:t-1}^{i} := s_{1:t-1}^{A_{t-1}^{i}}$.
- **Propagating:** sample $s_t^i \sim \pi_{\text{p}}(\cdot|s_0, \tilde{s}_{1:t-1}^i)$ for each $i \in [N]$, and form $s_{1:t}^i := (\tilde{s}_{1:t-1}^i, s_t^i)$.

And the above procedure is preformed iteratively until the terminal time step.

**Feynman-Kac model and the SMC sampler.** With the previous idea, we first recast $\{\tilde{\pi}_t\}_{t=1}^{T}$ in the standard Feynman–Kac (FK) notation to streamline the presentation and facilitate rigorous theoretical analysis. We set $E_t := \mathcal{S}_{1:t}$ and introduce a proposal kernel $\pi_{\text{p}}(\cdot|s_{1:t})$ that defines the Markov propagation $M_t(s_{1:t}, ds_{1:t+1}) := \pi_{\text{p}}(ds_{t+1}|s_{1:t})\, \delta_{s_{1:t}}(ds_{1:t})$. In other words, the SMC sampler sample from the proposal $\pi_{\text{p}}$ at each step. Then, as in (Zhao et al., 2024), the incremental weight (potential) is

$$G_t(s_{1:t}) := \frac{\pi_{\text{ref}}(s_t \mid s_{1:t-1})}{\pi_{\text{p}}(s_t \mid s_{1:t-1})} \cdot \frac{\hat{V}(s_0, s_{1:t})}{\hat{V}(s_0, s_{1:t-1})}.$$

With an additional terminal step, i.e. a step $T+1$ with the *identity* Markov kernel $M_{T+1} = \text{Id}$ on $E_T$ (so $E_{T+1} = E_T$), and take the last potential to be $G_T$, the target $\tilde{\pi}_T(s_{1:T}|s_0) \propto \pi_{\text{ref}}(s_{1:T}|s_0)\hat{V}(s_0, s_{1:T})$ is realized as the

terminal distribution of the induced FK flow, and any intermediate distribution $\tilde{\pi}_t$ equals $\eta_t$ reweighted by $G_t$. Then, the aforementioned **"Resampling"** and **"Propogating"** procedures can be rigorously described by Algorithm 1, with the specification of the $(G_t, M_t)$ pairs. By construction, there's one dimension of freedom in specifying $(G_t, M_t)$, namely the proposal distribution $\pi_{\text{p}}$, which we discuss further in Appendix D and in Proposition E.7 of Appendix E. Among our main theorems, we focus on the simplest and the most straightforward choice of proposal $\pi_{\text{p}}$, namely the naive proposal $\pi_{\text{p}} = \pi_{\text{ref}}$. That is to say, the SMC sampler samples from the output distribution of the pre-trained model itself. Under this circumstance, we have

$$G_t(s_{1:t}) = \frac{\hat{V}(s_0, s_{1:t})}{\hat{V}(s_0, s_{1:t-1})}.$$

In Appendix B in specific, we instantiate Algorithm 1 for the *naive* proposals in Algorithms 5.

**Optimal Twists.** (Zhao et al., 2024) shows that the twist function $V^*(s_0, s_{1:t})$ that recover the intermediate marginals of the reward-tilted distribution $\tilde{\pi}_{s_{1:t}}(s_{1:t}|s_0) := \sum_{s_{t+1:T}} \tilde{\pi}(s_{1:T}|s_0)$ should be the solution to the following bellman equation

$$\begin{aligned} V^*(s_0, s_{1:t}) &\propto \sum_{s_{t+1:T}} \pi_{\text{ref}}(s_{t+1:T}|s_0, s_{1:t})\phi(s_{1:T}) \\ &\propto \sum_{s_{t+1}} \pi_{\text{ref}}(s_{t+1}|s_0, s_{1:t})V^*(s_0, s_{1:t+1}). \end{aligned} \tag{3}$$

Intuitively, the twist function plays the role of a "look-ahead" evaluator: given the current partial trajectory $(s_0, s_{1:t})$, it estimates how good this prefix is by averaging over all possible future continuations under the reference policy, weighted by their final rewards. Since this quantity measures the expected desirability of the current state in terms of future outcomes, it naturally corresponds to a value function.

In practice however, computing the optimal twist function $V^*(s_0, s_{1:t})$ is intractable, and therefore approximate methods are employed to learn an approximate reward model. Some of these methods include:

- **Contrastive twist learning (CTL).** (Zhao et al., 2024) propose *contrastive twist learning*, which learns time-indexed prefix twists (reward-to-go) $\{\hat{V}_t^\theta\}_{t=1}^{T}$ by matching the target prefix marginals. Defining the twisted prefix model $\tilde{\pi}_t^\theta(s_{1:t}|s_0) \propto \pi_{\text{ref}}(s_{1:t}|s_0)\hat{V}_t^\theta(s_0, s_{1:t})$, CTL solves $\min_\theta \sum_{t=1}^{T} D_{\text{KL}}(\tilde{\pi}_t(s_{1:t}|s_0)\|\tilde{\pi}_t^\theta(s_{1:t}|s_0))$, yielding a contrastive (positive–negative) update whose expectations are estimated via importance sampling / SMC (Zhao et al., 2024).
- **Soft-RL.** Recall that the optimal twist satisfies a recursive Bellman equation defined by (3). Therefore, Soft-RL type method (Levine, 2018) fit a value/twist function to

satisfy a soft Bellman-type recursion (or path-consistency constraint (Nachum et al., 2017)) implied by the reward-tilted objective. The idea has been used in language model literature like (Lioutas et al., 2023; Mudgal et al., 2024; Guo et al., 2022; Hu et al., 2024).

- **Density-ratio twist learning (e.g., SIXO).** SIXO (Lawson et al., 2022) views twist learning as density-ratio estimation via noise-contrastive (binary) classification. The optimal prefix twist can be written as a posterior-to-prior ratio $V_t^*(s_0, s_{1:t}) \propto \frac{\tilde{\pi}_t(s_{1:t}|s_0)}{\pi_{\text{ref}}(s_{1:t}|s_0)}$, where $\tilde{\pi}_t$ is the $t$-step marginal induced by the reward-tilted target $\tilde{\pi}(s_{1:T}|s_0) \propto \pi_{\text{ref}}(s_{1:T}|s_0)\phi(s_0, s_{1:T})$. SIXO fits a classifier that distinguishes *positive* prefixes $s_{1:t} \sim \tilde{\pi}_t(\cdot|s_0)$ from *negative* prefixes $s_{1:t} \sim \pi_{\text{ref}}(\cdot|s_0)$; the optimal logit recovers $\log V_t^*(s_0, s_{1:t})$, yielding a learned twist.
- **CD-FUDGE.** The method proposed by (Yang & Klein, 2021) employs a function approximation type algorithm to preform a direct regression on the objective $\min_{V \in \mathcal{V}} \sum_{t=1}^{T-1} \mathbb{E}_{\pi_{\text{ref}}(\cdot|s_0, s_{1:t})}\left[(V(s_0, s_{1:t}) - \phi(s_{1:T}))^2\right]$, which is equivalent to solving a on-policy evaluation problem using Monte Carlo for any given prefix. In this case, error may be induced by the insufficiency of function class, finite sample error and optimization gap.

**Connection to Bellman error assumptions.** These objectives can be interpreted by the Bellman error they implicitly control. Soft-RL / path-consistency targets a *local* (one-step) consistency of the prefix reward-to-go along the FK flow, matching Assumption 3.2. In contrast, CD-FUDGE regresses directly to the terminal potential $\phi(s_{1:T})$ via Monte-Carlo rollouts, aligning more naturally with the *global* assumption set in Appendix H (Assumption H.1). Potentially, the FUDGE-style regression can also be reformulated as a local Bellman-error objective using double sampling.

Despite strong empirical success, the interplay between inference-time compute (the number of sampled reasoning paths), reward-model quality, and final performance remains poorly understood. In particular, existing methods show large gains from sampling more trajectories even with only approximate rewards, but it is unclear when such approximation is sufficient for effective scaling and when improving the reward model actually yields additional performance gains—insights that are key to allocating modeling effort and inference resources efficiently. To characterize the reward model, and the metric of how well the model approximates the optimal twist, we introduce the following two assumptions.

**Assumption 3.1** (Reward model ratio bound). We assume for some $L > 0$,

$$\max_{t \in [T]} \max_{s_{1:t} \in \mathcal{S}_{1:t}} \max\left\{ \frac{\hat{V}(s_0, s_{1:t-1})}{\hat{V}(s_0, s_{1:t})}, \frac{\hat{V}(s_0, s_{1:t})}{\hat{V}(s_0, s_{1:t-1})} \right\} \leq L.$$

**Assumption 3.2** (Bellman error uniform bound). We as-

sume for some $\varepsilon > 0$, the following holds:

$$\max_{t \in [T]} \max_{s_{1:t} \in \mathcal{S}_{1:t}} \max\left\{ \frac{\hat{V}(s_0, s_{1:t})}{\mathbb{E}_{s_{t+1} \sim \pi_{\text{ref}}(\cdot|s_0, s_{1:t})}[\hat{V}(s_0, s_{1:t+1})]}, \right.$$
$$\left. \frac{\mathbb{E}_{s_{t+1} \sim \pi_{\text{ref}}(\cdot|s_0, s_{1:t})}[\hat{V}(s_0, s_{1:t+1})]}{\hat{V}(s_0, s_{1:t})} \right\} \leq 1 + \varepsilon$$

*Remark* 3.3. The bellman error characterizes the quality of the reward model in the sense that the perfect reward model should satisfy the optimal twist condition (3), consequently has zero bellman error, and their distinction is therefore naturally characterized by the bellman error itself. We show later in Theorem 4.3 that single-particle guided SMC is exact when the bellman error equals zero, i.e. with a perfect reward model. Additionally, the assumption here takes the form of a *local bellman error*, and a different assumption to be consider is the *global bellman error*, which is adopted by (Rohatgi et al., 2025). We leave the discussion of the latter to Appendix H.

*Remark* 3.4. Assumptions 3.1 and 3.2 are stated in uniform form for a clean non-asymptotic presentation. The first controls the local oscillation of the Feynman–Kac weights, while the second controls the one-step Bellman consistency of the approximate twist. These are precisely the local quantities propagated in our SMC telescoping argument. Since the proof ultimately bounds expectations of local error terms against bounded test functions, the same mechanism can be localized to suitable on-policy or moment versions of these assumptions.

We also remark on the algorithms discussed in this paper.

*Remark* 3.5 (Token-level vs. sentence-level reasoning). In *token-level* reasoning, each step appends a single token, so the base model probability $\pi_{\text{ref}}(s_t|s_0, s_{1:t-1})$ is explicitly available from the transformer and can be evaluated exactly. In *sentence-level* reasoning, a single step appends a longer text span (a whole sentence/segment) as one "reasoning move"; the probability of such a span under $\pi_{\text{ref}}$ is typically not tractable to compute exactly and one usually only has sampling access to it. Algorithms designed for sentence-level reasoning can be applied verbatim to token-level reasoning (by taking each "sentence-step" to be one token). Throughout this paper, we focus on the sentence-level setting.

## 4. Lower Bounds With and Without Approximate Reward Model

In this section, we establish *exponential-in-horizon* lower bounds on the sampling time complexity, both *with* and *without* an approximate reward model. These bounds identify a *small Bellman error* regime as the only setting in which sub-exponential complexity can be achieved, aligning with

the requirement that also emerges from our single-particle guided SMC error analysis. Later in Section 5, Remark 5.3, we compare our lower bound with the upper bounds.

## 4.1. Information-theoretical lower bound

To isolate fundamental limitations, we work with a minimal interface that abstracts away all algorithmic details. Concretely, we allow a sampler to (i) draw next-token samples from the base model given any prefix, and (ii) query the reward model on prefixes. The resulting lower bounds depend only on this information access and therefore hold for any inference-time sampling strategy. We formalize this access pattern, and the corresponding lower bound result through an oracle model.

**Oracle model.** Fix $B \in \mathbb{N}$ and horizon $T$. Let $U := \bigcup_{t=0}^{T-1} [B]^t$ be the set of prefixes. An oracle is a table $o = \{o(u)[k]\}_{u \in U, k \geq 1} \in [B]^{U \times \mathbb{N}}$. Equip $\mathcal{O}_{B,T} := [B]^{U \times \mathbb{N}}$ with the product $\sigma$-algebra. Given $\pi_{\text{ref}}(\cdot \mid u)$ on $[B]$ for each $u \in U$, let $\mathbb{P}_{\pi_{\text{ref}}}$ be the product measure under which $\{o(u)[k]\}_{k \geq 1}$ are i.i.d. with law $\pi_{\text{ref}}(\cdot \mid u)$ for each fixed $u$, and are independent across distinct $u$.

**Inputs and targets.** An input is $I = \langle B, \pi_{\text{ref}}, \hat{V} \rangle$ where $\hat{V}$ is a reward model on prefixes with $\hat{V}(\varnothing) = 1$. It induces a target distribution on leaves $p_I(s_{1:T}) \propto \pi_{\text{ref}}(s_{1:T}) \hat{V}(s_{1:T})$.

**Algorithms and output laws.** A deterministic algorithm a takes $(B, \hat{V})$ and oracle access to $o$, adaptively queries finitely many coordinates $o(u)[k]$, and outputs $\widehat{S}_{1:T} \in [B]^T$. Equivalently, it induces a measurable map $\mathsf{a} : \hat{\mathcal{V}} \times \mathbb{N} \times \mathcal{O}_{B,T} \to [B]^T$. A randomized algorithm $A$ is a distribution $R \in \Delta(\mathcal{A})$ over such a's. Let $\mathcal{A}$ denote those satisfying the *no-guess* constraint: whenever a outputs $\widehat{S}_{1:T}$, it must have queried along that path, i.e. every prefix $\widehat{S}_{1:p}$ for all $p \leq T$. For $A \in \Delta(\mathcal{A})$ and input $I$, define the output distribution

$$q_{A,I}(\cdot) := \mathbb{E}_{\mathsf{a} \sim R} \left[ \mathbb{P}_{o \sim \mathbb{P}_{\pi_{\text{ref}}}} \big( \mathsf{a}(\hat{V}, B, o) \in \cdot \big) \right].$$

**Theorem 4.1** (LB1: worst-case time complexity). *Any randomized algorithm $A \in \mathcal{A}$ such that for every input $I = \langle B, \pi_{\text{ref}}, \hat{V} \rangle$ satisfying Assumption 3.1 with $L > 1$, the output leaf distribution obeys $\|q_{A,I} - p_I\|_{\text{TV}} \leq \frac{1}{3}$, if $A$ always stops before a time complexity TC, then $\text{TC} = \Omega(L^{2T/3})$.*

**Corollary 4.2** (LB2: worst-case complexity under a Bellman-error bound). *Any randomized algorithm $A \in \mathcal{A}$ such that for every input $I = \langle B, \pi_{\text{ref}}, \hat{V} \rangle$ satisfying Assumption 3.1 and 3.2 with $L > 1$ and $\varepsilon \in (0, L-1]$, the output leaf distribution obeys $\|q_{A,I} - p_I\|_{\text{TV}} \leq \frac{1}{3}$, if $A$ always stops before a time complexity TC, then $\text{TC} = \Omega\big((1+\varepsilon)^{2T/3}\big)$.*

*Sketch of proof.* See Appendix C for details. We build a hard family $\{I_u\}$ such that for each $u$ the target leaf distribution $p_{I_u}$ places $> 1/2$ mass on an event $A_u \subseteq [B]^T$ (the set of maximal-probability leaves). Hence $\|q_{A,I_u} - p_{I_u}\|_{\text{TV}} \leq 1/3$ forces any valid algorithm to hit $A_u$ with constant probability. By Yao's min-max principle, it suffices to bound the *average* success probability of any deterministic algorithm a under a random hard input $U \sim \text{Unif}([B]^{2T/3})$. With time budget $Q$ and the no-guess constraint, a can only "try" at most $Q$ candidate leaves/prefix-paths, so by a union bound $\Pr[\mathsf{a} \text{ hits } A_U] \leq Q/B^{2T/3}$. Therefore achieving constant success probability requires $Q = \Omega(B^{2T/3})$, which gives $\text{TC} = \Omega(L^{2T/3})$ (and $\Omega((1+\varepsilon)^{2T/3})$ under Assumption 3.2). □

## 4.2. What can we learn from single-particle guided SMC

A common and naive inference-time guidance primitive performs *local, step-wise reward-tilted sampling* and appears under different names across communities, including *action-level sampling* (Yang & Klein, 2021; Rohatgi et al., 2025), *locally constrained decoding* (Loula et al., 2025; Shin et al., 2021; Scholak et al., 2021; Poesia et al., 2022; Willard & Louf, 2023; Ugare et al., 2024), and *value-based decoding / SVDD* in diffusion guidance (Li et al., 2024; Uehara et al., 2025). To avoid terminology overload, we refer to this primitive as *single-particle guided SMC* (SP-gSMC), since it is exactly the $N = 1$ specialization of an SMC sampler using the following optimal proposal (Appendix D): $\pi_{\text{p}}(s_{t+1}|s_0, s_{1:t}) \propto \pi_{\text{ref}}(s_{t+1}|s_0, s_{1:t}) \hat{V}(s_0, s_{1:t+1})$.

Let $\hat{\pi}_t(\cdot|s_0)$ denote the resulting joint distribution on $s_{1:t}$ generated by sequentially sampling $s_k \sim \hat{\pi}(\cdot \mid s_0, s_{1:k-1})$ for $k = 1, \dots, t$. Thus, single-particle guided SMC can be viewed as the single-trajectory analogue of the optimal-proposal SMC update, bridging particle-based guidance and chain-based approach later introduced in Section 6. Also later in Remark D.3, we further discuss the efficiency of single-particle guided SMC comparing to its multi-particle counterpart, subject to the bellman error being small enough.

We have the following TV error control regarding the single-particle guided SMC.

**Theorem 4.3** (SP-gSMC TV error). *Assume Assumption 3.2 holds. Then for every $t \in [T]$, $\big\|\tilde{\pi}_t(\cdot|s_0) - \hat{\pi}_t(\cdot|s_0)\big\|_{\text{TV}} \leq 2t\varepsilon$.*

This result shows that, under naive guidance, the single-particle guided SMC output distribution $\hat{\pi}_T$ moves closer to the target $\tilde{\pi}_T$ (i.e., achieves $D_{\text{TV}}(\hat{\pi}_T, \tilde{\pi}_T) < 1$) only when the Bellman error satisfies $\varepsilon < \frac{1}{2T}$, equivalently $\varepsilon = O(1/T)$. Once $\varepsilon$ exceeds this threshold, the naive guidance becomes too weak to produce a meaningful improvement in total variation. Moreover, if the target sampling accuracy is $\delta_{\text{TV}}$, then the naive single-particle guided SMC cannot

attain the desired TV error if $\varepsilon \geq \frac{\delta_{\mathrm{TV}}}{2T}$, motivating the need to consider alternative algorithms beyond naive guidance.

The small bellman error $\varepsilon = O(1/T)$ regime also coincides with the exponential lower bound in Corollary 4.2, which indicates that this regime is critical for a sub-exponential complexity guarantee; see Remark 5.3 for further details.

## 5. Complexity of SMC based Inference Time Scaling

In this section, we analyze the computational complexity of the SMC sampler with respect to the number of particles, the time horizon, and the cost of Markov propagation and reward evaluation. This analysis is carried out under an LLM inference time-scaling regime, where performance is governed by the number of reward-model evaluations and the extent of the trajectory search.

### 5.1. SMC complexity

The idea of a SMC sampler is as mentioned in Section 3, using a particle system to mimic the Feynman-Kac flow. We design the Feynman-Kac model with a terminal step such that the FK flow of which matches our target at step $T + 1$, i.e. $\eta_{T+1} = \tilde{\pi}_T$. Accordingly, the sampling distribution of our SMC sampler is defined as $\hat{\pi}_T := \bar{\eta}_{T+1}^N := \mathbb{E}[\eta_{T+1}^N]$, namely, we output a particle *after* this terminal resampling step. Specifically, detailed algorithms for SMC with the naive proposal is in Algorithm 5. As a first step towards time complexity, we propose the following theorem about particle complexity, i.e. the number of particles to maintain for an SMC sampler to sample within a desired accuracy.

**Theorem 5.1** (Particles Complexity). *Fix the horizon $T \geq 2$ and a target accuracy $\delta_{\mathrm{TV}} \in (0, 1)$. If Assumptions 3.1 and 3.2 hold with $L, \varepsilon > 0$, let $\bar{\eta}_{T+1}^N := \mathbb{E}[\eta_{T+1}^N]$ be the output distribution of an SMC sampler (after the terminal resampling step $T+1$ with $M_{T+1} = \mathrm{Id}$). Then the following sufficient conditions guarantee $\|\bar{\eta}_{T+1}^N - \eta_{T+1}\|_{\mathrm{TV}} \leq \delta_{\mathrm{TV}}$ under naive proposal. That is, the number of particles $N \geq \frac{L^6 \, T \, (1+\varepsilon)^{6(T-1)}}{2 \, \delta_{\mathrm{TV}}}$.*

The proof of this theorem uses the following Theorem E.6, along with Lemma E.2. The detained proof can be found in Appendix E.

We also provide Theorem D.1 and Proposition E.7 that generalize the particle complexity result to arbitrary proposal. Note that the particle complexity $N$ naturally induces an $NT$-type baseline cost for SMC-style algorithms, since one must propagate $N$ particles across $T$ steps (with re-sampling/reweighting performed at each step). Additional overhead depends on the per-step cost of (i) sampling from the proposal $\pi_{\mathrm{p}}$ and (ii) computing the importance ratio $\pi_{\mathrm{ref}}/\pi_{\mathrm{p}}$ (and any auxiliary quantities used in reweighting).

These results therefore allow flexibility to balance proposal difficulty, importance-weight computation, and the required number of particles; see Remark E.8.

Although additional computational cost may arise for more sophisticated proposals $\pi_{\mathrm{p}}$, the naive-proposal SMC admits a direct particle-to-time conversion: each of the $T$ steps performs $O(N)$ work, hence Theorem 5.1 immediately implies the following time complexity corollary.

**Corollary 5.2** (Time complexity for naive proposal SMC). *Under Assumption 3.1 and 3.2, and we consider the regime in which $\varepsilon = O(1/T)$. The time complexity for a SMC sampler with naive proposal to achieve $\delta_{\mathrm{TV}}$ accuracy, i.e. $\|\tilde{\pi}_T - \hat{\pi}_T\| \leq \delta_{\mathrm{TV}}$ is $O\left(\frac{L^6 T^2}{\delta_{\mathrm{TV}}}\right)$.*

With a matching algorithmic guarantee in hand, we can now relate our complexity upper bound achieved by the SMC sampler with naive proposal to the worst-case limitations captured by our lower bounds.

*Remark* 5.3 (Lower vs. upper bounds). Our LB2 shows that with an approximate reward/value model (captured by a Bellman-error bound $\varepsilon$), any sampler still needs worst-case $\mathrm{TC} = \Omega\left((1 + \varepsilon)^{2T/3}\right)$ to achieve constant TV accuracy. This highlights the key gain of approximation: the exponential base becomes $1 + \varepsilon$, which can be made arbitrarily close to 1 as the reward model improves, even if the ratio bound constant $L$ is large. In contrast, without such approximation (only a ratio bound), the best available lower bound remains $\Omega(L^{2T/3})$.

Conversely, our upper bounds attain this favorable base (up to constant powers): the particle-complexity Theorem 5.1 yield $N$ scaling as $(1 + \varepsilon)^{O(T)}$ (times $\mathrm{Poly}(T, L, \delta_{\mathrm{TV}}^{-1})$), matching LB2 up to constant-factor gaps in the exponent. The same qualitative $(1 + \varepsilon)^{O(T)}$ dependence also holds for single-particle guided SMC with Metropolis-Hastings correcting (or SP-gSMC+MH, see the proof of Theorem 6.1 and G.2) which we will introduce in the next section. In the regime $\varepsilon = O(1/T)$, we have $(1+\varepsilon)^{O(T)} = O(1)$, so these exponential factors are absorbed and the overall complexity becomes polynomial in $(T, \delta_{\mathrm{TV}}^{-1})$.

### 5.2. On the choice of McKean kernels.

Our main TV guarantees are derived from Theorem E.6, which is *kernel-agnostic*: it provides a uniform upper bound that holds for any McKean transition kernel $K_{p,\eta}$ satisfying the consistency condition $\eta K_{p+1,\eta} = \Phi_{p+1}(\eta)$, and all of our previous results build on this worst-case control. However, the same FK model can admit many different McKean kernels, and their particle systems may exhibit different second-order behavior (hence potentially improved rates in structured cases). For completeness, Appendix E develops a *kernel-sensitive* refinement (Proposition E.9) and illustrates it with a concrete example of a *quantile-based stratified* tran-

sition (Corollary E.10), which can be viewed as a special stratification rule adapted to the quantiles of a designated function. Stratified resampling schemes are classical in the particle filtering literature; see, e.g., (Douc & Cappé, 2005).

## 6. Beyond SMC: Chain based approach

In this section, we will discuss another popular family of inference time scaling algorithm (Li et al., 2024; Uehara et al., 2025; Shin et al., 2021; Scholak et al., 2021; Poesia et al., 2022; Willard & Louf, 2023; Ugare et al., 2024; Rohatgi et al., 2025) other than SMC based sampling. These family of algorithms takes single-particle guided SMC as a starting point, using various methods such as backtracking (Rohatgi et al., 2025) or Metropolis-Hastings (Karan & Du, 2025) to correct the biased sampler. These corrections induce Markov chain mixing towards the target distribution, resulting in a sampling distribution arbitrarily close to the target distribution. This viewpoint treats sampling as a *mixing* problem rather than a pure Monte-Carlo averaging problem whose accuracy is dominated by the control of variance. Thus it will lead to a qualitatively different dependence on the desired TV accuracy $\delta_{\mathrm{TV}}$. At a high level, we consider two ways to build an MH-corrected SP-gSMC sampler, where we focus on the resampling-based proposal for SP-gSMC in this section, resembling the SVDD (Li et al., 2024; Uehara et al., 2025) in the diffusion guidance. The other method is discussed in detail in Appendix G.

### 6.1. Resampling-based SP-gSMC sampling with Metropolis-Hastings correction

**Resampling-pool MH.** The *ideal* SP-gSMC proposal at step $t$ samples $s_t \sim \pi_{\mathrm{ref}}(\cdot|s_0, s_{1:t-1})$ tilted by the local score $\hat{V}(s_0, s_{1:t})$. We approximate this step using a finite *resampling pool* of size $M$: draw $M$ i.i.d. candidates $s_t^{(1:M)} \sim \pi_{\mathrm{ref}}(\cdot|s_0, s_{1:t-1})$, compute their scores $v_t^{(j)}$ using reward model, form the empirical normalizer $\bar{Z}_t = \frac{1}{M}\sum_{j=1}^{M} v_t^{(j)}$, and select an index $A_t \sim \mathsf{Cat}(v_t^{(1:M)}/\sum_{\ell=1}^{M} v_t^{(\ell)})$, setting $s_t = s_t^{(A_t)}$. Although the resulting marginal law of $s_t$ is only an approximation of the ideal tilt when $M < \infty$, the *entire* randomized procedure has an exactly known probability on the *augmented* space that includes the pool and index variables. Consequently, the likelihood ratio between the target (reward-tilted) path law and this resampling-pool proposal is available in closed form: along a proposed trajectory we can update $w \leftarrow w \cdot \hat{V}(s_0, s_{1:t})/\bar{Z}_t$, so that the final $w$ is the exact Radon–Nikodym correction induced by replacing $Z_t$ with $\bar{Z}_t$. We then apply a standard MH accept–reject step *at the end of the trajectory* using this exact ratio, yielding a valid MH kernel with invariant distribution proportional to $\pi_{\mathrm{ref}}(s_{1:T}|s_0)\,\hat{V}(s_0, s_{1:T})$. Algorithm 2 summarizes the full

---

**Algorithm 2** SP-gSMC $M$-Resampling MH correction

1: **Input:** $T, H, M, \hat{V}, \pi_{\mathrm{ref}}, s_0, \delta$.
2: Init $(s_{1:T}^{\mathrm{acc}}, w^{\mathrm{acc}}) \leftarrow (\varnothing, 1)$.
3: **for** $h = 1, \ldots, H$ **do**
4:     Init $(s_{1:T}, w) \leftarrow (\varnothing, 1)$.
5:     **for** $t = 1, \ldots, T$ **do**
6:         Sample $s_t^{(j)} \overset{iid}{\sim} \pi_{\mathrm{ref}}(\cdot|s_0, s_{1:t-1})$, $j = 1, \ldots, M$.
7:         $v_t^{(j)} \leftarrow \hat{V}(s_0, (s_{1:t-1}, s_t^{(j)}))$, $\bar{Z}_t \leftarrow \frac{1}{M}\sum_{j=1}^{M} v_t^{(j)}$.
8:         Draw $A_t \sim \mathsf{Cat}\left(\frac{v_t^{(1:M)}}{\sum_{\ell=1}^{M} v_t^{(\ell)}}\right)$; set $s_t \leftarrow s_t^{(A_t)}$.
9:         $w \leftarrow w \cdot \frac{\hat{V}(s_0, s_{1:t})}{\bar{Z}_t}$.
10:     **end for**
11:     **if** $h = 1$ **then**
12:         $(s_{1:T}^{\mathrm{acc}}, w^{\mathrm{acc}}) \leftarrow (s_{1:T}, w)$; **continue**
13:     **end if**
14:     Draw $\alpha \sim \mathsf{Unif}[0, 1]$.
15:     **if** $\alpha < 1 \wedge \frac{w^{\mathrm{acc}}\hat{V}(s_{1:T})}{w\,\hat{V}(s_{1:T}^{\mathrm{acc}})}$ **then**
16:         $(s_{1:T}^{\mathrm{acc}}, w^{\mathrm{acc}}) \leftarrow (s_{1:T}, w)$.
17:     **end if**
18: **end for**
19: **Output:** $s_{1:T}^{\mathrm{acc}}$.

---

procedure over a total of $H$ MH iterations.

Then, the first result shows that, despite using a finite resampling pool $M$, we can retain an exponentially fast mixing behavior in $\delta_{\mathrm{TV}}$.

**Theorem 6.1.** *For any* $0 < \delta \lesssim \delta_{\mathrm{TV}}$, *using time* $\tilde{O}(LT^3 \log(\delta^{-1})\log(\delta_{\mathrm{TV}}^{-1}))$, *there exist a good event* $\mathcal{E}$ *with* $\mathbb{P}(\mathcal{E}) \geq 1 - \delta$, *conditioning on which Algorithm 2 achieves* $\delta_{\mathrm{TV}}$ *accuracy, i.e.* $\|\tilde{\pi}_T - \hat{\pi}_T|_{\mathcal{E}}\|_{\mathrm{TV}} \leq \delta_{\mathrm{TV}}$.

The proof can be found in Appendix F.

**Why this is possible.** The key point is that the resampling procedure admits an *exact probability computation on an augmented space*: even though the marginal proposal on tokens is only approximate when $M < \infty$, the algorithm keeps track of the pool-dependent likelihood ratio (via $\bar{Z}_t$'s), so the MH acceptance ratio is *exact*. Under Assumptions 3.1–3.2, the acceptance probability is furthermore bounded away from 0 and 1 on a good event, which implies a *constant-order Dobrushin contraction coefficient*. As a result, $H = O(\log(\delta_{\mathrm{TV}}^{-1}))$ MH steps suffice. Finally, since we only need a constant-order contraction, it is enough to estimate each $\bar{Z}_t$ to relative error $O(1/T)$, which is achieved with $M = O(T^2)$ samples.

## 6.2. Relation to multi-particle SMC and Takeaway

The single-particle guided SMC viewpoint clarifies that chain-based methods and particle methods share the same underlying update: both implement a one-particle SMC step, and differ mainly in *how* additional computation is used to drive the error down. This perspective provides a convenient bridge between *particle-based* guidance, which reduces error by increasing the number of particles, and *chain-based* guidance, which reduces error via Markov chain mixing. SMC can achieve arbitrary $\delta_{\text{TV}}$ accuracy by scaling the particle count, typically leading to a polynomial dependence on $\delta_{\text{TV}}^{-1}$. By contrast, Metropolis–Hastings corrections target the same limit through *mixing*: the relevant computational knob is the number of MH steps $H$. When the acceptance probability is exact and uniformly bounded on a high-probability good event, the resulting kernel contracts geometrically in TV, yielding $H = O\left(\log(\delta_{\text{TV}}^{-1})\right)$. Our resampling-pool MH construction preserves this logarithmic dependence, whereas the rejection-based construction discussed in Appendix G generally loses it due to linear accumulation of acceptance-ratio *estimation* errors.

## Impact Statement

This paper presents work whose goal is to advance the theoretical understanding of inference-time scaling methods in machine learning. The techniques studied here are general-purpose algorithmic and analytical tools, and we do not anticipate any immediate negative societal or ethical consequences beyond those already well known for machine learning research. We therefore believe that no additional discussion of broader impacts is required.

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

## Table of Contents

# A. Notations

| Symbol | Meaning |
| --- | --- |
| **Reasoning / LM setup** | |
| $T$ | Horizon (number of reasoning steps). |
| $[T]$ | The set $\{1, 2, \ldots, T\}$. |
| $\mathcal{S}_t$ / $\mathcal{S}_0$ | Reasoning/token space at time $t$ / Initial prompt space. |
| $\mathcal{S}_{1:t}$ | Prefix space up to time $t$, recursively $\mathcal{S}_{1:t} := \mathcal{S}_{1:t-1} \times \mathcal{S}_t$. |
| $s_{1:t}$ / $s_0$ | A prefix (trajectory) in $\mathcal{S}_{1:t}$ / The initial prompt. |
| $\pi_{\mathrm{ref}}(\cdot \mid s_{1:t-1})$ | Reference / proposal policy (e.g., base language model). |
| $\pi_{\mathrm{p}}(\cdot \mid s_{1:t-1})$ | Proposal distribution, the sampler draw sample from this distribution. (naive case being $\pi_{\mathrm{p}} = \pi_{\mathrm{ref}}$) |
| $\phi(s_{1:T})$ | Task-dependent final reward function. |
| $\hat{V}(s_0, s_{1:t})$ | (Imperfect) reward/value model defined on prefixes. $\hat{V}(s_0, s_{1:T}) = \phi(s_{1:T})$ |
| $\tilde{\pi}(s_{1:T}\mid s_0)$ | Tilted target distribution $\tilde{\pi}(s_{1:T}\mid s_0) \propto \pi_{\mathrm{ref}}(s_{1:T}\mid s_0)\,\phi(s_{1:T})$, $\tilde{\pi}_T(s_{1:T}\mid s_0) = \tilde{\pi}(s_{1:T}\mid s_0)$. |
| $\tilde{\pi}_t(s_{1:t}\mid s_0)$ | Tilted intermediate distribution $\tilde{\pi}_t(s_{1:t}\mid s_0) \propto \pi_{\mathrm{ref}}(s_{1:t}\mid s_0)\hat{V}(s_0, s_{1:t})$, $\tilde{\pi}_T(s_{1:T}\mid s_0) = \tilde{\pi}(s_{1:T}\mid s_0)$. |
| $\hat{\pi}_T$ | Sampling distribution of any specified algorithm. For SMC, $\hat{\pi}_T := \bar{\eta}_{T+1}^N := \mathbb{E}[\eta_{T+1}^N]$. |
| $L$ | Reward model ratio bound (Assumption 3.1). |
| $\varepsilon$ | Bellman error bound (Assumption 3.2). |
| $C_{\mathrm{act}}$ | Local one-step ratio constant: $C_{\mathrm{act}} := \sup_{t\in[T]} \sup_{s_{1:t}\in\mathcal{S}_{1:t}} \dfrac{\hat{V}(s_0, s_{1:t})}{\mathbb{E}_{s_t\sim\pi_{\mathrm{ref}}(\cdot\mid s_{1:t-1})}\hat{V}(s_0, s_{1:t})}$, $C_{\mathrm{act}} \leq$ $L(1 + \varepsilon)$ under Assumptions 3.1 and 3.2. |
| $\delta_{\mathrm{TV}}$ | The desired TV error between the sampling distribution $\hat{\pi}_T$ and the target distribution $\tilde{\pi}_T$. |
| **Information-theoretic lower bound** | |
| $B$ | Branching factor (alphabet size) of the abstract token space $[B]$. |
| $U$ | Set of prefixes: $U := \bigcup_{t=0}^{T-1}[B]^t$. |
| $o(u)[k]$ | The $k$-th oracle next-token draw associated with prefix $u \in U$. |
| $\mathcal{O}_{B,T}$ | Oracle space $\mathcal{O}_{B,T} := [B]^{U\times\mathbb{N}}$ (equipped with the product $\sigma$-algebra). |
| $\mathbb{P}_{\pi_{\mathrm{ref}}}$ | Product measure on $\mathcal{O}_{B,T}$ induced by the base model $\pi_{\mathrm{ref}}(\cdot \mid u)$. |
| $I$ | Input instance $I = \langle B, \pi_{\mathrm{ref}}, \hat{V}\rangle$ (with $\hat{V}(\varnothing) = 1$). |
| $p_I$ | Target leaf distribution $p_I(s_{1:T}) \propto \pi_{\mathrm{ref}}(s_{1:T})\hat{V}(s_{1:T})$. |
| $\mathcal{A}$ | Class of (deterministic) algorithms satisfying the *no-guess* constraint. |
| $q_{A,I}$ | Output distribution of a randomized algorithm $A$ on input $I$. |
| **Feynman–Kac model** | |
| $E_t$ | FK state space at time $t$; in our language model setting $E_t := \mathcal{S}_{1:t}$. |
| $\mathcal{P}(E)$ | Probability measures on a measurable space $E$. |
| $M_t(x, dy)$ / $M_t(f)(x)$ | Markov kernel from $E_{t-1}$ to $E_t$ (proposal dynamics), where $M_t(f)(x) = \int f(y)\,M_t(x, dy)$. |
| $G_t : E_t \to [0, \infty)$ | Potential / incremental weight function (often derived from $\hat{V}$). |
| $Q_t$ | Unnormalized FK operator: $Q_t(f)(x) := G_{t-1}(x)\,M_t(f)(x)$. |
| $Q_{p,n}$ | Multi-step unnormalized semigroup: $Q_{p,n} := Q_{p+1}\cdots Q_n$. |
| $\eta_t$ | Limiting FK flow (target) distribution on $E_t$. |
| $\Phi_t$ | Normalized FK map: $\eta_t = \Phi_t(\eta_{t-1})$; more generally $\Phi_t(\mu)(f) = \dfrac{\mu(Q_t(f))}{\mu(Q_t(1))}$. |
| $\Phi_{p,n}$ | Normalized semigroup: $\Phi_{p,n}(\mu)(f) := \dfrac{\mu(Q_{p,n}(f))}{\mu(Q_{p,n}(1))}$. (So $\Phi_{p,n} = \Phi_n \circ \cdots \circ \Phi_{p+1}$.) |
| **Particle system (SMC)** | |
| $N$ | Number of particles. |
| $\xi_t^{(N,i)}$ | The $i$-th within the collection of $N$ particles at time $t$. |
| $\eta_t^N$ | Empirical measure of particles at time $t$ (random approximation of $\eta_t$). |
| $\bar{\eta}_t^N$ | Mean measure $\bar{\eta}_t^N := \mathbb{E}[\eta_t^N]$. |
| $\mathcal{F}_t^N$ | Natural filtration generated by the particle system up to time $t$. |
| $\nu_t^N$ | One-step predictive measure $\nu_t^N := \Phi_t(\eta_{t-1}^N)$. |

| Symbol | Meaning |
| --- | --- |
| **Mixing / stability coefficients and oscillation norms** | |
| $\mathrm{osc}(f)$ | Oscillation: $\sup f - \inf f$. |
| $\mathrm{Osc}_1(E)$ | Unit oscillation class: $\{f : E \to \mathbb{R} \; : \; \mathrm{osc}(f) \leq 1\}$. |
| $\beta(K)$ | Dobrushin coefficient of an integral operator/kernel $K$ (used to control contraction). |
| **FK first-order expansion (used in Appendix)** | |
| $D_\eta \Phi$ | First-order integral operator at base point $\eta$, with $(D_\eta \Phi)(1) = 0$. |
| $\beta(D\Phi)$ | Uniform Dobrushin bound: $\beta(D\Phi) := \sup_\eta \beta(D_\eta \Phi) < \infty$. |
| $R^\Phi(\mu, \eta)$ | Second-order remainder signed measure in the above expansion. |
| **FK semigroup-specific auxiliaries** | |
| $P_{p,n}$ | Markov kernel induced by $Q_{p,n}$: $P_{p,n}(f) := \dfrac{Q_{p,n}(f)}{Q_{p,n}(1)}$. |
| $G_{p,n,\eta}$ | Normalized potential: $G_{p,n,\eta} := \dfrac{Q_{p,n}(1)}{\eta(Q_{p,n}(1))}$. |
| $q_{p,n}$ | Ratio constant: $q_{p,n} := \sup_{x,y \in E_p} \dfrac{Q_{p,n}(1)(x)}{Q_{p,n}(1)(y)} \in [1, \infty]$. |

# B. Algorithms

## B.1. Rejection Sampling

We use rejection sampling as a basic primitive for drawing *(conditionally) exact* samples from *tilted* distributions

$$\nu(\mathrm{d}z) \ \propto \ g(z)\,\mu(\mathrm{d}z),$$

where $\mu$ is a tractable base law (e.g., $\pi_{\mathrm{ref}}(\cdot|s_0, s_{1:t})$) and $g$ is a nonnegative score (e.g., a reward). Conceptually, rejection sampling turns *unnormalized* weights $g$ into *exact sampling* from $\nu$ using only proposals from $\mu$ and accept–reject decisions.

---

**Algorithm 3** Rejection Sampling (Rohatgi et al., 2025)

1: **Input:** function $g : \mathcal{Z} \to \mathbb{R}_{\geq 0}$, base distribution $\mu$, threshold $M > 0$, failure probability $\delta$
2: **Output:** sample $\hat{z} \in \mathcal{Z}$
3: Set $N \leftarrow 4M \log(4/\delta)$
4: Sample $z_1, \ldots, z_N \sim \mu$ i.i.d.
5: Compute $\widehat{Z} \leftarrow \frac{1}{N} \sum_{i=1}^{N} g(z_i)$
6: **for** $i = 1$ to $N$ **do**
7:     Sample $z \sim \mu$
8:     Sample $\xi \sim \mathrm{Bernoulli}\left(\min\left\{\frac{g(z)}{M\widehat{Z}}, 1\right\}\right)$
9:     **if** $\xi = 1$ **then**
10:         **return** $z$
11:     **end if**
12: **end for**
13: Sample $z \sim \mu$
14: **return** $z$ {failure case}

---

The key guarantee is that, although the algorithm uses an empirical normalizer $\widehat{Z}$, its output is *exact on a high-probability event*. This is characterized by the following theorem.

**Theorem B.1** (Theorem E.1 in (Foster et al., 2025))**.** *Let $g : \mathcal{Z} \to [0, \infty)$ and $\mu \in \Delta(\mathcal{Z})$, and define $\nu(\mathrm{d}z) := \frac{g(z)}{Z}\mu(\mathrm{d}z)$ with $Z := \mathbb{E}_{z \sim \mu}[g(z)] \in (0, \infty)$. Let $C_\infty := \left\| \frac{g}{Z} \right\|_\infty$, fix $\delta \in (0, 1)$, and assume $M \geq 4C_\infty$. Run Algorithm 3 with $(g, \mu, M, \delta)$ and let $\hat{z}$ be its output. Then there exists an event $\mathcal{E}_{\mathrm{accept}}$ with $\mathbb{P}(\mathcal{E}_{\mathrm{accept}}) \geq 1 - \delta$ such that $\mathbb{P}(\hat{z} \in \cdot \mid \mathcal{E}_{\mathrm{accept}}) = \nu(\cdot)$.*

*Remark* B.2. Algorithm 3 is a *truncated, self-normalized* variant of textbook rejection sampling: it replaces the unknown normalizer $Z = \mathbb{E}_\mu[g]$ by an empirical estimate $\widehat{Z}$, and runs for at most $N$ accept–reject trials (falling back to $z \sim \mu$ upon failure). Despite these modifications, the output is *exact on a high-probability event* (Theorem B.1): conditioned on $\mathcal{E}_{\mathrm{accept}}$, the returned sample has law $\nu$. Notably, Algorithm 3 has a hyperparameter $M$, which is required to be greater equal to $4C_\infty$. Otherwise the rejection sampling fails mathematically. Therefore, approximate knowledge of $C_\infty$ must be known to preform this algorithm. **If** such $C_\infty$ is known precisely, then the **ideal** overall runtime is $O(C_\infty \log(\delta^{-1}))$.

Suppose an approximate guess of the upper bound of $C_\infty$ is available, denoted as $C_\infty^\star$. Then preforming Algorithm 3 for the purpose of sampling takes a total runtime of $O(C_\infty^\star \log(\delta^{-1}))$, where one conservatively set the hyperparameter $M = 4C_\infty^\star$. A wrong guess may yield $C_\infty^\star \gg C_\infty$, resulting in a waste of computation.

When an algorithm additionally requires estimating the normalizing constant to a small relative error $\epsilon$, we may couple this estimation with Algorithm 3 by using the same budget $N = \Theta(M \log(\delta^{-1}))$ both to form $\widehat{Z}$ and to run truncated accept–reject trials. In this regime, the dominant cost is the normalization estimation, which typically requires $N = \tilde{O}(C_\infty \epsilon^{-2})$. Equivalently, this corresponds to choosing $M = \tilde{O}(C_\infty \epsilon^{-2})$, which (for $\epsilon$ sufficiently small, e.g. in Algorithm 6 or Algorithm 8) automatically implies $M \geq 4C_\infty$, so the envelope requirement for conditional exactness does not impose an additional constraint. In particular, one need not provide a separate prior upper bound $C_\infty^\star$ on $C_\infty$ to ensure that the rejection sampler is well-defined in the target-accuracy regime. Thus, the aforementioned $C_\infty^\star \gg C_\infty$ issue is no longer problematic.

## B.2. Empirical expectation

**Lemma B.3** (Freedman's lemma, e.g. (Agarwal et al., 2014))**.** *Let $(X_t)_{t \leq T}$ be a sequence of real-valued martingale difference sequence adapted to a filtration $(\mathscr{F}_t)_{t \leq T}$. If $|X_t| \leq R$ almost surely, then for any $\eta \in (0, 1/R)$, with probability*

---

**Algorithm 4** Empirical expectation

---
1: **Input:** function $g : \mathcal{S}_{1:t} \to \mathbb{R}_{\geq 0}$, pre-trained moel $\pi_{\text{ref}}$, current state trajectory $s_{1:t}$, MC total step $N_m$.
2: Sample $N_m$ individual samples $s_t^{[i]}, i \in [N_m]$ from $\pi_{\text{ref}}(\cdot|s_0, s_{1:t-1})$.
3: **return** $\frac{1}{N_m} \sum_{i=1}^{N_m} g(s_t^{[i]})$

---

*at least $1 - \delta$,*

$$\sum_{t=1}^{T} X_t \leq \eta \sum_{t=1}^{T} \mathbb{E}_{t-1}[X_t^2] + \frac{\log(\delta^{-1})}{\eta}.$$

*where $\mathbb{E}_t[\cdot] := \mathbb{E}[\cdot|\mathscr{F}_t]$.*

**Lemma B.4.** *Let $(X_t)_{t \leq T}$ be an adapted sequence with $0 \leq X_t \leq R$ almost surely. Write*

$$\mu_t := \mathbb{E}_{t-1}[X_t], \qquad D_t := X_t - \mu_t.$$

*Then for any $\epsilon \in (0,1)$ and $\delta \in (0,1)$, w.p. $\geq 1 - \delta$,*

$$\sum_{t=1}^{T} X_t \leq (1+\epsilon) \sum_{t=1}^{T} \mu_t + \frac{R}{\epsilon} \log \frac{4}{\delta}, \tag{4}$$

$$\sum_{t=1}^{T} \mu_t \leq \frac{1}{1-\epsilon} \sum_{t=1}^{T} X_t + \frac{R}{\epsilon(1-\epsilon)} \log \frac{4}{\delta}. \tag{5}$$

*In particular, when $\epsilon = 1/2$ and we loosen constants, this recovers the original form with coefficients $3/2$ and $2$ up to multiplicative constants.*

*Proof.* Define

$$\mu_t := \mathbb{E}_{t-1}[X_t], \qquad D_t := X_t - \mu_t.$$

Then $(D_t)$ is a martingale difference sequence with respect to $(\mathcal{F}_t)$, and since $0 \leq X_t \leq R$ we have $|D_t| \leq R$. Moreover

$$\mathbb{E}_{t-1}[D_t^2] = \text{Var}(X_t \mid \mathcal{F}_{t-1}) \leq \mathbb{E}_{t-1}[X_t^2] \leq R\mathbb{E}_{t-1}[X_t] = R\mu_t. \tag{$*$}$$

Fix $\epsilon \in (0,1)$ and $\delta \in (0,1)$. We will apply Lemma A.2 to both $(D_t)$ and $(-D_t)$ with suitable parameters and then take a union bound.

**Step 1: Upper bound on $\sum X_t$.** Apply Lemma A.2 to $(D_t)$ with

$$\eta = \frac{\epsilon}{R}, \qquad \delta' = \frac{\delta}{2}.$$

Note that $\eta \in (0, 1/R)$ since $\epsilon \in (0,1)$. Lemma A.2 then yields, with probability at least $1 - \delta/2$,

$$\sum_{t=1}^{T} D_t \leq \eta \sum_{t=1}^{T} \mathbb{E}_{t-1}[D_t^2] + \frac{\log((\delta/2)^{-1})}{\eta}.$$

Using $(*)$ and the choice of $\eta$,

$$\eta \sum_{t=1}^{T} \mathbb{E}_{t-1}[D_t^2] \leq \eta R \sum_{t=1}^{T} \mu_t = \epsilon \sum_{t=1}^{T} \mu_t.$$

Also

$$\frac{\log((\delta/2)^{-1})}{\eta} = \frac{R}{\epsilon} \log \frac{2}{\delta} \leq \frac{R}{\epsilon} \log \frac{4}{\delta}.$$

Hence on an event $\mathcal{E}_1$ with $\Pr(\mathcal{E}_1) \geq 1 - \delta/2$,

$$\sum_{t=1}^{T} D_t \leq \epsilon \sum_{t=1}^{T} \mu_t + \frac{R}{\epsilon} \log \frac{4}{\delta}.$$

Since $\sum X_t = \sum \mu_t + \sum D_t$, we obtain

$$\sum_{t=1}^{T} X_t \leq (1 + \epsilon) \sum_{t=1}^{T} \mu_t + \frac{R}{\epsilon} \log \frac{4}{\delta},$$

which is exactly (4).

**Step 2: Control of $\sum \mu_t$ in terms of $\sum X_t$.** Now apply Lemma A.2 to the martingale difference sequence

$$Y_t := -D_t = \mu_t - X_t,$$

again with the same $\eta = \epsilon/R$ and $\delta' = \delta/2$. We have $|Y_t| \leq R$ and

$$\mathbb{E}_{t-1}[Y_t^2] = \mathbb{E}_{t-1}[D_t^2] \leq R\mu_t$$

as before, so Lemma A.2 implies that on an event $\mathcal{E}_2$ with $\Pr(\mathcal{E}_2) \geq 1 - \delta/2$,

$$\sum_{t=1}^{T} Y_t \leq \eta \sum_{t=1}^{T} \mathbb{E}_{t-1}[Y_t^2] + \frac{\log((\delta/2)^{-1})}{\eta} \leq \epsilon \sum_{t=1}^{T} \mu_t + \frac{R}{\epsilon} \log \frac{4}{\delta}.$$

Since $\sum Y_t = \sum \mu_t - \sum X_t$, this becomes

$$\sum_{t=1}^{T} \mu_t - \sum_{t=1}^{T} X_t \leq \epsilon \sum_{t=1}^{T} \mu_t + \frac{R}{\epsilon} \log \frac{4}{\delta},$$

i.e.

$$(1 - \epsilon) \sum_{t=1}^{T} \mu_t \leq \sum_{t=1}^{T} X_t + \frac{R}{\epsilon} \log \frac{4}{\delta}.$$

Dividing by $1 - \epsilon$ yields

$$\sum_{t=1}^{T} \mu_t \leq \frac{1}{1 - \epsilon} \sum_{t=1}^{T} X_t + \frac{R}{\epsilon(1 - \epsilon)} \log \frac{4}{\delta},$$

which is (5).

**Step 3: Union bound.** Finally, define $\mathcal{E} := \mathcal{E}_1 \cap \mathcal{E}_2$. By a union bound,

$$\Pr(\mathcal{E}) \geq 1 - \Pr(\mathcal{E}_1^c) - \Pr(\mathcal{E}_2^c) \geq 1 - \frac{\delta}{2} - \frac{\delta}{2} = 1 - \delta.$$

On $\mathcal{E}$, both (4) and (5) hold, so the lemma follows. $\qquad \square$

**Lemma B.5.** *Suppose $N_m \geq 8C_{\mathrm{act}} \log \frac{4}{\delta}$. With probability greater than $1 - \delta$, the relative error of estimating expectation in Algorithm 4 with $g := \hat{V}$ is*

$$1 - 2\sqrt{2}\sqrt{\frac{C_{\mathrm{act}} \log \frac{4}{\delta}}{N_m}} \leq \frac{\tilde{V}^{N_m}}{\mathbb{E}_{s_t \sim \pi_{\mathrm{ref}}(\cdot | s_{1:t-1}, s_0)}[\hat{V}(s_0, s_{1:t})]} \leq 1 + 2\sqrt{2}\sqrt{\frac{C_{\mathrm{act}} \log \frac{4}{\delta}}{N_m}},$$

*where $\tilde{V}^{N_m} := \frac{1}{N_m} \sum_{i=1}^{N_m} \hat{V}(s_0, (s_{1:t-1}, s_t^{[i]}))$ and $C_{\mathrm{act}} := \sup_{t \in [T]} \sup_{s_{1:t} \in \mathcal{S}_{1:t}} \frac{\hat{V}(s_0, s_{1:t})}{\mathbb{E}_{s_t \sim \pi_{\mathrm{ref}}(\cdot | s_{1:t-1})} \hat{V}(s_0, s_{1:t})}.$*

---

**Algorithm 5** Naive-proposal SMC (with $V(s_0) := 1$, start from $\eta_1$, terminal $t = T{+}1$ with $M_{T+1} = \mathrm{Id}$)

---

1: **Input:** prompt $s_0$, horizon $T$, #particles $N$, base model $\pi_{\mathrm{ref}}$, score $V$.
2: **Convention:** $V(s_0) := 1$ and $V(S_{1:0}) := V(s_0) = 1$.
3: **Init ($t = 1$):** sample $S_1^i \sim \eta_1(\cdot) = \pi_{\mathrm{ref}}(\cdot \mid s_0)$ i.i.d. for $i = 1, \dots, N$.
4: **for** $t = 2, 3, \dots, T$ **do**
5:      **Weights:** $w_{t-1}^j \leftarrow \dfrac{V(S_{1:t-1}^j)}{V(S_{1:t-2}^j)}, \quad \bar{w}_{t-1}^j \leftarrow w_{t-1}^j / \sum_{\ell=1}^N w_{t-1}^\ell$.
6:      **Resample:** $A_{t-1}^i \sim \mathrm{Cat}(\bar{w}_{t-1}^{1:N})$ i.i.d. for $i = 1, \dots, N$.
7:      **Propagate (naive proposal):** sample $S_t^i \sim \pi_{\mathrm{ref}}(\cdot \mid s_0, S_{1:t-1}^{A_{t-1}^i})$, set $S_{1:t}^i \leftarrow (S_{1:t-1}^{A_{t-1}^i}, S_t^i)$.
8: **end for**
9: **Terminal ($t = T{+}1$, $M_{T+1} = \mathrm{Id}$):** $w_T^j \leftarrow \dfrac{V(S_{1:T}^j)}{V(S_{1:T-1}^j)}, \bar{w}_T^j \leftarrow w_T^j / \sum_{\ell=1}^N w_T^\ell$.
10: **Resample:** $A_T^i \sim \mathrm{Cat}(\bar{w}_T^{1:N})$ i.i.d. for $i = 1, \dots, N$.
11: **Identity propagate:** set $S_{1:T}^i \leftarrow S_{1:T}^{A_T^i}$ for all $i = 1, \dots, N$.
12: Draw $I \sim \mathrm{Unif}(\{1, \dots, N\})$ and **Output** $S_{1:T}^I$.

---

*Proof of Lemma B.5.* Applying Lemma B.4 for any fixed prefix $s_{1:t-1}$, we get with probability at least $1 - \delta$,

$$\tilde{V}^{N_m} \leq (1 + \epsilon)\mathbb{E}[\hat{V}(s_0, s_{1:t})] + \frac{\sup_{s_t} \hat{V}(s_0, s_{1:t})}{N_m \epsilon} \log \frac{4}{\delta},$$

$$(1 - \epsilon)\mathbb{E}[\hat{V}(s_0, s_{1:t})] \leq \tilde{V}^{N_m} + \frac{\sup_{s_t} \hat{V}(s_0, s_{1:t})}{N_m \epsilon(1 - \epsilon)} \log \frac{4}{\delta}.$$

Thus the relative error satisfies:

$$1 - \epsilon - \frac{C_{\mathrm{act}}}{N_m \epsilon(1 - \epsilon)} \log \frac{4}{\delta} \leq \frac{\tilde{V}^{N_m}}{\mathbb{E}[\hat{V}(s_0, s_{1:t})]} \leq 1 + \varepsilon + \frac{C_{\mathrm{act}}}{N_m \epsilon} \log \frac{4}{\delta},$$

and balancing the error we have for any $N_m \geq 8 C_{\mathrm{act}} \log \dfrac{4}{\delta}$

$$1 - 2\sqrt{2}\sqrt{\frac{C_{\mathrm{act}} \log \frac{4}{\delta}}{N_m}} \leq \frac{\tilde{V}^{N_m}}{\mathbb{E}_{s_t \sim \pi_{\mathrm{ref}}(\cdot \mid s_{1:t-1}, s_0)}[\hat{V}(s_0, s_{1:t})]} \leq 1 + 2\sqrt{2}\sqrt{\frac{C_{\mathrm{act}} \log \frac{4}{\delta}}{N_m}}$$

$\square$

---

**Algorithm 6** Optimal-proposal SMC (RS proposal + MC reweight; terminal $t = T+1$ with $M_{T+1} = \text{Id}$)

---

1: **Input:** prompt $s_0$, horizon $T$, #particles $N$, MC budget $N_m$, base model $\pi_{\text{ref}}$, score $V$, RS threshold $M$, RS failure prob. $\delta$.

2: **Convention:** $V(s_0) := 1$ and $V(S_{1:0}) := 1$.

3: **Init ($t = 1$) via RS:** for each $i = 1, \dots, N$, set

$$S_1^i \leftarrow \text{RejectionSampling}\Big(g_1(\cdot) = V(\cdot),\ \mu = \pi_{\text{ref}}(\cdot \mid s_0),\ M,\ \delta/O(NT)\Big),$$

where RejectionSampling is Algorithm 3.

4: **for** $t = 2, 3, \dots, T$ **do**

5:     **for** $j = 1, \dots, N$ **do**

6:         **MC estimate of $Z_{t-1}$ on prefix $S_{1:t-2}^j$:**

$$\widehat{Z}_{t-1}(S_{1:t-2}^j) \leftarrow \frac{1}{N_m} \sum_{m=1}^{N_m} V\big(S_{1:t-2}^j, \tilde{s}_{t-1}^{(j,m)}\big), \quad \tilde{s}_{t-1}^{(j,m)} \sim \pi_{\text{ref}}(\cdot \mid s_0, S_{1:t-2}^j) \text{ i.i.d.}$$

7:         **Weight (prefix shifted back by one):**

$$w_{t-1}^j \leftarrow G_{t-1}(S_{1:t-1}^j) := \frac{\widehat{Z}_{t-1}(S_{1:t-2}^j)}{V(S_{1:t-2}^j)}.$$

8:     **end for**

9:     Normalize $\bar{w}_{t-1}^j \leftarrow w_{t-1}^j / \sum_{\ell=1}^N w_{t-1}^\ell$.

10:     **Resample:** $A_{t-1}^i \sim \text{Cat}(\bar{w}_{t-1}^{1:N})$ i.i.d. for $i = 1, \dots, N$.

11:     **Propagate via RS (optimal proposal):** for each $i = 1, \dots, N$, set

$$S_t^i \leftarrow \text{RejectionSampling}\Big(g_t(\cdot) = V(S_{1:t-1}^{A_{t-1}^i}, \cdot),\ \mu = \pi_{\text{ref}}(\cdot \mid s_0, S_{1:t-1}^{A_{t-1}^i}),\ M,\ \delta/O(NT)\Big),$$

    and set $S_{1:t}^i \leftarrow (S_{1:t-1}^{A_{t-1}^i}, S_t^i)$.

12: **end for**

13: **Terminal ($t = T+1$, $M_{T+1} = \text{Id}$):**

14: **for** $j = 1, \dots, N$ **do**

15:     **MC estimate of $Z_T$ on prefix $S_{1:T-1}^j$:**

$$\widehat{Z}_T(S_{1:T-1}^j) \leftarrow \frac{1}{N_m} \sum_{m=1}^{N_m} V\big(S_{1:T-1}^j, \tilde{s}_T^{(j,m)}\big), \quad \tilde{s}_T^{(j,m)} \sim \pi_{\text{ref}}(\cdot \mid s_0, S_{1:T-1}^j) \text{ i.i.d.}$$

16:     **Terminal weight:**

$$w_T^j \leftarrow G_T(S_{1:T}^j) := \frac{\widehat{Z}_T(S_{1:T-1}^j)}{V(S_{1:T-1}^j)}.$$

17: **end for**

18: Normalize $\bar{w}_T^j \leftarrow w_T^j / \sum_{\ell=1}^N w_T^\ell$.

19: **Resample:** $A_T^i \sim \text{Cat}(\bar{w}_T^{1:N})$ i.i.d. for $i = 1, \dots, N$.

20: **Identity propagate:** set $S_{1:T}^i \leftarrow S_{1:T}^{A_T^i}$ for all $i = 1, \dots, N$.

21: Draw $I \sim \text{Unif}(\{1, \dots, N\})$ and **Output** $S_{1:T}^I$.

---

---

**Algorithm 7** Action-level $M$-Resampling with Metropolis–Hastings Corrections: Detailed

---

1: **Input:** Horizon $T$, number of Metropolis–Hastings steps $H$, resampling batch size $M$, reward model $\hat{V}$, base model $\pi_{\text{ref}}$, initial prompt $s_0$, failure probability $\delta$.
2: Initialize accepted trajectory $s_{1:T}^{\text{acc}} \leftarrow \varnothing$ and its weight $w^{\text{acc}} \leftarrow 1$.
3: **for** $h = 1, \ldots, H$ **do**
4:      Initialize proposal trajectory container $s_{1:T}^{[h]} \leftarrow \varnothing$ and weight $w^{[h]} \leftarrow 1$.
5:      **for** $t = 1, \ldots, T$ **do**
6:          Draw $M$ candidates $\{s_t^{(h,j)}\}_{j=1}^M \overset{iid}{\sim} \pi_{\text{ref}}(\cdot \mid s_0, s_{1:t-1}^{[h]})$.
7:          Compute rewards $v_t^{(h,j)} := \hat{V}\big(s_0, (s_{1:t-1}^{[h]}, s_t^{(h,j)})\big)$ for $j = 1, \ldots, M$.
8:          Form empirical normalizer $\bar{Z}_t^{[h]} := \frac{1}{M} \sum_{j=1}^M v_t^{(h,j)}$.
9:          Resample one action index $A_t \sim \text{Cat}\left(\left\{\frac{v_t^{(h,j)}}{\sum_{\ell=1}^M v_t^{(h,\ell)}}\right\}_{j=1}^M\right)$ and set $s_t^{[h]} \leftarrow s_t^{(h,A_t)}$.
10:          Update proposal weight $w^{[h]} \leftarrow w^{[h]} \cdot \frac{\hat{V}(s_0, s_{1:t}^{[h]})}{\bar{Z}_t^{[h]}}$.
11:      **end for**
12:      **if** $h = 1$ **then**
13:          Set $s_{1:T}^{\text{acc}} \leftarrow s_{1:T}^{[h]}$ and $w^{\text{acc}} \leftarrow w^{[h]}$.
14:          **continue for**
15:      **end if**
16:      Draw $\alpha \sim \text{Unif}[0, 1]$.
17:      **if** $\alpha < 1 \wedge \frac{w^{\text{acc}} \hat{V}(s_{1:T}^{[h]})}{w^{[h]} \hat{V}(s_{1;T}^{\text{acc}})}$ **then**
18:          Set $s_{1:T}^{\text{acc}} \leftarrow s_{1:T}^{[h]}$ and $w^{\text{acc}} \leftarrow w^{[h]}$.
19:      **end if**
20: **end for**
21: **Output:** $s_{1:T}^{\text{acc}}$.

---

**Algorithm 8** Action-level sampling with Metropolis-Hastings corrections

---

1: **Input:** Total horizon $T$, number of particles $N$, number of Metropolis Hasting steps $H$, reward model $\hat{V}$, pre-trained base model $\pi_{\text{ref}}$, initial prompt $s_0$, failure event probability $\delta$.
2: Initialize current acceptance trajectory container $s_{1:T}^{\text{acc}} \leftarrow \varnothing$
3: **for** $h = 1, \cdots, H$ **do**
4:      Initialize cumulative rejection weight $w \leftarrow 1, w' \leftarrow 1$
5:      **for** $t = 1, \cdots, T$ **do**
6:          $s_t^{[h]} \leftarrow$ **Call** Rejection Sampling (Algorithm 3)
         $(\hat{V}(s_0, (s_{1:t-1}, \cdot)), \pi_{\text{ref}}(\cdot|s_0, s_{1:t-1}), O(L) > C_{\text{act}}, O(\delta/T \log(\delta^{-1})))$     *We specify* $s_{1:0} := \varnothing$.
7:          $w \leftarrow w \cdot \hat{V}(s_{1:t}^{[h]}) / \hat{\mathbb{E}}_{s' \sim \pi_{\text{ref}}(\cdot|s_0, s_{1:t-1}^{[h]})}[\hat{V}(s_0, (s_{1:t-1}^{[h]}, s_t'))]$     $\hat{\mathbb{E}}$ *is the empirical expectation from Algorithm 4.*
8:          $w' \leftarrow w' \cdot \hat{V}(s_{1:t}^{\text{acc}}) / \hat{\mathbb{E}}_{s_t' \sim \pi_{\text{ref}}(\cdot|x_0, s_{1:t-1}^{\text{acc}})}[\hat{V}(s_0, (s_{1:t-1}^{\text{acc}}, s_t'))]$     *If* $s_{1:T}^{\text{acc}} = \varnothing$, *we define* $\hat{Q}(\varnothing) \equiv 1$
9:      **end for**
10:      **if** $h = 1$ **then**
11:          **Continue for**
12:      **end if**
13:      **if** $\alpha \sim \text{Uniform}[0, 1]$, $\alpha < \frac{w'}{\hat{V}(s_{1:T}^{\text{acc}})} \cdot \frac{\hat{V}(s_{1:T}^{[h]})}{w}$ **then**
14:          Set $s_{1:T}^{\text{acc}} \leftarrow s_{1:T}^{[h]}$
15:      **end if**
16: **end for**
17: **Output:** $s_{1:T}^{\text{acc}}$

---

## C. Proofs in Section 4

*Proof of Theorem 4.1.* Write $T = 3m$. We assume $L$ is integer, and can be easily generalized to any real number greater equal to 1. Fix an arbitrary randomized algorithm $A$, and view it as $R \in \Delta(\mathcal{A})$ over deterministic algorithms a. Measure time by the number of visited depth-$2m$ prefixes. Assume that every a in the support of $R$ visits at most $Q$ depth-$2m$ prefixes on every run.

**Hard input family.** Set $B := L$ and let $\pi_{\text{unif}}$ be the uniform reference on leaves $[B]^T$. For each $u \in [B]^{2m}$ define $\hat{V}_u$ by $\hat{V}_u(\varnothing) = 1$ and

$$\frac{\hat{V}_u(s_{1:t})}{\hat{V}_u(s_{1:t-1})} = \begin{cases} 1, & 1 \leq t \leq 2m, \\ L, & 2m < t \leq 3m \text{ and } s_{1:2m} = u, \\ 1/L, & 2m < t \leq 3m \text{ and } s_{1:2m} \neq u. \end{cases} \tag{6}$$

Let $I_u := \langle B, \pi_{\text{unif}}, \hat{V}_u \rangle$ and denote by $p_{I_u}$ the induced target distribution on leaves.

For each $u \in [B]^{2m}$ define

$$A_u := \{s_{1:T} \in [B]^T : s_{1:2m} = u\}.$$

Under (6), every leaf in $A_u$ has unnormalized weight proportional to $L^m$, while every leaf in $A_u^c$ has unnormalized weight proportional to $L^{-m}$; hence $A_u$ is exactly the set of *maximal-probability leaves* under $p_{I_u}$. A direct computation gives

$$p_{I_u}(A_u) = \frac{1}{2 - L^{-2m}} > \frac{1}{2}. \tag{7}$$

Let $\hat{U} := \hat{S}_{1:2m}$. Then $\{\hat{U} = u\}$ is exactly the event $\{\hat{S}_{1:T} \in A_u\}$.

**Step 1 (TV control forces constant probability of sampling maximal leaves).** By assumption, $\|q_{A,I_u} - p_{I_u}\|_{\text{TV}} \leq 1/3$ for every $u \in [B]^{2m}$. Applying the definition of total variation to the event $A_u$ and using (7), we obtain

$$q_{A,I_u}(A_u) \geq p_{I_u}(A_u) - \frac{1}{3} > \frac{1}{6}. \tag{8}$$

Equivalently, for every $u \in [B]^{2m}$,

$$\mathbb{E}_{\mathsf{a} \sim R} \, \mathbb{P}_{o \sim \mathbb{P}_{\pi_{\text{unif}}}} \left( \hat{U} = u \mid \mathsf{a}, I_u \right) > \frac{1}{6}. \tag{9}$$

**Step 2 (Yao / inequality form via an explicit union bound).** Define an input distribution $\mathcal{D}$ as follows: $B = L$ with probability 1, $\pi_{\text{ref}} = \pi_{\text{unif}}$ with probability 1, and $\hat{V} = \hat{V}_U$ where $U \sim \text{Unif}([B]^{2m})$. Then $\mathcal{D} \in \Delta(\mathcal{I})$, where $\mathcal{I} := \{I_u : u \in [B]^{2m}\}$.

Fix any deterministic $\mathsf{a} \in \mathcal{A}$ that visits at most $Q$ depth-$2m$ prefixes. Let $\mathcal{U}_o(\mathsf{a})[1], \ldots, \mathcal{U}_o(\mathsf{a})[Q] \in [B]^{2m}$ denote the (oracle-dependent) list of visited depth-$2m$ prefixes when running $\mathsf{a}$ with oracle $o$. By the no-guess constraint, the event $\{\hat{U} = U\}$ implies that the visited list must contain $U$:

$$\{\hat{U} = U\} \subseteq \bigcup_{i=1}^{Q} \{U = \mathcal{U}_o(\mathsf{a})[i]\}.$$

Therefore, using independence of $U$ and $o$ under $\mathcal{D}$,

$$\begin{aligned}
\mathbb{E}_{U \sim \text{Unif}} \left[ \mathbb{P}_{o \sim \mathbb{P}_{\pi_{\text{unif}}}} \left( \hat{U} = U \mid \mathsf{a}, I_U \right) \right] &\leq \mathbb{E}_U \left[ \mathbb{P}_o \left( \bigcup_{i=1}^{Q} \{U = \mathcal{U}_o(\mathsf{a})[i]\} \right) \right] \\
&\leq \mathbb{E}_U \left[ \sum_{i=1}^{Q} \mathbb{P}_o \left( U = \mathcal{U}_o(\mathsf{a})[i] \right) \right] \\
&= \sum_{i=1}^{Q} \mathbb{E}_o \left[ \mathbb{P}_U \left( U = \mathcal{U}_o(\mathsf{a})[i] \mid o \right) \right] \\
&= \sum_{i=1}^{Q} \mathbb{E}_o \left[ \frac{1}{B^{2m}} \right] = \frac{Q}{B^{2m}}.
\end{aligned} \tag{10}$$

Now average over $a \sim R$ and use linearity:

$$\mathbb{E}_{I \sim \mathcal{D}} \, \mathbb{E}_{a \sim R} \, \mathbb{P}_{o \sim \mathbb{P}_{\pi_{\text{unif}}}} \big( \widehat{U} = U \mid a, I \big) \leq \frac{Q}{B^{2m}}. \tag{11}$$

Finally, since $\inf_{I \in \mathcal{I}} f(I) \leq \mathbb{E}_{I \sim \mathcal{D}}[f(I)]$ for any measurable $f$, we obtain the inequality-form Yao step

$$\inf_{I \in \mathcal{I}} \, \mathbb{E}_{a \sim R} \, \mathbb{P}_{o \sim \mathbb{P}_{\pi_{\text{unif}}}} \big( \widehat{U} = u(I) \mid a, I \big) \leq \frac{Q}{B^{2m}}, \tag{12}$$

where $u(I)$ denotes the unique index $u$ with $I = I_u$.

**Step 3 (Combine).** From (9) and the definition $\mathcal{I} := \{I_u : u \in [B]^{2m}\}$, we have

$$\inf_{I \in \mathcal{I}} \, \mathbb{E}_{a \sim R} \, \mathbb{P}_{o \sim \mathbb{P}_{\pi_{\text{unif}}}} \big( \widehat{U} = u(I) \mid a, I \big) > \frac{1}{6}. \tag{13}$$

Combining (12) and (13) yields $Q/B^{2m} \geq 1/6$, hence $Q \geq B^{2m}/6$. Recalling $B = L$ and $2m = 2T/3$, we conclude that the worst-case time complexity is $\Omega(L^{2T/3})$. $\qquad\square$

*Proof of Corollary 4.2.* We reduce to Theorem 4.1 by instantiating a hard family that satisfies Assumption 3.1 and 3.2.

Let $B := 1 + \varepsilon$ and let $\pi_{\text{unif}}$ be the uniform reference on $[B]^T$. For each $u \in [B]^{2m}$, define $\hat{V}_u$ by the same construction as (6) but with $L$ replaced everywhere by $1 + \varepsilon$:

$$\frac{\hat{V}_u(s_{1:t})}{\hat{V}_u(s_{1:t-1})} = \begin{cases} 1, & 1 \leq t \leq 2m, \\ 1 + \varepsilon, & 2m < t \leq 3m \text{ and } s_{1:2m} = u, \\ (1 + \varepsilon)^{-1}, & 2m < t \leq 3m \text{ and } s_{1:2m} \neq u. \end{cases} \tag{14}$$

(One checks as before that each $I_u = \langle B, \pi_{\text{unif}}, \hat{V}_u \rangle$ satisfies Assumption 3.1 and 3.2.)

Now apply the proof of Theorem 4.1 verbatim (with $L$ replaced by $1 + \varepsilon$ and $\mathbb{P}_{\pi_{\text{unif}}}$ as the oracle measure induced by $\pi_{\text{ref}} = \pi_{\text{unif}}$): the TV condition $\|q_{A, I_u} - p_{I_u}\|_{\text{TV}} \leq 1/3$ forces $q_{A, I_u}(A_u) > 1/6$, and the same union-bound/Yao argument yields a lower bound $Q \geq \frac{1}{6} B^{2m}$ on the worst-case number of visited depth-$2m$ prefixes. Since $2m = 2T/3$, this gives the stated $\Omega((1 + \varepsilon)^{2T/3})$ lower bound. $\qquad\square$

**Lemma C.1.** *Let $(\Omega, \mathcal{F}, \mu)$ be a measure space. Let $\pi, \phi, \psi : \Omega \to [0, \infty)$ be measurable with*

$$0 < Z_\phi := \int \pi(x)\phi(x) \, d\mu(x) < \infty, \qquad 0 < Z_\psi := \int \pi(x)\psi(x) \, d\mu(x) < \infty.$$

*Define probability measures $P, Q$ by*

$$P(dx) = \frac{\pi(x)\phi(x)}{Z_\phi} \, d\mu(x), \qquad Q(dx) = \frac{\pi(x)\psi(x)}{Z_\psi} \, d\mu(x).$$

*Assume that for some $\varepsilon \in (0, 1)$,*

$$\operatorname*{ess\,sup}_{x \in \Omega} \max \left\{ \frac{\phi(x)}{\psi(x)}, \frac{\psi(x)}{\phi(x)} \right\} \leq 1 + \varepsilon.$$

*Then*

$$\|P - Q\|_{\text{TV}} \leq \frac{\varepsilon}{1 - \varepsilon}.$$

*Proof of Lemma C.1.* Set

$$Z_\phi = \int \pi\phi \, d\mu, \qquad Z_\psi = \int \pi\psi \, d\mu.$$

On the one hand,

$$\frac{dQ}{d\mu}(x) = \frac{\pi(x)\psi(x)}{Z_\psi},$$

hence

$$\frac{dP}{dQ}(x) = \frac{\frac{dP}{d\mu}(x)}{\frac{dQ}{d\mu}(x)} = \frac{\pi(x)\phi(x)/Z_\phi}{\pi(x)\psi(x)/Z_\psi} = \frac{\phi(x)}{\psi(x)} \cdot \frac{Z_\psi}{Z_\phi}.$$

On the other hand,

$$Z_\phi = \int \pi(x)\phi(x)\, d\mu(x) = \int \frac{\phi(x)}{\psi(x)}\, \pi(x)\psi(x)\, d\mu(x) = Z_\psi\, \mathbb{E}_Q\left[\frac{\phi}{\psi}\right],$$

so

$$\frac{Z_\psi}{Z_\phi} = \frac{1}{\mathbb{E}_Q(\phi/\psi)}.$$

Thus

$$\frac{dP}{dQ}(x) = \frac{\phi(x)/\psi(x)}{\mathbb{E}_Q(\phi/\psi)}.$$

Define

$$a(x) := \frac{\phi(x)}{\psi(x)}.$$

From the assumption

$$\max\left\{\frac{\phi(x)}{\psi(x)}, \frac{\psi(x)}{\phi(x)}\right\} \le 1 + \varepsilon$$

we obtain

$$\frac{1}{1+\varepsilon} \le a(x) \le 1 + \varepsilon \quad \text{a.e.,}$$

and hence

$$|a(x) - 1| \le \varepsilon \quad \text{a.e.}$$

Therefore

$$|\,\mathbb{E}_Q(a) - 1\,| \le \mathbb{E}_Q|a - 1| \le \varepsilon, \qquad \mathbb{E}_Q(a) \ge 1 - \varepsilon > 0.$$

Now

$$\frac{dP}{dQ}(x) = \frac{a(x)}{\mathbb{E}_Q(a)},$$

so

$$\|P - Q\|_{\text{TV}} = \frac{1}{2}\int \left|\frac{dP}{dQ} - 1\right| dQ = \frac{1}{2}\,\mathbb{E}_Q\left|\frac{a}{\mathbb{E}_Q(a)} - 1\right| = \frac{1}{2}\,\mathbb{E}_Q\left|\frac{a - \mathbb{E}_Q(a)}{\mathbb{E}_Q(a)}\right|.$$

Using $\mathbb{E}_Q(a) \ge 1 - \varepsilon$ and the triangle inequality,

$$\mathbb{E}_Q\left|\frac{a - \mathbb{E}_Q(a)}{\mathbb{E}_Q(a)}\right| \le \frac{1}{1-\varepsilon}\,\mathbb{E}_Q|a - \mathbb{E}_Q(a)| \le \frac{1}{1-\varepsilon}\,\mathbb{E}_Q\big(|a - 1| + |\mathbb{E}_Q(a) - 1|\big) \le \frac{1}{1-\varepsilon}\,(\varepsilon + \varepsilon) = \frac{2\varepsilon}{1-\varepsilon}.$$

Hence

$$\|P - Q\|_{\text{TV}} \le \frac{1}{2} \cdot \frac{2\varepsilon}{1-\varepsilon} = \frac{\varepsilon}{1-\varepsilon},$$

which is the desired bound. $\qquad\square$

*Proof of Theorem 4.3.* For each $t \ge 0$, define the intermediate distribution on prefixes

$$q_t(s_{1:t} \mid s_0) := \frac{\pi_{\text{ref}}(s_{1:t} \mid s_0)\, \mathbb{E}_{s_{t+1} \sim \pi_{\text{ref}}(\cdot \mid s_0, s_{1:t})}\big[\hat{V}(s_0, s_{1:t+1})\big]}{\mathbb{E}_{s_{1:t+1} \sim \pi_{\text{ref}}(\cdot \mid s_0)}\big[\hat{V}(s_0, s_{1:t+1})\big]}. \tag{15}$$

By construction,

$$\tilde{\pi}_{t+1}(s_{1:t+1} \mid s_0) = q_t(s_{1:t} \mid s_0)\, \hat{\pi}(s_{t+1} \mid s_0, s_{1:t}). \tag{16}$$

Also, by the definition of the sequential sampler,

$$\hat{\pi}_{t+1}(s_{1:t+1} \mid s_0) = \hat{\pi}_t(s_{1:t} \mid s_0)\,\hat{\pi}(s_{t+1} \mid s_0, s_{1:t}). \tag{17}$$

Using (16)–(17) and the triangle inequality,

$$
\begin{aligned}
\left\|\tilde{\pi}_{t+1} - \hat{\pi}_{t+1}\right\|_{\mathrm{TV}} &= \left\|q_t\,\hat{\pi}(\cdot \mid s_{1:t}) - \hat{\pi}_t\,\hat{\pi}(\cdot \mid s_{1:t})\right\|_{\mathrm{TV}} \\
&\leq \left\|\tilde{\pi}_t\,\hat{\pi}(\cdot \mid s_{1:t}) - \hat{\pi}_t\,\hat{\pi}(\cdot \mid s_{1:t})\right\|_{\mathrm{TV}} + \left\|q_t\,\hat{\pi}(\cdot \mid s_{1:t}) - \tilde{\pi}_t\,\hat{\pi}(\cdot \mid s_{1:t})\right\|_{\mathrm{TV}}.
\end{aligned}
$$

For the two terms, since applying the same Markov kernel cannot increase total variation (data processing inequality),

$$\left\|\tilde{\pi}_t\,\hat{\pi}(\cdot \mid s_{1:t}) - \hat{\pi}_t\,\hat{\pi}(\cdot \mid s_{1:t})\right\|_{\mathrm{TV}} \leq \left\|\tilde{\pi}_t - \hat{\pi}_t\right\|_{\mathrm{TV}}, \qquad \left\|q_t\,\hat{\pi}(\cdot \mid s_{1:t}) - \tilde{\pi}_t\,\hat{\pi}(\cdot \mid s_{1:t})\right\|_{\mathrm{TV}} \leq \left\|q_t - \tilde{\pi}_t\right\|_{\mathrm{TV}}.$$

By Lemma C.1 together with Assumption 3.2, we have $\|q_t - \tilde{\pi}_t\|_{\mathrm{TV}} \leq 2\varepsilon$. Hence,

$$\|\tilde{\pi}_{t+1} - \hat{\pi}_{t+1}\|_{\mathrm{TV}} \leq \|\tilde{\pi}_t - \hat{\pi}_t\|_{\mathrm{TV}} + 2\varepsilon.$$

With the base case $\|\tilde{\pi}_0 - \hat{\pi}_0\|_{\mathrm{TV}} = 0$, telescoping over $t$ yields $\|\tilde{\pi}_t - \hat{\pi}_t\|_{\mathrm{TV}} \leq 2t\varepsilon$. $\qquad\square$

# D. Optimal proposal SMC

**Selection of proposal $\pi_{\mathrm{p}}$: Naive v.s. Optimal.** As mentioned in (Zhao et al., 2024), the different choice of proposal $\pi_{\mathrm{p}}$ from which the samples are drawn may lead to different variance in the estimation of normalizing factor. We claim that this is the exact same case for sampling. The simplest and most straightforward choice is the naive proposal, which is when one set $\pi_{\mathrm{p}} = \pi_{\mathrm{ref}}$, i.e. we sample from the output distribution of the pre-trained model itself. Under this circumstance, $G_t(s_{1:t}) = \frac{\hat{V}(s_0, s_{1:t})}{\hat{V}(s_0, s_{1:t-1})}$. Another choice, namely the optimal proposal that minimizes the variance of empirical normalizing factor (Zhao et al., 2024) is $\pi_{\mathrm{p}}(s_{t+1}|s_0, s_{1:t}) \propto \pi_{\mathrm{ref}}(s_{t+1}|s_0, s_{1:t}) \hat{V}(s_0, s_{1:t+1})$, in which case $G_t(s_{1:t}) = \frac{\mathbb{E}_{s_t \sim \pi_{\mathrm{ref}}(\cdot|s_0, s_{1:t-1})}[\hat{V}(s_0, s_{1:t})]}{\hat{V}(s_0, s_{1:t-1})}$. However, it's hard to get the exact optimal proposal, thus we use rejection sampling to approximate, which is shown in Algorithm 3. In Appendix B in specific, we instantiate Algorithm 1 for the *optimal* proposals in Algorithms 6.

**Particle complexity.** Specifically for the optimal proposal, the potential function (incremental weight) $G_t(s_{1:t}) = \frac{\mathbb{E}_{s_t \sim \pi_{\mathrm{ref}}(\cdot|s_0, s_{1:t-1})}[\hat{V}(s_0, s_{1:t})]}{\hat{V}(s_0, s_{1:t-1})}$ contains a conditional expectation over the next reward. In our setting, we consider using a practical monte carlo method to estimate this value, which requires an extra $N_m$ factor in the computation complexity. That is to say, every time we need to access $\mathbb{E}_{s_t \sim \pi_{\mathrm{ref}}(\cdot|s_0, s_{1:t-1})}[\hat{V}(s_0, s_{1:t})]$, we use $\frac{1}{N_m} \sum_{i=1}^{N_m} \hat{V}(s_0, (s_{1:t-1}, S_t))$ as an estimate, where $\{S_i\}_{i=1}^{N_m}$ are i.i.d. samples drawn independently from any other sources of randomness. Detailed algorithms for SMC with the optimal proposal is in Algorithm 6. Correspondingly, the theorem characterizing the particle complexity of optimal proposal SMC is as followed.

**Theorem D.1** (Particles Complexity). *Fix the horizon $T \geq 2$ and a target accuracy $\delta_{\mathrm{TV}} \in (0, 1)$. If Assumptions 3.1 and 3.2 hold with $L, \varepsilon > 0$, let $\bar{\eta}_{T+1}^N := \mathbb{E}[\eta_{T+1}^N]$ be the output distribution of an SMC sampler (after the terminal resampling step $T+1$ with $M_{T+1} = \mathrm{Id}$). Then the following sufficient conditions guarantee $\|\bar{\eta}_{T+1}^N - \eta_{T+1}\|_{\mathrm{TV}} \leq \delta_{\mathrm{TV}}$ under the optimal proposal. That is, the number of particles $N \geq \frac{((1+\varepsilon)^4 - 1) T^2 (1+\varepsilon)^{6T}}{2 \delta_{\mathrm{TV}}}$.*

The detained proof can be found in Appendix E.

*Remark* D.2. Theorem D.1 showcases a significant advantage of the optimal proposal, that it's particle complexity is irrelevant to $L$, potentially significantly smaller than that of naive proposal. Moreover, the particle complexity decreases as the bellman error $\varepsilon$ decreases, and specifically when the bellman error $\varepsilon = 0$, the sampling is exact with only one single particle—-that is—-the SMC sampler with optimal proposal degenerates to an action-level sampler (SP-gSMC). However, the following Proposition D.4 indicates that this efficiency in particle complexity comes at a cost—-namely that the estimation of incremental weight is difficult (especially in the sentence-level setting). Therefore, as we should mention later in Remark E.8, there's a trade-off between different sources of overhead to be taken into consideration when designing the proposal.

*Remark* D.3. Comparing Theorem 5.1 with Theorem 4.3, it can be noticed that for the optimal proposal specifically, under the $\varepsilon = O(\delta_{\mathrm{TV}}/T)$ regime, the particle complexity for a SMC sampler is $O\left(\frac{\varepsilon T^2}{\delta_{\mathrm{TV}}}\right) = O(T)$, while the simplest Single-particle guided SMC sampler only uses 1 particle. Inspecting the proof one can find that the extra $T$ complexity comes from the local variance of using $N$ particles amplified by the stability coefficient of the Feynman-Kac semi-group, which is proportion to $T$. On the other hand, despite that Single-particle guided SMC sampler is essentially a SMC sampler for the target Feynman-Kac flow with much variance (comparing to multi-particles), since it only has one particle, there's no actual reweighting step, so it has the flexibility to correspond to any another Feynman-Kac flows that share the same Markov transition kernel $M_t$, with differences only in $G_t$ and subsequently the local variance. Among all choices of $G_t$, a constant potential e.g. $G_t = 1$ gives the Single-particle guided SMC sampler 0 variance as a SMC sampler, trading off with a $O(\varepsilon T)$ bias. Therefore, all these observations indicate that with $o(T)$ particles, an optimal SMC sampler may not outperform the sampler using a single particle.

**Time complexity.** From Theorem 5.1 to Corollary 5.2 is straightforward, i.e. by timing a factor $T$ representing the horizon. However, due to the error induced by estimating the conditionally expected next reward, the complexity for SMC with the optimal proposal is not directly $T$ times the particle complexity, and need a more careful discussion in the following Proposition D.4.

**Proposition D.4** (Time complexity for a SMC sampler with the optimal proposal). *Under Assumption 3.1 and 3.2, and we consider the regime in which $\varepsilon = O(1/T)$ and $\delta_{\mathrm{TV}} = O(1)$. For any $0 < \delta \lesssim \delta_{\mathrm{TV}}$, using time $\tilde{O}\left(\frac{\varepsilon L T^5 \log(\delta^{-1})}{\delta_{\mathrm{TV}}^3}\right)$ there is*

*a good event $\mathcal{E}$ such that $\mathbb{P}(\mathcal{E}) \geq 1 - \delta$, conditioning on which a SMC sampler with the optimal proposal achieves $\delta_{\mathrm{TV}}$ accuracy, i.e. $\|\tilde{\pi}_T - \hat{\pi}_T|\mathcal{E}\|_{\mathrm{TV}} \leq \delta_{\mathrm{TV}}$.*

We postpone the proof to Appendix E.

*Remark* D.5 (On the inefficiency of SMC sampler with optimal proposal). As mentioned in Remark D.2, the disadvantage of an SMC sampler with optimal proposal is that it takes too much cost to evaluate the incremental weight—-requiring a Monte Carlo estimation of an empirical expectation to $O(\delta_{\mathrm{TV}}/T)$ accuracy. However, in a token level setting, things may be different, in the sense that the expectation is calculable within $O(|\mathcal{S}|)$ amount of time. Calculation of the time complexity within the token-level setting is a rather simple and straightforward corollary of Proposition D.1, but is outside the scope of this paper.

# E. Proofs in Section 5 and Appendix D

**Notation for particles.** In the proof, we adopt the notation used by (del Moral & Rio, 2011), where $\xi_t^{(N,i)}$ represents the $i$-th particle within the pool of $N$ particles at time step $t$. The evolution of the particles are exactly the same as in Section 2.2.

**Dobrushin (ergodic) coefficient and tensor product.** Let $M : (E, \mathcal{E}) \to (F, \mathcal{F})$ be a bounded integral operator with *constant mass*, i.e., $M(1)(x) = M(1)(y)$ for all $x, y \in E$. Following (del Moral & Rio, 2011), define the *Dobrushin coefficient*

$$\beta(M) := \sup_{f \in \mathrm{Osc}_1(F)} \mathrm{osc}(Mf), \qquad \mathrm{osc}(f) := \sup f - \inf f.$$

In particular, for a Markov kernel $P$ (so $P(1) = 1$), $\beta(P)$ is also called the *Dobrushin ergodic coefficient*.

For finite signed measures $\mu, \nu$ on $(E, \mathcal{E})$, define the tensor product measure $\mu \otimes \nu$ on $(E \times E, \mathcal{E} \otimes \mathcal{E})$ by

$$(\mu \otimes \nu)(h) := \iint h(x, y)\, \mu(dx)\, \nu(dy),$$

for any bounded measurable $h$ on $E \times E$. For measurable functions $g, h$ on $E$, we write $(g \otimes h)(x, y) := g(x)h(y)$.

**Lemma E.1** (FK first-order expansion, remainder, and stability (Appendix A.3 of (del Moral & Rio, 2011))). *Consider a Feynman–Kac model with Markov kernels $(M_k)_{k \geq 1}$ and potentials $(G_k)_{k \geq 0}$. Define*

$$Q_k(f)(x) := G_{k-1}(x)\, M_k(f)(x), \qquad Q_{p,n} := Q_{p+1} \cdots Q_n, \qquad Q_{n,n} := \mathrm{Id},$$

*and the normalized FK flow*

$$\Phi_{p,n}(\mu)(f) := \frac{\mu(Q_{p,n}(f))}{\mu(Q_{p,n}(1))}.$$

*Let*

$$P_{p,n}(f)(x) := \frac{Q_{p,n}(f)(x)}{Q_{p,n}(1)(x)}, \qquad G_{p,n,\eta}(x) := \frac{Q_{p,n}(1)(x)}{\eta(Q_{p,n}(1))}, \qquad q_{p,n} := \sup_{x,y \in E_p} \frac{Q_{p,n}(1)(x)}{Q_{p,n}(1)(y)}.$$

*(i) First-order (linear) operator. For any $\eta \in \mathcal{P}(E_p)$, define the linear operator $D_\eta \Phi_{p,n}$ by*

$$D_\eta \Phi_{p,n}(f)(x) := G_{p,n,\eta}(x)\, P_{p,n}\big(f - \Phi_{p,n}(\eta)(f)\big)(x). \tag{18}$$

*It is the first-order (Gateaux/Fréchet) linearization of the map $\mu \mapsto \Phi_{p,n}(\mu)$ at $\eta$: for any finite signed measure $\nu$ with $\nu(1) = 0$,*

$$\nu\big(D_\eta \Phi_{p,n}(f)\big) = \frac{d}{d\epsilon} \Phi_{p,n}(\eta + \epsilon\nu)(f)\Big|_{\epsilon=0}.$$

*(ii) First-order decomposition + explicit second-order remainder. For any $\mu, \eta \in \mathcal{P}(E_p)$ and bounded measurable $f$ on $E_n$,*

$$\Phi_{p,n}(\mu)(f) - \Phi_{p,n}(\eta)(f) = (\mu - \eta)\big(D_\eta \Phi_{p,n}(f)\big) + R^{\Phi_{p,n}}(\mu, \eta)(f), \tag{19}$$

*where the remainder admits the explicit tensor form*

$$R^{\Phi_{p,n}}(\mu, \eta)(f) := -\frac{1}{\mu(G_{p,n,\eta})} (\mu - \eta)^{\otimes 2}\big(G_{p,n,\eta} \otimes D_\eta \Phi_{p,n}(f)\big). \tag{20}$$

*(iii) Dobrushin stability. For any $f \in \mathrm{Osc}_1(E_n)$,*

$$\beta(D\Phi_{p,n}) := \sup_{\eta \in \mathcal{P}(E_p)} \beta(D_\eta \Phi_{p,n}) \leq 2\, q_{p,n}\, \beta(P_{p,n}) \leq 2\, q_{p,n}. \tag{21}$$

*Proof.* Fix $p \leq n$, $\mu, \eta \in \mathcal{P}(E_p)$ and a bounded measurable $f$ on $E_n$. Write for brevity

$$g := Q_{p,n}(1), \qquad h := Q_{p,n}(f), \qquad d := \eta(g), \qquad G := G_{p,n,\eta} = \frac{g}{d}, \qquad P := P_{p,n} = \frac{Q_{p,n}(\cdot)}{Q_{p,n}(1)} = \frac{\cdot}{g}.$$

Note that $\eta(G) = 1$ and $\mu(G) = \mu(g)/d > 0$. Also set $\phi := \Phi_{p,n}(\eta)(f) = \eta(h)/d = \eta(G\, P(f))$.

**Step 1: exact quotient identity.** By definition,

$$\Phi_{p,n}(\mu)(f) - \Phi_{p,n}(\eta)(f) = \frac{\mu(h)}{\mu(g)} - \frac{\eta(h)}{\eta(g)} = \frac{\mu(h)d - \eta(h)\mu(g)}{\mu(g)d}.$$

Using $h = g\,P(f)$ and $\eta(h) = d\,\phi$, we obtain

$$\mu(h)d - \eta(h)\mu(g) = d\,\mu(gP(f)) - \mu(g)\,d\,\phi = d\,\mu\big(g(P(f) - \phi)\big) = d^2\,\mu\big(G\,P(f - \phi)\big).$$

Similarly,

$$\eta(h)d - \eta(h)\eta(g) = d\,\eta(gP(f)) - d^2\phi = 0,$$

so $d^2\,\mu(GP(f - \phi)) = d^2(\mu - \eta)(GP(f - \phi))$. Therefore

$$\Phi_{p,n}(\mu)(f) - \Phi_{p,n}(\eta)(f) = \frac{d^2}{\mu(g)d}\,(\mu - \eta)\big(G\,P(f - \phi)\big) = \frac{1}{\mu(G)}\,(\mu - \eta)\big(D_\eta\Phi_{p,n}(f)\big),$$

where we used the definition $D_\eta\Phi_{p,n}(f) = G\,P(f - \Phi_{p,n}(\eta)(f)) = G\,P(f - \phi)$. This proves the (exact) identity

$$\Phi_{p,n}(\mu)(f) - \Phi_{p,n}(\eta)(f) = \frac{1}{\mu(G)}\,(\mu - \eta)\big(D_\eta\Phi_{p,n}(f)\big). \tag{22}$$

**Step 2: first-order decomposition + explicit remainder.** Since $\eta(G) = 1$, we have

$$\frac{1}{\mu(G)} - 1 = \frac{1 - \mu(G)}{\mu(G)} = -\frac{(\mu - \eta)(G)}{\mu(G)}.$$

Plugging this into (22) yields

$$\Phi_{p,n}(\mu)(f) - \Phi_{p,n}(\eta)(f) = (\mu - \eta)\big(D_\eta\Phi_{p,n}(f)\big) - \frac{(\mu - \eta)(G)}{\mu(G)}\,(\mu - \eta)\big(D_\eta\Phi_{p,n}(f)\big).$$

By the tensor-product convention $(\mu - \eta)^{\otimes 2}(G \otimes u) = (\mu - \eta)(G)\,(\mu - \eta)(u)$, the last term is exactly

$$-\frac{1}{\mu(G)}(\mu - \eta)^{\otimes 2}\Big(G \otimes D_\eta\Phi_{p,n}(f)\Big),$$

which is (20). This proves (19)–(20).

**Step 3: Dobrushin stability.** Let $f \in \mathrm{Osc}_1(E_n)$ and keep $\phi = \Phi_{p,n}(\eta)(f)$. Since $P$ is a Markov operator, $P(f)$ takes values in the convex hull of $f$, hence $\mathrm{osc}(P(f)) \leq \beta(P)\,\mathrm{osc}(f) \leq \beta(P)$ and $\|P(f) - c\|_\infty \leq \mathrm{osc}(P(f))$ for any constant $c$. Using $\phi = \eta(GP(f))$, we can write for any $x \in E_p$,

$$D_\eta\Phi_{p,n}(f)(x) = G(x)\Big(P(f)(x) - \eta(GP(f))\Big).$$

Therefore,

$$\big\|D_\eta\Phi_{p,n}(f)\big\|_\infty \leq \|G\|_\infty\,\big\|P(f) - \eta(GP(f))\big\|_\infty \leq \|G\|_\infty\,\mathrm{osc}(P(f)) \leq q_{p,n}\,\beta(P),$$

since $\|G\|_\infty = \sup_x g(x)/\eta(g) \leq \sup_x g(x)/\inf_y g(y) = q_{p,n}$.

Next, for any $x, x' \in E_p$,

$$\big|D_\eta\Phi_{p,n}(f)(x) - D_\eta\Phi_{p,n}(f)(x')\big|$$
$$\leq |G(x) - G(x')|\,\big|P(f)(x) - \eta(GP(f))\big| + \|G\|_\infty\,|P(f)(x) - P(f)(x')|.$$

Taking sup over $x, x'$ and using $\|P(f) - \eta(GP(f))\|_\infty \leq \mathrm{osc}(P(f)) \leq \beta(P)$, $\mathrm{osc}(P(f)) \leq \beta(P)$, $\|G\|_\infty \leq q_{p,n}$, and

$$\mathrm{osc}(G) = \sup_{x,x'}\Big|\frac{g(x)}{\eta(g)} - \frac{g(x')}{\eta(g)}\Big| = \frac{\sup g - \inf g}{\eta(g)} \leq \frac{\sup g - \inf g}{\inf g} \leq q_{p,n},$$

we get

$$\mathrm{osc}\big(D_\eta\Phi_{p,n}(f)\big) \leq \mathrm{osc}(G)\,\beta(P) + \|G\|_\infty\,\beta(P) \leq 2\,q_{p,n}\,\beta(P).$$

Since this holds for all $f \in \mathrm{Osc}_1(E_n)$, it follows that $\beta(D_\eta\Phi_{p,n}) \leq 2q_{p,n}\beta(P)$, and taking the supremum over $\eta$ gives (21). Finally, as $P$ is a Markov kernel, $\beta(P) \leq 1$, so $\beta(D\Phi_{p,n}) \leq 2q_{p,n}$. $\square$

**Lemma E.2** (Control of FK ratio and first-order stability). *Under Assumption 3.1 and Assumption 3.2, consider:*

*(a) Naive proposal.* $M_t(s_{1:t-1}, ds_{1:t}) = \pi_{\mathrm{ref}}(s_t|s_0, s_{1:t-1})$, $G_t(s_{1:t}) = \dfrac{\hat{V}(s_0, s_{1:t})}{\hat{V}(s_0, s_{1:t-1})}$.

*(b) Optimal proposal.* $M_t(s_{1:t-1}, ds_{1:t}) = \dfrac{\pi_{\mathrm{ref}}(s_t|s_0, s_{1:t-1})\hat{V}(s_0, s_{1:t})}{\mathbb{E}_{s_t \sim \pi_{ref}(\cdot|s_0, s_{1:t-1})}[\hat{V}(s_0, s_{1:t})]}$, $G_t(s_{1:t}) = \dfrac{\mathbb{E}_{s_t \sim \pi_{ref}(\cdot|s_0, s_{1:t-1})}[\hat{V}(s_0, s_{1:t})]}{\hat{V}(s_0, s_{1:t-1})}$.

*Then for any $n \geq m$,*

1. *In case (a): $q_{m,n} \leq L^2(1+\varepsilon)^{2(n-m-1)}$ for $n > m$, and $q_{m,m} = 1$.*

2. *In case (b): $q_{m,n} \leq (1+\varepsilon)^{2(n-m)}$ for $n > m$, and $q_{m,m} = 1$.*

3. *In both cases: $\beta(D\Phi_{m,n}) \leq 2\,q_{m,n}$.*

*Proof.* For $q_{m,n}$, we prove the naive proposal case, and the optimal proposal case follows from a similar argument.

When $m = n$, according to the definition $Q_{m,n} = \mathrm{Id}$, so $q_{m,n} = \sup_{x,y} \dfrac{Q_{m,n}(1)(x)}{Q_{m,n}(1)(y)} = 1$.

**Intuition for $Q_{m,n}(1)(x)$.** The quantity $Q_{m,n}(1)(x)$ can be interpreted as the *lookahead-to-go* normalizing factor starting from the current prefix/state $x \in E_m$: it is the conditional expectation (under the proposal dynamics from time $m$ to $n-1$) of the product of future incremental potentials,

$$Q_{m,n}(1)(x) = \mathbb{E}\left[\prod_{t=m}^{n-1} G_t(S_{1:t}) \,\Big|\, S_{1:m} = x\right].$$

Its variability across different $x$ can be understood via two effects. First, the *future* contribution $\mathbb{E}[\prod_{t=m+1}^{n-1} G_t \mid S_{1:m} = x]$ is "smoothed" by the conditional expectation; under a small Bellman error assumption, the corresponding one-step lookahead ratios are uniformly close to 1, so this future term is close (almost everywhere) to a constant and hence becomes nearly independent of $x$. Second, the *current* incremental potential $G_m(x)$ is $\sigma(S_{1:m})$-measurable and therefore not smoothed—its fluctuation depends strongly on the proposal choice (typically larger under the naive proposal where $G_m = \hat{V}(s_0, s_{1:m})/\hat{V}(s_0, s_{1:m-1})$, and smaller under the optimal proposal where $G_m$ is a conditional-expectation ratio).

When $n > m$, we use induction to show that $L^{-1} \cdot (1+\varepsilon)^{-(n-m-1)} \leq Q_{m,n}(1)(x) \leq L \cdot (1+\varepsilon)^{n-m-1}$. Recall that for all $x \in \mathcal{S}_{1:m}$,

$$Q_{m,n}(1)(x) = \mathbb{E}\left[\prod_{p=m}^{n-1} G_p(X_p)\Big| X_m = x\right].$$

We first prove that $Q_{m,m+1}(1)(x) \in [L^{-1}, L]$, which is obvious. Then assume $Q_{m,n}(1)(x) \in [L^{-1} \cdot (1+\varepsilon)^{-(n-m-1)}, L \cdot (1+\varepsilon)^{n-m-1}]$, we need to prove that $Q_{m,n+1}(1)(x) \in [L^{-1} \cdot (1+\varepsilon)^{-(n-m)}, L \cdot (1+\varepsilon)^{n-m}]$. This is because

$$Q_{m,n+1}(1)(x) = \mathbb{E}\left[\prod_{p=m}^{n} G_p(X_p)\Big| X_m = x\right] = \mathbb{E}\left[\mathbb{E}[G_n(X_n)|X_{n-1}]\prod_{p=m}^{n-1} G_p(X_p)\Big| X_m = x\right]$$

$$\leq \sup \mathbb{E}[G_n(X_n)|X_{n-1}] \cdot \mathbb{E}\left[\prod_{p=m}^{n-1} G_p(X_p)\Big| X_m = x\right]$$

$$= \sup\left\{\frac{\mathbb{E}_{s_t \sim \pi_{\mathrm{ref}}(\cdot|s_0, s_{1:t-1})}[\hat{V}(s_0, s_{1:t})]}{\hat{V}(s_0, s_{1:t-1})}\right\} \cdot Q_{m,n}(1)(x)$$

$$\leq (1+\varepsilon) \cdot L \cdot (1+\varepsilon)^{n-m-1} = L \cdot (1+\varepsilon)^{n-m}.$$

Similarly for the other direction,

$$Q_{m,n+1}(1)(x) \geq \inf \mathbb{E}[G_n(X_n)|X_{n-1}] \cdot \mathbb{E}\left[\prod_{p=m}^{n-1} G_p(X_p)\Big| X_m = x\right]$$

$$= \inf \left\{ \frac{\mathbb{E}_{s_t \sim \pi_{\mathrm{ref}}(\cdot|s_0,s_{1:t-1})}[\hat{V}(s_0,s_{1:t})]}{\hat{V}(s_0,s_{1:t-1})} \right\} \cdot Q_{m,n}(1)(x)$$

$$\geq (1+\varepsilon)^{-1} \cdot L^{-1} \cdot (1+\varepsilon)^{-(n-m-1)} = L^{-1} \cdot (1+\varepsilon)^{-(n-m)}.$$

and that finishes the induction. Then we conclude that $q_{m,n} = \sup_{x,y} \dfrac{Q_{m,n}(1)(x)}{Q_{m,n}(1)(y)} \leq L^2 \cdot (1+\varepsilon)^{2(n-m-1)}$ for $n > m$.

Finally, by Lemma E.1 and (21),

$$\beta(D\Phi_{m,n}) \leq 2\,q_{m,n}\,\beta(P_{m,n}) \leq 2\,q_{m,n},$$

since $P_{m,n}$ is a Markov kernel and hence $\beta(P_{m,n}) \leq 1$.

$\square$

**Lemma E.3** (Random telescoping identity). *For any $n \geq 1$ and any bounded measurable test function $f$ on $E_n$,*

$$[\eta_n^N - \eta_n](f) = \sum_{p=1}^{n} \left( \Phi_{p,n}(\eta_p^N) - \Phi_{p,n}(\Phi_p(\eta_{p-1}^N)) \right)(f). \tag{23}$$

*Consequently,*

$$[\bar{\eta}_n^N - \eta_n](f) = \sum_{p=1}^{n} \mathbb{E}\left[ \left( \Phi_{p,n}(\eta_p^N) - \Phi_{p,n}(\Phi_p(\eta_{p-1}^N)) \right)(f) \right], \qquad \bar{\eta}_n^N := \mathbb{E}[\eta_n^N]. \tag{24}$$

*Proof.* The identity (23) is purely algebraic: for each $p \geq 1$,

$$\Phi_{p-1,n} = \Phi_{p,n} \circ \Phi_p,$$

hence

$$\Phi_{p,n}(\Phi_p(\eta_{p-1}^N)) = \Phi_{p-1,n}(\eta_{p-1}^N).$$

Summing $\Phi_{p,n}(\eta_p^N) - \Phi_{p-1,n}(\eta_{p-1}^N)$ over $p = 1, \dots, n$ telescopes to $\eta_n^N(f) - \eta_n(f)$, since $\Phi_{n,n} = \mathrm{Id}$ and $\eta_0^N = \eta_0$. Taking expectations gives (24). $\square$

**Lemma E.4** (One-step conditional covariance under a McKean kernel). *Fix $p \geq 1$ and let $\mathcal{F}_{p-1}^N$ be the $\sigma$–field generated by the particle system up to time $p - 1$. Assume that, conditionally on $\mathcal{F}_{p-1}^N$, the particles are updated independently via the McKean kernel $K_{p,\eta_{p-1}^N}$, i.e.,*

$$\xi_p^{(N,i)} \mid \mathcal{F}_{p-1}^N \sim K_{p,\eta_{p-1}^N}(\xi_{p-1}^{(N,i)}, \cdot), \qquad i = 1, \dots, N,$$

*and are conditionally independent across $i$. Define the one-step predictive measure*

$$\nu_p^N := \Phi_p(\eta_{p-1}^N) = \eta_{p-1}^N K_{p,\eta_{p-1}^N} \in \mathcal{P}(E_p),$$

*where the second equality follows from the McKean consistency (1). Then for any bounded measurable $u, v$ on $E_p$,*

$$\mathbb{E}\left([\eta_p^N - \nu_p^N](u) \Big| \mathcal{F}_{p-1}^N\right) = 0, \tag{25}$$

$$\mathbb{E}\left([\eta_p^N - \nu_p^N](u)\,[\eta_p^N - \nu_p^N](v) \Big| \mathcal{F}_{p-1}^N\right) = \frac{1}{N}\,\eta_{p-1}^N\left(K_{p,\eta_{p-1}^N}\left((u - K_{p,\eta_{p-1}^N}u)\,(v - K_{p,\eta_{p-1}^N}v)\right)\right). \tag{26}$$

*In particular,*

$$\mathbb{E}\left([\eta_p^N - \nu_p^N](u)^2 \Big| \mathcal{F}_{p-1}^N\right) = \frac{1}{N}\,\eta_{p-1}^N\left(K_{p,\eta_{p-1}^N}\left((u - K_{p,\eta_{p-1}^N}u)^2\right)\right). \tag{27}$$

*Moreover,*

$$\left| \mathbb{E}\left([\eta_p^N - \nu_p^N](u)\,[\eta_p^N - \nu_p^N](v) \mid \mathcal{F}_{p-1}^N\right) \right| \leq \frac{1}{4N}\,\mathrm{osc}(u)\,\mathrm{osc}(v). \tag{28}$$

*Proof.* Conditionally on $\mathcal{F}_{p-1}^N$, we have

$$\eta_p^N(u) = \frac{1}{N}\sum_{i=1}^N u(\xi_p^{(N,i)}), \qquad \nu_p^N(u) = \eta_{p-1}^N K_{p,\eta_{p-1}^N}(u) = \frac{1}{N}\sum_{i=1}^N K_{p,\eta_{p-1}^N} u(\xi_{p-1}^{(N,i)}).$$

Hence

$$[\eta_p^N - \nu_p^N](u) = \frac{1}{N}\sum_{i=1}^N \Big( u(\xi_p^{(N,i)}) - K_{p,\eta_{p-1}^N} u(\xi_{p-1}^{(N,i)}) \Big).$$

Taking conditional expectation and using that $\mathbb{E}[u(\xi_p^{(N,i)}) \mid \mathcal{F}_{p-1}^N] = K_{p,\eta_{p-1}^N} u(\xi_{p-1}^{(N,i)})$ yields (25).

For the second identity, expand the product and use conditional independence across $i$: all cross terms vanish because each summand has conditional mean 0, giving

$$\mathbb{E}\Big( [\eta_p^N - \nu_p^N](u)\,[\eta_p^N - \nu_p^N](v) \,\Big|\, \mathcal{F}_{p-1}^N \Big)$$

$$= \frac{1}{N^2}\sum_{i=1}^N \mathbb{E}\Big[ \big(u(\xi_p^{(N,i)}) - K_{p,\eta_{p-1}^N} u(\xi_{p-1}^{(N,i)})\big)\big(v(\xi_p^{(N,i)}) - K_{p,\eta_{p-1}^N} v(\xi_{p-1}^{(N,i)})\big) \,\Big|\, \mathcal{F}_{p-1}^N \Big]$$

$$= \frac{1}{N^2}\sum_{i=1}^N K_{p,\eta_{p-1}^N}\big((u - K_{p,\eta_{p-1}^N}u)(v - K_{p,\eta_{p-1}^N}v)\big)(\xi_{p-1}^{(N,i)}),$$

which is (26). Taking $u = v$ gives (27).

Finally, for any probability measure $\kappa$,

$$\big|\mathrm{Cov}_\kappa(u,v)\big| \le \sqrt{\mathrm{Var}_\kappa(u)\,\mathrm{Var}_\kappa(v)} \le \frac{1}{4}\,\mathrm{osc}(u)\,\mathrm{osc}(v),$$

hence for each $x$,

$$\big|K_{p,\eta}((u - K_{p,\eta}u)(v - K_{p,\eta}v))(x)\big| \le \frac{1}{4}\,\mathrm{osc}(u)\,\mathrm{osc}(v).$$

Averaging over $\eta_{p-1}^N$ and multiplying by $1/N$ gives (28). $\qquad\square$

**Lemma E.5** (Local bias comes only from the second-order remainder)**.** *Under the same condition as Lemma E.4. Fix $n \ge 1$, $p \in \{1,\dots,n\}$, and $f \in \mathrm{Osc}_1(E_n)$. Let $\nu_p^N = \Phi_p(\eta_{p-1}^N)$ as above and write $\beta_{p,n} := \beta(D\Phi_{p,n})$. Then*

$$\Big|\mathbb{E}\Big[\big(\Phi_{p,n}(\eta_p^N) - \Phi_{p,n}(\nu_p^N)\big)(f)\Big]\Big| \le \frac{1}{4N}\,(q_{p,n}^2 - 1)\,\beta_{p,n}. \tag{29}$$

*In particular, if $q_{p,n} = 1$, then this local bias term is 0.*

*Proof.* Apply the first-order expansion (Lemma E.1) to $\Phi_{p,n}$ at $\eta = \nu_p^N$ and $\mu = \eta_p^N$:

$$\big(\Phi_{p,n}(\eta_p^N) - \Phi_{p,n}(\nu_p^N)\big)(f) = [\eta_p^N - \nu_p^N]\big(D_{\nu_p^N}\Phi_{p,n}(f)\big) + R^{\Phi_{p,n}}(\eta_p^N, \nu_p^N)(f).$$

Taking conditional expectation given $\mathcal{F}_{p-1}^N$ and using (25) kills the linear term, hence

$$\mathbb{E}\Big[\big(\Phi_{p,n}(\eta_p^N) - \Phi_{p,n}(\nu_p^N)\big)(f)\,\Big|\,\mathcal{F}_{p-1}^N\Big] = \mathbb{E}\big[R^{\Phi_{p,n}}(\eta_p^N, \nu_p^N)(f) \mid \mathcal{F}_{p-1}^N\big].$$

Now use the explicit Feynman–Kac remainder form (20):

$$R^{\Phi_{p,n}}(\eta_p^N, \nu_p^N)(f) = -\frac{1}{\eta_p^N(G_{p,n,\nu_p^N})}\,[\eta_p^N - \nu_p^N](G_{p,n,\nu_p^N})\,[\eta_p^N - \nu_p^N]\big(D_{p,n,\nu_p^N}(f)\big)$$

in which $D_{p,n,\eta}$ is a shorthand for $D_\eta \Phi_{p,n}$. By the ratio bound in Lemma E.1, $G_{p,n,\nu_p^N}(x) \in [q_{p,n}^{-1}, q_{p,n}]$, hence

$$\frac{1}{\eta_p^N(G_{p,n,\nu_p^N})} \le q_{p,n}, \qquad \mathrm{osc}(G_{p,n,\nu_p^N}) \le q_{p,n} - q_{p,n}^{-1}.$$

Therefore, conditionally on $\mathcal{F}_{p-1}^N$,

$$\left| \mathbb{E}\big[ R^{\Phi_{p,n}}(\eta_p^N, \nu_p^N)(f) \mid \mathcal{F}_{p-1}^N \big] \right| \le q_{p,n} \, \mathbb{E}\Big[ \big| [\eta_p^N - \nu_p^N](G_{p,n,\nu_p^N}) \, [\eta_p^N - \nu_p^N](D_{p,n,\nu_p^N}(f)) \big| \ \Big| \ \mathcal{F}_{p-1}^N \Big]$$

$$\le q_{p,n} \left( \mathbb{E}\big[ [\eta_p^N - \nu_p^N](G_{p,n,\nu_p^N})^2 \mid \mathcal{F}_{p-1}^N \big] \right)^{1/2} \left( \mathbb{E}\big[ [\eta_p^N - \nu_p^N](D_{p,n,\nu_p^N}(f))^2 \mid \mathcal{F}_{p-1}^N \big] \right)^{1/2}$$

$$\le q_{p,n} \, \frac{1}{4N} \operatorname{osc}(G_{p,n,\nu_p^N}) \operatorname{osc}(D_{p,n,\nu_p^N}(f)),$$

where the last step uses (28) with $u = v$. Using $q_{p,n}\operatorname{osc}(G_{p,n,\nu_p^N}) \le q_{p,n}(q_{p,n} - q_{p,n}^{-1}) = q_{p,n}^2 - 1$ yields

$$\left| \mathbb{E}\big[ R^{\Phi_{p,n}}(\eta_p^N, \nu_p^N)(f) \mid \mathcal{F}_{p-1}^N \big] \right| \le \frac{1}{4N} (q_{p,n}^2 - 1) \, \beta_{p,n}.$$

Taking expectations over $\mathcal{F}_{p-1}^N$ gives (29). $\qquad \square$

**Theorem E.6** (SMC sampler TV bias bound). *For a SMC sampler associated with a FK model, any $N \ge 1$ and any $n \ge 1$,*

$$\|\bar{\eta}_n^N - \eta_n\|_{\mathrm{TV}} \le \frac{1}{4N} \sum_{p=1}^n (q_{p,n}^2 - 1) \, \beta(D\Phi_{p,n}). \tag{30}$$

*In particular, if $q_{p,n} = 1$ for all $p \le n$ (e.g. perfectly flat potentials), then $\bar{\eta}_n^N = \eta_n$.*

*Proof of Theorem E.6.* Fix any $f \in \operatorname{Osc}_1(E_n)$. By the bias telescoping (24),

$$[\bar{\eta}_n^N - \eta_n](f) = \sum_{p=1}^n \mathbb{E}\Big[ \big( \Phi_{p,n}(\eta_p^N) - \Phi_{p,n}(\Phi_p(\eta_{p-1}^N)) \big)(f) \Big].$$

Since $\Phi_p(\eta_{p-1}^N) = \nu_p^N$, each summand is bounded by Lemma E.5, hence

$$|[\bar{\eta}_n^N - \eta_n](f)| \le \frac{1}{4N} \sum_{p=1}^n (q_{p,n}^2 - 1) \, \beta(D\Phi_{p,n}).$$

Taking the supremum over $f \in \operatorname{Osc}_1(E_n)$ yields (30). $\qquad \square$

*Proof of Theorem 5.1 and Theorem D.1.* Start from the local TV-bias bound (Theorem E.6):

$$\|\bar{\eta}_{T+1}^N - \eta_{T+1}\|_{\mathrm{TV}} \le \frac{1}{4N} \sum_{p=1}^{T+1} (q_{p,T+1}^2 - 1) \, \beta(D\Phi_{p,T+1}). \tag{31}$$

By Lemma E.1, $\beta(D\Phi_{p,T+1}) \le 2q_{p,T+1}\beta(P_{p,T+1}) \le 2q_{p,T+1}$, so (31) yields

$$\|\bar{\eta}_{T+1}^N - \eta_{T+1}\|_{\mathrm{TV}} \le \frac{1}{2N} \sum_{p=1}^{T+1} q_{p,T+1} \, (q_{p,T+1}^2 - 1). \tag{32}$$

Note the last term $p = T + 1$ vanishes since $q_{T+1,T+1} = 1$.

**(i) Naive proposal.** By Lemma E.2, for $p \le T$, $q_{p,T+1} \le L^2(1 + \varepsilon)^{2(T-p)}$. Using $q(q^2 - 1) \le q^3$ for $q \ge 1$,

$$\sum_{p=1}^T q_{p,T+1}(q_{p,T+1}^2 - 1) \le \sum_{p=1}^T q_{p,T+1}^3 \le L^6 \sum_{p=1}^T (1 + \varepsilon)^{6(T-p)} = L^6 \sum_{j=0}^{T-1} a^j, \quad a := (1 + \varepsilon)^6 \ge 1.$$

Using the elementary bound $\sum_{j=0}^{m-1} a^j \le m\, a^{m-1}$ for $a \ge 1$ (with $m = T$) gives

$$\sum_{p=1}^{T} q_{p,T+1}(q_{p,T+1}^2 - 1) \le L^6\, T\, (1+\varepsilon)^{6(T-1)}.$$

Plugging into (32) yields

$$\|\bar{\eta}_{T+1}^N - \eta_{T+1}\|_{\mathrm{TV}} \le \frac{L^6\, T\, (1+\varepsilon)^{6(T-1)}}{2N},$$

so $N \ge \frac{L^6\, T\, (1+\varepsilon)^{6(T-1)}}{2\, \delta_{\mathrm{TV}}}$ ensures $\|\bar{\eta}_{T+1}^N - \eta_{T+1}\|_{\mathrm{TV}} \le \delta_{\mathrm{TV}}$.

**(ii) Optimal proposal.** By Lemma E.2, for $p \le T$, $q_{p,T+1} \le (1+\varepsilon)^{2(T+1-p)}$. Let $k := T+1-p \in \{1, \ldots, T\}$. Then

$$q_{p,T+1}(q_{p,T+1}^2 - 1) \le (1+\varepsilon)^{2k}\big((1+\varepsilon)^{4k} - 1\big) \le k\big((1+\varepsilon)^4 - 1\big)(1+\varepsilon)^{6k},$$

where we used $b^k - 1 \le k(b-1)b^k$ for $b \ge 1$ with $b = (1+\varepsilon)^4$. Hence

$$\sum_{p=1}^{T} q_{p,T+1}(q_{p,T+1}^2 - 1) \le \big((1+\varepsilon)^4 - 1\big) \sum_{k=1}^{T} k\, a^k, \qquad a := (1+\varepsilon)^6 \ge 1.$$

Using the crude bound $\sum_{k=1}^{m} k\, a^k \le m^2 a^m$ for $a \ge 1$ (here $m = T$), we obtain

$$\sum_{p=1}^{T} q_{p,T+1}(q_{p,T+1}^2 - 1) \le \big((1+\varepsilon)^4 - 1\big) T^2\, (1+\varepsilon)^{6T}.$$

Plugging into (32) gives

$$\|\bar{\eta}_{T+1}^N - \eta_{T+1}\|_{\mathrm{TV}} \le \frac{\big((1+\varepsilon)^4 - 1\big) T^2\, (1+\varepsilon)^{6T}}{2N},$$

so $N \ge \frac{((1+\varepsilon)^4 - 1)\, T^2\, (1+\varepsilon)^{6T}}{2\, \delta_{\mathrm{TV}}}$ ensures $\|\bar{\eta}_{T+1}^N - \eta_{T+1}\|_{\mathrm{TV}} \le \delta_{\mathrm{TV}}$.   $\square$

**Proposition E.7** (Particles Complexity of arbitrary proposal). *Fix the horizon $T \ge 2$ and a target accuracy $\delta_{\mathrm{TV}} \in (0, 1)$. Under Assumptions 3.1 and 3.2, let $\bar{\eta}_{T+1}^N := \mathbb{E}[\eta_{T+1}^N]$ be the output distribution of an SMC sampler (after the terminal resampling step $T+1$ with $M_{T+1} = \mathrm{Id}$). We further suppose*

$$\max_{t \in [T]} \max_{s_{1:t} \in \mathcal{S}_{1:t}} \max \left\{ \frac{\pi_{\mathrm{ref}}(s_t \mid s_{1:t-1})}{\pi_{\mathrm{p}}(s_t \mid s_{1:t-1})} \cdot \frac{\hat{V}(s_0, s_{1:t})}{\hat{V}(s_0, s_{1:t-1})}, \frac{\pi_{\mathrm{p}}(s_t \mid s_{1:t-1})}{\pi_{\mathrm{ref}}(s_t \mid s_{1:t-1})} \cdot \frac{\hat{V}(s_0, s_{1:t-1})}{\hat{V}(s_0, s_{1:t})} \right\} \le L_{\mathrm{p}}.$$

*Then a sufficient condition that guarantees $\|\bar{\eta}_{T+1}^N - \eta_{T+1}\|_{\mathrm{TV}} \le \delta_{\mathrm{TV}}$ is*

$$N \ge \frac{L_{\mathrm{p}}^6 T(1+\varepsilon)^{6(T-1)}}{2\delta_{\mathrm{TV}}}.$$

*Proof.* The proof follows a similar pattern of Theorem 5.1.   $\square$

*Remark* E.8. Proposition E.7 indicates that choosing the proposal $\pi_{\mathrm{p}}$ allows one to balance the overhead induced by the particle complexity and executing the proposal itself. A well designed proposal may have moderate $L_{\mathrm{p}}$, yet may be hard to sample from or whose corresponding incremental weight $G_t = \frac{\pi_{\mathrm{ref}}(s_t \mid s_{1:t-1})}{\pi_{\mathrm{p}}(s_t \mid s_{1:t-1})} \cdot \frac{\hat{V}(s_0, s_{1:t})}{\hat{V}(s_0, s_{1:t-1})}$ may be difficult to calculate to the desired accuracy. The naive proposal (Corollary 5.2) and the optimal proposal (Proposition D.4) are two extremes in this sense. The naive proposal is extremely easy to sample from, and the incremental weight is trivially obtained using the reward model without further calculation, but the particle complexity may scale *in the worst case* as $O(L^6)$. On the other hand, the optimal proposal has potentially very low particle complexity, scaling as $O(\varepsilon T)$, and can even degenerate to requiring only one single particle when the bellman error is small enough. But the cost of estimating the incremental weight is relatively large. **Therefore, there is an important trade-off between the number of particles and the difficulty of proposal (including the calculation of incremental weight).**

We also note that we are considering sentence-level reasoning algorithm. In the token-level, one may access the explicit probability of next tokens, thus the trade-off may also be different. However the particle complexity is exactly the same as in Theorem 5.1 and Proposition E.7.

*Proof of Proposition D.4.* In this proof, whenever we write $\mathbb{E}[X|\mathcal{F}, E]$ for some random variable $X$, sigma algebra $\mathcal{F}$ and event $E$, we mean $\mathbb{E}[X|\sigma(\mathcal{F} \cup \{E\})]\mathbb{I}_E$, since we usually do not care about what happen outside of $E$. Then it's not difficult to see that, since

$$\mathbb{E}[X|\sigma(\mathcal{F} \cup \{E\})] = \mathbb{E}[X\mathbb{I}_E|\mathcal{F}]\mathbb{I}_E/\mathbb{P}(E|\mathcal{F}) + \mathbb{E}[X\mathbb{I}_{E^c}|\mathcal{F}]\mathbb{I}_{E^c}/\mathbb{P}(E^c|\mathcal{F}),$$

we have

$$\mathbb{E}[X|\mathcal{F}, E] = \mathbb{E}[X\mathbb{I}_E|\mathcal{F}]\mathbb{I}_E/\mathbb{P}(E|\mathcal{F}).$$

**Setup and filtrations.** Our FK horizon is $n := T+1$, where the terminal step $T+1$ uses $M_{T+1} = \mathrm{Id}$ and applies the last potential $G_T$ (no new token is generated). The Monte-Carlo normalizer estimation and rejection-sampling proposals only occur for token steps $t = 1, \ldots, T$.

Given $N$ particles, they in total run $T$ token-generation steps in the algorithm. At each such step, we estimate $N$ conditional expectations using Monte Carlo, each by averaging over $N_m$ samples. For each $t \in [T], i \in [N]$, the estimation is denoted by a random variable

$$\hat{Z}_{t,i} := \frac{1}{N_m} \sum_{m=1}^{N_m} \hat{V}\big(s_0, (S_{1:t-1}, \tilde{S}_t^{[m]})\big),$$

where each $\tilde{S}_t^{[m]}$ is conditionally independently drawn from $\pi_{\mathrm{ref}}$ given $\mathcal{F}_{t-1}^{N,i-1}$, which denotes the sigma algebra generated by all previous particles (until time $t-1$), including using rejection sampling for optimal proposal, and MC estimation random variables. The indexed family of sigma algebras form a natural filtration on the sample space $\Omega$:

$$(\mathcal{F}_1^{N,0}, \mathcal{F}_1^{N,1}, \mathcal{F}_1^{N,2}, \cdots, \mathcal{F}_1^{N,N}, \mathcal{F}_2^{N,0}, \cdots, \mathcal{F}_2^{N,N}, \cdots, \mathcal{F}_T^{N,0}, \cdots, \mathcal{F}_T^{N,N}).$$

Ideally, each $\hat{Z}_{t,i}$ is a good estimate of

$$Z_{t,i} := \mathbb{E}_{s_t \sim \pi_{\mathrm{ref}}(\cdot|s_0, S_{1:t-1})}[\hat{V}(s_0, (S_{1:t-1}, s_t))] = \mathbb{E}[\hat{Z}_{t,i}|\mathcal{F}_{t-1}^{N,i-1}].$$

**Good events for MC estimation.** Now we define the failure events for any $\delta > 0$, and

$$\xi := 2\sqrt{2}\sqrt{\frac{C_{\mathrm{act}} \log \frac{4}{\delta}}{N_m}}$$

as

$$\mathcal{B}_{t,i} := \left\{ \left| \frac{\hat{Z}_{t,i}}{Z_{t,i}} - 1 \right| > \xi \right\}.$$

Here $C_{\mathrm{act}}$ is defined in Lemma B.5.

Since $\hat{Z}_{t,i}, Z_{t,i}$ are both $\mathcal{F}_{t-1}^{N,i}$-measurable, we have $\mathcal{B}_{t,i} \in \mathcal{F}_{t-1}^{N,i}$.

Then we use a conditional version of Lemma B.5, which gives

$$\mathbb{P}(\mathcal{B}_{t,i}^c|\mathcal{F}_{t-1}^{N,i-1}) \geq 1 - \delta \qquad a.s.$$

then it's obvious that

$$\mathbb{P}\left( \bigcup_{t \leq T} \bigcup_{i \leq N} \mathcal{B}_{t,i} \right) \leq \sum_{\substack{t \leq T \\ i \leq N}} \mathbb{E}\Big[\mathbb{E}[\mathbb{I}_{\mathcal{B}_{t,i}}|\mathcal{F}_{t-1}^{N,i-1}]\Big] < NT\delta.$$

**Good events for rejection sampling (optimal proposal).** Afterwards, since our optimal proposal is sampled from rejection sampling (Algorithm 3), we need to define the bad event such that unsuccessful sampling occurs. Consider the random variable denoting whether or not the proposal of the $i$-th particle from time step $t-1$ to $t$ being successful as $I_{t,i} \in \{0,1\}$, i.e. let $\mathcal{E}_{acc}$ being the good event at $(t,i)$ in Theorem B.1, $I_{t,i} := \mathbb{I}_{\mathcal{E}_{acc}}$, then $\mathcal{B}'_{t,i} := \{I_{t,i} = 0\}$. Denote the sampling result at $(t,i)$ as a random variable $W_{t,i}$. Then the conditional version of Theorem B.1 implies that when using at most $N_s := 4C_{\mathrm{act}}\log(\delta^{-1})$ propose-reject steps, $\mathbb{P}(\mathcal{B}'_{t,i}|\mathcal{F}^{N,N}_{t-1}) < \delta$. Thus

$$\mathbb{P}\left(\bigcup_{t \leq T}\bigcup_{i \leq N}\mathcal{B}'_{t,i}\right) \leq \sum_{\substack{t \leq T \\ i \leq N}}\mathbb{E}\left[\mathbb{E}[\mathbb{I}_{\mathcal{B}'_{t,i}}|\mathcal{F}^{N,N}_{t-1}]\right] < NT\delta.$$

Note that $\sigma(I_{t,1}, W_{t,1}), \sigma(I_{t,2}, W_{t,2}), \cdots, \sigma(I_{t,N}, W_{t,N})$ are conditionally independent w.r.t. $\mathcal{F}^{N,N}_{t-1}$. And $\mathcal{F}^{N,0}_t = \sigma\left(\mathcal{F}^{N,N}_{t-1}\cup\left(\bigcup_{i=1}^N \sigma(I_{t,i}, W_{t,i})\right)\right)$.

And we now define for estimating conditional expectation, the good event at each step $\mathcal{E}_t := (\bigcup_{i=1}^N \mathcal{B}_{t,i})^c$ for $t \leq T$, and the overall good event as their intersection $\mathcal{E} := \bigcap_{t \leq T}\mathcal{E}_t$, $\mathcal{E}_{\leq t} := \bigcap_{t' \leq t}\mathcal{E}_{t'}$, $\mathcal{E}_{>t} := \bigcap_{t < t' \leq T}\mathcal{E}_{t'}$. We have also $\mathbb{P}(\mathcal{E}_t) \geq 1 - N\delta$, $\mathbb{P}(\mathcal{E}_{\leq t}) \geq 1 - NT\delta$ and $\mathbb{P}(\mathcal{E}_{>t}) \geq 1 - NT\delta$. For rejection sampling, the good event at each step is $\mathcal{E}'_t := \bigcap_{i=1}^N \mathcal{E}'_{t,i}$ where $\mathcal{E}'_{t,i} := (\mathcal{B}'_{t,i})^c$ for $t \leq T$, and $\mathcal{E}' := \bigcap_{t \leq T}\mathcal{E}'_t$, $\mathcal{E}'_{\leq t} := \bigcap_{t' \leq t}\mathcal{E}'_{t'}$, $\mathcal{E}'_{>t} := \bigcap_{t < t' \leq T}\mathcal{E}'_{t'}$. We have also $\mathbb{P}(\mathcal{E}'_t) \geq 1 - N\delta$, $\mathbb{P}(\mathcal{E}'_{\leq t}) \geq 1 - NT\delta$ and $\mathbb{P}(\mathcal{E}'_{>t}) \geq 1 - NT\delta$.

**Telescoping (with terminal step $n = T + 1$).** To apply our previous proof regarding telescoping, for any test function $f \in \mathrm{Osc}_1$ we define the random variable

$$Y_p := \left(\Phi_{p,n}(\eta^N_p) - \Phi_{p,n}(\Phi_p(\eta^N_{p-1}))\right)(f), \qquad n := T + 1.$$

Note that $Y_p$ is the difference between two random probability measures operating on the test function $f$, so $\|Y_p\|_\infty \leq \mathrm{osc}(f) = 1$. For the terminal FK step $p = T + 1$, there is no rejection sampling; we may set $\mathcal{E}'_{T+1} := \Omega$ and $\mathcal{E}_{T+1} := \Omega$ so that the notation below remains uniform.

Then we have

$$\mathbb{E}[Y_p|\mathcal{E} \cap \mathcal{E}'] = \mathbb{E}[Y_p\mathbb{I}_{\mathcal{E}}\mathbb{I}_{\mathcal{E}'}]/\mathbb{P}(\mathcal{E} \cap \mathcal{E}') = \mathbb{E}[Y_p\mathbb{I}_{\mathcal{E}_{\leq p}}\mathbb{I}_{\mathcal{E}'_{<p}}\mathbb{I}_{\mathcal{E}'_p}\mathbb{I}_{\mathcal{E}'_{>p}}\mathbb{I}_{\mathcal{E}_{>p}}]/\mathbb{P}(\mathcal{E} \cap \mathcal{E}')$$

$$= \mathbb{E}\left[\mathbb{I}_{\mathcal{E}_{\leq p}}\mathbb{I}_{\mathcal{E}'_{<p}}\mathbb{E}[Y_p\mathbb{I}_{\mathcal{E}'_p}\mathbb{I}_{\mathcal{E}'_{>p}}\mathbb{I}_{\mathcal{E}_{>p}}|\mathcal{F}^{N,N}_{p-1}]\right]/\mathbb{P}(\mathcal{E} \cap \mathcal{E}'),$$

and the last step is because $\mathbb{I}_{\mathcal{E}_{\leq p}}\mathbb{I}_{\mathcal{E}'_{<p}}$ is $\mathcal{F}^{N,N}_{p-1}$-measurable.

We then focus on $\left|\mathbb{E}[Y_p\mathbb{I}_{\mathcal{E}_{>p}}\mathbb{I}_{\mathcal{E}'_p}\mathbb{I}_{\mathcal{E}'_{>p}}|\mathcal{F}^{N,N}_{p-1}]\right|$, which can be bounded as follow

$$\left|\mathbb{E}[Y_p\mathbb{I}_{\mathcal{E}_{>p}}\mathbb{I}_{\mathcal{E}'_p}\mathbb{I}_{\mathcal{E}'_{>p}}|\mathcal{F}^{N,N}_{p-1}]\right| = \left|\mathbb{E}[Y_p\mathbb{I}_{\mathcal{E}'_p}|\mathcal{F}^{N,N}_{p-1}] - \mathbb{E}[Y_p\mathbb{I}_{\mathcal{E}'_p}\mathbb{I}_{\mathcal{E}^c_{>p}\cup(\mathcal{E}'_{>p})^c}|\mathcal{F}^{N,N}_{p-1}]\right|$$

$$\leq \left|\mathbb{E}[Y_p\mathbb{I}_{\mathcal{E}'_p}|\mathcal{F}^{N,N}_{p-1}]\right| + \underbrace{\|Y_p\mathbb{I}_{\mathcal{E}'_p}\|_\infty}_{\leq 1}\cdot\underbrace{\mathbb{P}(\mathcal{E}^c_{>p}\cup(\mathcal{E}'_{>p})^c|\mathcal{F}^{N,N}_{p-1})}_{<2NT\delta}$$

$$< \left|\mathbb{E}[Y_p|\mathcal{F}^{N,N}_{p-1},\mathcal{E}'_p]\right| + \underbrace{\left|\mathbb{E}[Y_p\mathbb{I}_{\mathcal{E}'_p}|\mathcal{F}^{N,N}_{p-1}]\right|}_{\leq 1}\mathbb{I}_{(\mathcal{E}'_p)^c} + 2NT\delta.$$

Where we used the fact that $\mathbb{E}[Y_p\mathbb{I}_{\mathcal{E}'_p}|\mathcal{F}^{N,N}_{p-1}]\mathbb{I}_{\mathcal{E}'_p} = \mathbb{E}[Y_p|\mathcal{F}^{N,N}_{p-1},\mathcal{E}'_p]\cdot\mathbb{P}(\mathcal{E}'_p|\mathcal{F}^{N,N}_{p-1})$.

Thus we write

$$\mathbb{E}[Y_p|\mathcal{E} \cap \mathcal{E}'] = \mathbb{E}\left[\mathbb{I}_{\mathcal{E}_{\leq p}}\mathbb{I}_{\mathcal{E}'_{<p}}\mathbb{E}[Y_p\mathbb{I}_{\mathcal{E}'_p}\mathbb{I}_{\mathcal{E}'_{>p}}\mathbb{I}_{\mathcal{E}_{>p}}|\mathcal{F}^{N,N}_{p-1}]\right]/\mathbb{P}(\mathcal{E} \cap \mathcal{E}')$$

$$< \mathbb{E}\left[\mathbb{I}_{\mathcal{E}_{\leq p}}\mathbb{I}_{\mathcal{E}'_{<p}}\left(\left|\mathbb{E}[Y_p|\mathcal{F}^{N,N}_{p-1},\mathcal{E}'_p]\right| + \mathbb{I}_{(\mathcal{E}'_p)^c} + 2NT\delta\right)\right]/\mathbb{P}(\mathcal{E} \cap \mathcal{E}')$$

$$\leq \frac{1}{1 - 2NT\delta}\left(\sup_{\mathcal{E}_{\leq p}}\left|\mathbb{E}[Y_p \mid \mathcal{F}^{N,N}_{p-1},\mathcal{E}'_p]\right| + N\delta + 2NT\delta\right).$$

**(a) Link to the perturbed FK mapping.** Let $\Psi_G$ denote the Boltzmann–Gibbs transform

$$\Psi_G(\mu)(dx) = \frac{G(x)\mu(dx)}{\mu(G)}.$$

Under $\omega \in \mathcal{E}_{\leq p}$, we have for the (random) normalizers

$$(1 - \xi)Z_{p,i} \leq \hat{Z}_{p,i} \leq (1 + \xi)Z_{p,i}, \qquad \forall i \in [N].$$

In particular, the induced potential ratio satisfies the uniform bounds

$$1 - \xi \leq \frac{G'_p}{G_p} \leq 1 + \xi \qquad \Rightarrow \qquad \left|\frac{G'_p}{G_p} - 1\right| \leq \xi.$$

A standard calculation then yields, for any probability measure $\mu$ and any $f \in \mathrm{Osc}_1$,

$$\left|(\Psi_{G'_p}(\mu) - \Psi_{G_p}(\mu))(f)\right| = \left|\frac{\mu(G'_p f)}{\mu(G'_p)} - \frac{\mu(G_p f)}{\mu(G_p)}\right| \leq \frac{\xi}{1 - \xi},$$

and therefore

$$\|\Psi_{G'_p}(\mu) - \Psi_{G_p}(\mu)\|_{\mathrm{TV}} \leq \frac{\xi}{1 - \xi}. \tag{33}$$

Since mutation by a Markov kernel is TV-contractive (and $M_{T+1} = \mathrm{Id}$ is a special case), the corresponding one-step FK mappings satisfy

$$\|\Phi'_p(\mu) - \Phi_p(\mu)\|_{\mathrm{TV}} \leq \frac{\xi}{1 - \xi}, \qquad \forall \mu. \tag{34}$$

**(b) Local (linear) bias bound.** Recall $u = D_{\nu_p^N}\Phi_{p,n}(f)$ and $\nu_p^N = \Phi_p(\eta_{p-1}^N)$. By the definition of $\Phi'_p$ and the above conditional expectation identity,

$$\mathbb{E}\left([\eta_p^N - \nu_p^N](u) \,\big|\, \mathcal{F}_{p-1}^{N,N}, \mathcal{E}'_p\right) = \frac{1}{N}\sum_{i=1}^N \left(K'_{p,\eta_{p-1}^N} u(\xi_{p-1}^{(N,i)}) - K_{p,\eta_{p-1}^N} u(\xi_{p-1}^{(N,i)})\right)$$

$$= \eta_{p-1}^N\left((K'_{p,\eta_{p-1}^N} - K_{p,\eta_{p-1}^N})u\right)$$

$$= \left(\Phi'_p(\eta_{p-1}^N) - \Phi_p(\eta_{p-1}^N)\right)(u).$$

Using $|(\mu - \nu)(g)| \leq \mathrm{osc}(g)\|\mu - \nu\|_{\mathrm{TV}}$ with $g = u$ and (34) yields, for every $\omega \in \mathcal{E}_{\leq p}$,

$$\left|\mathbb{E}\left([\eta_p^N - \nu_p^N](u) \,\big|\, \mathcal{F}_{p-1}^{N,N}, \mathcal{E}'_p\right)\right| \leq \mathrm{osc}(u) \|\Phi'_p(\eta_{p-1}^N) - \Phi_p(\eta_{p-1}^N)\|_{\mathrm{TV}}$$

$$\leq \frac{\xi}{1 - \xi} \beta(D\Phi_{p,n}), \tag{35}$$

where we used $\mathrm{osc}(u) \leq \beta(D\Phi_{p,n})\mathrm{osc}(f) \leq \beta(D\Phi_{p,n})$ since $f \in \mathrm{Osc}_1$.

**(c) Second-order remainder term.** By the same argument as in Theorem E.6, conditionally on $\mathcal{F}_{p-1}^{N,N}, \mathcal{E}'_p$,

$$\sup_{f \in \mathrm{Osc}_1} \left|\mathbb{E}\left[R^{\Phi_{p,n}}(\eta_p^N, \nu_p^N)(f) \,\big|\, \mathcal{F}_{p-1}^{N,N}, \mathcal{E}'_p\right]\right| \leq (\frac{1}{4N} + O(\xi^2))(q_{p,n}^2 - 1)\beta(D\Phi_{p,n}). \tag{36}$$

**(d) Combine (b)(c) into $\sup_{\mathcal{E}_{\leq p}} |\mathbb{E}[Y_p \mid \mathcal{F}_{p-1}^{N,N}, \mathcal{E}'_p]|$.** Combining (35) and (36), we obtain for every $\omega \in \mathcal{E}_{\leq p}$,

$$\left|\mathbb{E}\left[(\Phi_{p,n}(\eta_p^N) - \Phi_{p,n}(\nu_p^N))(f) \,\big|\, \mathcal{F}_{p-1}^{N,N}, \mathcal{E}'_p\right]\right| \leq \frac{\xi}{1 - \xi}\beta(D\Phi_{p,n}) + (\frac{1}{4N} + O(\xi^2))(q_{p,n}^2 - 1)\beta(D\Phi_{p,n}).$$

Hence,

$$\sup_{\mathcal{E}_{\leq p}} \left| \mathbb{E}\left[Y_p \mid \mathcal{F}_{p-1}^{N,N}, \mathcal{E}_p'\right] \right| \leq \frac{\xi}{1-\xi} \beta(D\Phi_{p,n}) + (\frac{1}{4N} + O(\xi^2))(q_{p,n}^2 - 1)\beta(D\Phi_{p,n}). \tag{37}$$

**(e) Sum over $p$ (telescoping) to control TV under $\mathcal{E}$.** Taking $\sup_{f \in \mathrm{Osc}_1}$ and summing over $p \leq n = T+1$ yields

$$\left\| \tilde{\pi}_T - \hat{\pi}_T |_{\mathcal{E}} \right\|_{\mathrm{TV}} = \sup_{f \in \mathrm{Osc}_1} \left| \sum_{p=1}^{T+1} \mathbb{E}[Y_p \mid \mathcal{E}] \right| \leq \sum_{p=1}^{T+1} \sup_{f \in \mathrm{Osc}_1} |\mathbb{E}[Y_p \mid \mathcal{E}]|$$

$$\leq \sum_{p=1}^{T+1} \left[ \frac{1}{1-2NT\delta} \left( \frac{\xi}{1-\xi} \beta(D\Phi_{p,T+1}) + (\frac{1}{4N} + O(\xi^2))(q_{p,T+1}^2 - 1)\beta(D\Phi_{p,T+1}) \right) + \frac{2NT\delta + N\delta}{1-2NT\delta} \right]. \tag{38}$$

**(f) Plug in the *exact* FK semigroup stability constants.** Under Assumption 3.2, we have the usual bounds for the *exact* FK semigroup (with terminal horizon $T+1$):

$$q_{p,T+1} \leq (1+\varepsilon)^{2(T+1-p)}, \qquad \beta(D\Phi_{p,T+1}) \leq 2q_{p,T+1} \leq 2(1+\varepsilon)^{2(T+1-p)}.$$

Consequently,

$$\sum_{p=1}^{T+1} \beta(D\Phi_{p,T+1}) \leq 2\sum_{k=0}^{T}(1+\varepsilon)^{2k} \leq 2(T+1)(1+\varepsilon)^{2T}, \tag{39}$$

$$\sum_{p=1}^{T+1}(q_{p,T+1}^2 - 1)\beta(D\Phi_{p,T+1}) \leq 2\sum_{p=1}^{T+1} q_{p,T+1}(q_{p,T+1}^2 - 1) \lesssim \varepsilon(T+1)^2(1+\varepsilon)^{6T}, \tag{40}$$

where (40) is the same geometric-series-with-$k$ factor bound used in our earlier optimal-proposal analysis. In the regime $\varepsilon = O(1/T)$ and $\delta_{\mathrm{TV}} = O(1)$, the exponential factors are absorbed into constants.

**(g) Choose parameters $N, N_m$ and instantiate the confidence $\delta$.** First, instantiate the $\delta$ used above as $\delta \leftarrow \delta/(4N(T+1))$ so that $\mathbb{P}(\mathcal{E} \cap \mathcal{E}') \geq 1 - \delta$. Next, choose $\xi \leq 1/2$ and set

$$\xi \asymp \frac{\delta_{\mathrm{TV}}}{T+1}, \qquad N \asymp \frac{\varepsilon(T+1)^2}{\delta_{\mathrm{TV}}}.$$

With these choices, using (38)–(40) and $2NT\delta$ negligible, the RHS of (38) is $\leq \delta_{\mathrm{TV}}$. Finally, recall $\xi = 2\sqrt{2}\sqrt{\frac{C_{\mathrm{act}}\log\frac{4}{\delta}}{N_m}}$. Thus it suffices to take

$$N_m \asymp \frac{C_{\mathrm{act}}(T+1)^2}{\delta_{\mathrm{TV}}^2}\log\left(\frac{N(T+1)}{\delta}\right).$$

Finally recall that we took $N_s := 4C_{\mathrm{act}}\log(\delta^{-1})$ time to finish a rejection sampling optimal proposal. Noticably, this is negligible comparing to $N_m$, as rejection sampling only require estimating the normalizing factor by constant relative error, while the latter requires up to $O(\delta_{\mathrm{TV}}/T)$.

**(h) Time complexity.** The total number of reward-model evaluations (or reference-kernel calls inside the MC normalizer estimation) is of order $N \cdot T \cdot N_m$ (the terminal step $T+1$ only adds an $O(N)$ resampling cost), hence

$$\mathrm{Time} \asymp NT(N_m + N_s) = \tilde{O}\left(\frac{\varepsilon(T+1)^2}{\delta_{\mathrm{TV}}} \cdot T \cdot \frac{C_{\mathrm{act}}(T+1)^2}{\delta_{\mathrm{TV}}^2}\log(\delta^{-1})\right) = \tilde{O}\left(\frac{\varepsilon C_{\mathrm{act}} T^5 \log(\delta^{-1})}{\delta_{\mathrm{TV}}^3}\right).$$

Moreover, since $C_{\mathrm{act}} \leq L(1+\varepsilon)$, in the regime $\varepsilon = O(1/T)$ we can write

$$\mathrm{Time} = \tilde{O}\left(\frac{\varepsilon L T^5 \log(\delta^{-1})}{\delta_{\mathrm{TV}}^3}\right),$$

which proves the result. $\qquad\square$

**SMC with arbitrary McKean kernel.** The following proposition provides a more general form of Theorem E.6, providing distinctions between different McKean kernels $K_{p,\eta}$. Even though various different kernels may correspond to the vary same Feynman-Kac flow, namely if they all satisfy (1), Theorem E.6 only provide an upper bound for all of them, and Proposition E.9 characterizes this distinction.

**Proposition E.9.** *Consider a SMC sampler with McKean transition kernel $K_{p,\eta}$ satisfying (1) for a FK model, then any $N \geq 1$ and any $n \geq 1$, set $\nu := \Phi_p(\eta_{p-1}^N)$, we have*

$$\|\bar{\eta}_n^N - \eta_n\|_{\mathrm{TV}} \leq \frac{1}{2N} \sum_{p=1}^n \left( q_{p,n} \, \beta(D\Phi_{p,n}) \cdot \mathbb{E}\left[ \sqrt{\eta_{p-1}^N \left( K_{p,\eta_{p-1}^N} \left( (G_{p,n,\nu} - K_{p,\eta_{p-1}^N} G_{p,n,\nu})^2 \right) \right)} \right] \right). \tag{41}$$

*Proof of Proposition E.9.* Fix $n \geq 1$ and $N \geq 1$, and let $f \in \mathrm{Osc}_1(E_n)$. By the bias telescoping (24),

$$[\bar{\eta}_n^N - \eta_n](f) = \sum_{p=1}^n \mathbb{E}\left[ \left( \Phi_{p,n}(\eta_p^N) - \Phi_{p,n}(\Phi_p(\eta_{p-1}^N)) \right)(f) \right].$$

For each $p$, set $\nu := \Phi_p(\eta_{p-1}^N) = \nu_p^N$. Applying the first-order expansion (Lemma E.1) to $\Phi_{p,n}$ at $(\eta, \mu) = (\nu, \eta_p^N)$ gives

$$\left( \Phi_{p,n}(\eta_p^N) - \Phi_{p,n}(\nu) \right)(f) = [\eta_p^N - \nu]\left( D_{p,n,\nu}(f) \right) + R^{\Phi_{p,n}}(\eta_p^N, \nu)(f).$$

Taking conditional expectation given $\mathcal{F}_{p-1}^N$ and using Lemma E.4–(25) kills the linear term, hence

$$\mathbb{E}\left[ \left( \Phi_{p,n}(\eta_p^N) - \Phi_{p,n}(\nu) \right)(f) \,\Big|\, \mathcal{F}_{p-1}^N \right] = \mathbb{E}\left[ R^{\Phi_{p,n}}(\eta_p^N, \nu)(f) \mid \mathcal{F}_{p-1}^N \right].$$

Using the explicit FK remainder form (20),

$$R^{\Phi_{p,n}}(\eta_p^N, \nu)(f) = -\frac{1}{\eta_p^N(G_{p,n,\nu})} [\eta_p^N - \nu](G_{p,n,\nu}) [\eta_p^N - \nu]\left( D_{p,n,\nu}(f) \right),$$

in which again, $D_{p,n,\eta}$ is a shorthand for $D_\eta \Phi_{p,n}$. By the definition of $q_{p,n}$ we have $1/\eta_p^N(G_{p,n,\nu}) \leq q_{p,n}$, hence

$$\left| \mathbb{E}\left[ R^{\Phi_{p,n}}(\eta_p^N, \nu)(f) \mid \mathcal{F}_{p-1}^N \right] \right| \leq q_{p,n} \, \mathbb{E}\left[ \left| [\eta_p^N - \nu](G_{p,n,\nu}) \right| \left| [\eta_p^N - \nu]\left( D_{p,n,\nu}(f) \right) \right| \,\Big|\, \mathcal{F}_{p-1}^N \right]$$

$$\leq q_{p,n} \left( \mathbb{E}\left[ [\eta_p^N - \nu](G_{p,n,\nu})^2 \mid \mathcal{F}_{p-1}^N \right] \right)^{1/2} \left( \mathbb{E}\left[ [\eta_p^N - \nu]\left( D_{p,n,\nu}(f) \right)^2 \mid \mathcal{F}_{p-1}^N \right] \right)^{1/2}.$$

By Lemma E.4–(27) applied to $u = G_{p,n,\nu}$,

$$\mathbb{E}\left[ [\eta_p^N - \nu](G_{p,n,\nu})^2 \mid \mathcal{F}_{p-1}^N \right] = \frac{1}{N} \eta_{p-1}^N \left( K_{p,\eta_{p-1}^N} \left( (G_{p,n,\nu} - K_{p,\eta_{p-1}^N} G_{p,n,\nu})^2 \right) \right).$$

For the $D$-term, Lemma E.4 and the elementary bound $\eta K((h - Kh)^2) \leq \frac{1}{4} \mathrm{osc}(h)^2$ yield

$$\mathbb{E}\left[ [\eta_p^N - \nu]\left( D_{p,n,\nu}(f) \right)^2 \mid \mathcal{F}_{p-1}^N \right] = \frac{1}{N} \eta_{p-1}^N K_{p,\eta_{p-1}^N} \left( (D_{p,n,\nu}(f) - K_{p,\eta_{p-1}^N} D_{p,n,\nu}(f))^2 \right) \leq \frac{1}{4N} \mathrm{osc}(D_{p,n,\nu}(f))^2.$$

Using $\mathrm{osc}(D_{p,n,\nu}(f)) \leq \beta(D\Phi_{p,n}) \mathrm{osc}(f) \leq \beta(D\Phi_{p,n})$ for $f \in \mathrm{Osc}_1(E_n)$, we obtain

$$\left( \mathbb{E}\left[ [\eta_p^N - \nu]\left( D_{p,n,\nu}(f) \right)^2 \mid \mathcal{F}_{p-1}^N \right] \right)^{1/2} \leq \frac{1}{2\sqrt{N}} \beta(D\Phi_{p,n}).$$

Combining the above displays gives, for any $f \in \mathrm{Osc}_1(E_n)$,

$$\left| \mathbb{E}\left[ \left( \Phi_{p,n}(\eta_p^N) - \Phi_{p,n}(\nu) \right)(f) \right] \right| \leq \frac{1}{2N} q_{p,n} \beta(D\Phi_{p,n}) \mathbb{E}\left[ \sqrt{\eta_{p-1}^N \left( K_{p,\eta_{p-1}^N} \left( (G_{p,n,\nu} - K_{p,\eta_{p-1}^N} G_{p,n,\nu})^2 \right) \right)} \right].$$

Summing over $p = 1, \dots, n$ and taking the supremum over $f \in \mathrm{Osc}_1(E_n)$ yields (41). $\qquad\square$

**Corollary E.10** (Example McKean Kernel: stratified (quantile) transition). *Consider the **naive proposal** FK model and work in the regime $\varepsilon = O(1/T)$. Recall we append a terminal step $T+1$ with $M_{T+1} = \mathrm{Id}$ so that the target is $\eta_{T+1}$. Let $\bar{\eta}^N_{T+1} := \mathbb{E}[\eta^N_{T+1}]$ denote the SMC output distribution.*

*Define the proxy of $Q_{p,T+1}(1)$ by the one-step ratio for any $p \in [T]$ as*

$$\widetilde{Q}_{p,T+1}(1)(s_{1:p}) := \frac{\hat{V}(s_0, s_{1:p})}{\hat{V}(s_0, s_{1:p-1})},$$

*and let $\nu^N_p := \Phi_p(\eta^N_{p-1})$. For each $p$, construct a partition $(\mathsf{E}_{p,i})^N_{i=1}$ of $E_p$ by $\nu^N_p$-quantiles of $\widetilde{Q}_{p,T+1}(1)$, so that $\nu^N_p(\mathsf{E}_{p,i}) = 1/N$ and $\mathsf{E}_{p,i}$ are ordered by increasing $\widetilde{Q}_{p,T+1}(1)$. Define the index-stratified transition*

$$K^{\mathrm{q}}_{p,\eta^N_{p-1}}(X^i_{p-1}, dy) := \nu^N_p(dy \mid \mathsf{E}_{p,i}), \qquad i = 1, \ldots, N.$$

*Then a sufficient condition for $\|\bar{\eta}^N_{T+1} - \eta_{T+1}\|_{\mathrm{TV}} \leq \delta_{\mathrm{TV}}$ is*

$$N \gtrsim \max\left\{\left(\frac{L^6 T}{\delta_{\mathrm{TV}}}\right)^{2/3}, \; \frac{\varepsilon L^6 T^2}{\delta_{\mathrm{TV}}}\right\}.$$

*Proof of Corollary E.10.* Let $n := T+1$ be the FK horizon (with $M_{T+1} = \mathrm{Id}$). We follow the kernel-sensitive bias decomposition used in Proposition E.9, but note that $K^{\mathrm{q}}$ is *stratified* rather than i.i.d. across particles. The key point is that stratification still preserves the same *conditional unbiasedness* (McKean consistency), so the linear term vanishes exactly as in the i.i.d. case; the remaining contribution is controlled by an average of *local conditional variances* across strata.

**Step 1 (FK constants under naive proposal, horizon $n = T + 1$).** By Lemma E.2, for the naive proposal and $p \leq T$,

$$q_{p,n} \leq L^2(1+\varepsilon)^{2(n-p-1)} = L^2(1+\varepsilon)^{2(T-p)} \qquad \text{and} \qquad \beta(D\Phi_{p,n}) \leq 2L^2(1+\varepsilon)^{2(T-p)}.$$

Hence

$$q_{p,n}\beta(D\Phi_{p,n}) \leq 2L^4(1+\varepsilon)^{4(T-p)} = O(L^4) \quad \text{under } \varepsilon = O(1/T),$$

and therefore $\sum_{p=1}^T q_{p,n}\beta(D\Phi_{p,n}) = O(L^4 T)$.

**Step 2 (Stratified update: conditional unbiasedness and a well-defined variance functional).** Fix $p$ and write $\nu := \nu^N_p = \Phi_p(\eta^N_{p-1})$ and $\mathsf{E}_i := \mathsf{E}_{p,i}$. Under the stratified transition, conditional on $\mathcal{F}^N_{p-1}$, the new particles are independent with

$$X^i_p \sim \nu(\cdot \mid \mathsf{E}_i), \qquad i = 1, \ldots, N.$$

Since $\nu(\mathsf{E}_i) = 1/N$, for any bounded measurable $h$,

$$\frac{1}{N}\sum_{i=1}^N \nu(\cdot \mid \mathsf{E}_i) = \nu(\cdot),$$

so the stratified update is conditionally unbiased:

$$\mathbb{E}[\eta^N_p(h) \mid \mathcal{F}^N_{p-1}] = \frac{1}{N}\sum_{i=1}^N \nu(h \mid \mathsf{E}_i) = \nu(h).$$

Thus the same first-order (linear) term cancellation used in Proposition E.9 continues to hold.

What differs is the second moment. Define the *stratified variance functional*

$$\mathcal{V}^{\mathrm{q}}_{p,\nu}(h) := \frac{1}{N}\sum_{i=1}^N \mathrm{Var}_{\nu(\cdot|\mathsf{E}_i)}(h). \tag{42}$$

Then, using independence across $i$ conditional on $\mathcal{F}_{p-1}^N$,

$$\mathbb{E}\left[(\eta_p^N(h) - \nu(h))^2 \mid \mathcal{F}_{p-1}^N\right] = \frac{1}{N^2}\sum_{i=1}^N \mathrm{Var}_{\nu(\cdot|\mathsf{E}_i)}(h) = \frac{1}{N}\cdot\frac{1}{N}\mathcal{V}_{p,\nu}^{\mathsf{q}}(h).$$

Moreover, for each stratum, $\mathrm{Var}_{\nu(\cdot|\mathsf{E}_i)}(h) \le \frac{1}{4}\mathrm{osc}_{\mathsf{E}_i}(h)^2$, so

$$\sqrt{\mathcal{V}_{p,\nu}^{\mathsf{q}}(h)} \le \frac{1}{2}\sqrt{\frac{1}{N}\sum_{i=1}^N \mathrm{osc}_{\mathsf{E}_i}(h)^2} \le \frac{1}{2\sqrt{N}}\sum_{i=1}^N \mathrm{osc}_{\mathsf{E}_i}(h), \tag{43}$$

where the last step uses $\sqrt{\sum a_i^2} \le \sum |a_i|$.

**Step 3 (Normalization and mismatch from using $\widetilde{Q}_{p,n}(1)$-quantiles).** Recall $G_{p,n,\nu}(x) = Q_{p,n}(1)(x)/\nu(Q_{p,n}(1))$ and quantiles are invariant to the normalization. Oscillations rescale as

$$\mathrm{osc}_{\mathsf{E}_i}(G_{p,n,\nu}) = \frac{1}{\nu(Q_{p,n}(1))}\mathrm{osc}_{\mathsf{E}_i}(Q_{p,n}(1)).$$

Under naive proposal, Lemma E.2 implies $1/\nu(Q_{p,n}(1)) = O(L)$ in the regime $\varepsilon = O(1/T)$ (absorbing $(1+\varepsilon)^{O(T)}$).

Next, chaining the one-step Bellman-error control over the remaining horizon (as in the proof of Lemma E.2), one obtains a pointwise multiplicative approximation

$$Q_{p,n}(1)(x) = \widetilde{Q}_{p,n}(1)(x)\cdot(1\pm O(\varepsilon T)), \qquad \forall x \in E_p.$$

Let $\alpha := O(\varepsilon T)$. This yields the deterministic oscillation bound on each stratum:

$$\mathrm{osc}_{\mathsf{E}_i}(Q_{p,n}(1)) \le (1+\alpha)\,\mathrm{osc}_{\mathsf{E}_i}(\widetilde{Q}_{p,n}(1)) + O(\alpha)\,\|\widetilde{Q}_{p,n}(1)\|_\infty.$$

Summing over $i$ and using the ordering of strata by $\widetilde{Q}_{p,n}(1)$,

$$\sum_{i=1}^N \mathrm{osc}_{\mathsf{E}_i}(\widetilde{Q}_{p,n}(1)) \le \mathrm{osc}(\widetilde{Q}_{p,n}(1)) = O(L), \qquad \|\widetilde{Q}_{p,n}(1)\|_\infty = O(L),$$

by Assumption 3.1. Hence

$$\sum_{i=1}^N \mathrm{osc}_{\mathsf{E}_i}(Q_{p,n}(1)) = O(L) + O(\alpha)\,O(L)\,N,$$

and therefore, using $1/\nu(Q_{p,n}(1)) = O(L)$,

$$\sum_{i=1}^N \mathrm{osc}_{\mathsf{E}_i}(G_{p,n,\nu}) = O(L^2) + O(\alpha)\,O(L^2)\,N.$$

Plugging $h = G_{p,n,\nu}$ into (43) gives

$$\sqrt{\mathcal{V}_{p,\nu}^{\mathsf{q}}(G_{p,n,\nu})} \le O\left(\frac{L^2}{\sqrt{N}}\right) + O(\alpha)\,O(L^2).$$

**Step 4 (Combine).** Insert the above display into the kernel-sensitive bias bound (whose prefactor is $1/N$), and use Step 1 to obtain

$$\|\bar{\eta}_{T+1}^N - \eta_{T+1}\|_{\mathrm{TV}} \le O\left(\sum_{p=1}^T \frac{L^4}{N}\left(\frac{L^2}{\sqrt{N}} + \alpha L^2\right)\right) = O\left(\frac{L^6 T}{N^{3/2}}\right) + O\left(\frac{L^6 \alpha T}{N}\right).$$

Since $\alpha = O(\varepsilon T)$ and we are in the regime $\varepsilon = O(1/T)$, the claimed sufficient condition follows. $\qquad\square$

*Remark* E.11 (From i.i.d. McKean transitions to stratified resampling). Proposition E.9 is stated for a (possibly $\eta$-dependent) Markov kernel $K_{p,\eta}$ that is applied *identically* across particles at time $p$, so that, conditional on $\mathcal{F}^N_{p-1}$, the offspring $\{X^i_p\}^N_{i=1}$ are i.i.d. from the same law $\Phi_p(\eta^N_{p-1}) = \eta^N_{p-1} K_{p,\eta^N_{p-1}}$. In Corollary E.10, the "quantile/stratified" update instead samples

$$X^i_p \sim \nu^N_p(\cdot \mid \mathsf{E}_{p,i}), \qquad i = 1, \ldots, N,$$

where $\{\mathsf{E}_{p,i}\}^N_{i=1}$ is a measurable partition such that $\nu^N_p(\mathsf{E}_{p,i}) = 1/N$. This update is *not* representable as applying a single kernel identically to all particles; however, it still satisfies the same McKean consistency (conditional unbiasedness): for any bounded measurable $h$,

$$\mathbb{E}\left[\eta^N_p(h) \mid \mathcal{F}^N_{p-1}\right] = \frac{1}{N}\sum^N_{i=1} \nu^N_p(h \mid \mathsf{E}_{p,i}) = \nu^N_p(h) = \Phi_p(\eta^N_{p-1})(h).$$

Consequently, the first-order (linear) term in the telescoping / first-order decomposition underlying Proposition E.9 cancels *verbatim* for stratified resampling as well. The only modification is in the second-moment control: the i.i.d. conditional variance proxy $\eta^N_{p-1} K_{p,\eta^N_{p-1}}((h - K_{p,\eta^N_{p-1}} h)^2)$ should be replaced by the *stratified variance functional*

$$\mathcal{V}^{\mathrm{q}}_{p,\nu}(h) := \frac{1}{N}\sum^N_{i=1} \mathrm{Var}_{\nu(\cdot|\mathsf{E}_{p,i})}(h), \qquad \nu := \nu^N_p,$$

together with the deterministic oscillation bound $\sqrt{\mathcal{V}^{\mathrm{q}}_{p,\nu}(h)} \leq \frac{1}{2\sqrt{N}}\sum^N_{i=1} \mathrm{osc}_{\mathsf{E}_{p,i}}(h)$. With this substitution, the remaining steps of Proposition E.9 apply unchanged, yielding Corollary E.10.

Stratified resampling has been extensively studied in the particle filtering literature; see, e.g., (Douc & Cappé, 2005). The update considered here can be viewed as a particular stratification rule that is *adapted* to the quantiles of a designated function (in our case, the proxy lookahead-to-go), so that each stratum carries equal mass while concentrating resolution where this function varies most.

## F. Proofs in Section 6

**Lemma F.1** (Uniform TV contraction for independent Metropolis–Hastings)**.** *Let* $(\mathsf{E}, \mathcal{E})$ *be a measurable space. Let* $\pi_{\mathrm{ref}}$ *be a probability measure on* $\mathsf{E}$ *and let* $G, R : \mathsf{E} \to (0, \infty)$ *be measurable with*

$$0 < Z_G := \pi_{\mathrm{ref}}(G) < \infty, \qquad 0 < Z_R := \pi_{\mathrm{ref}}(R) < \infty.$$

*Define the target and proposal distributions*

$$\tilde{\pi}(dx) := \frac{G(x)}{Z_G} \, \pi_{\mathrm{ref}}(dx), \qquad q(dx) := \frac{R(x)}{Z_R} \, \pi_{\mathrm{ref}}(dx).$$

*Consider the* independent *Metropolis–Hastings kernel* $K$ *with target* $\tilde{\pi}$ *and proposal* $q$, *i.e.*

$$K(x, dy) = \alpha(x, y) \, q(dy) + \left(1 - \int \alpha(x, z) \, q(dz)\right) \delta_x(dy), \qquad \alpha(x, y) := 1 \wedge \frac{\tilde{\pi}(dy) \, q(dx)}{\tilde{\pi}(dx) \, q(dy)} = 1 \wedge \frac{G(y) R(x)}{G(x) R(y)}.$$

*Assume there exists* $b \geq 1$ *such that*

$$\operatorname*{ess\,sup}_{x \in \mathsf{E}} \max\left\{ \frac{G(x)}{R(x)}, \frac{R(x)}{G(x)} \right\} \, \leq \, b.$$

*Then:*

*(i)* (***Minorization***) *For every* $x \in \mathsf{E}$,

$$K(x, \cdot) \, \geq \, \frac{1}{b^2} \, \tilde{\pi}(\cdot).$$

*(ii)* (***Uniform TV contraction***) *For any probability measures* $\mu, \nu$ *on* $(\mathsf{E}, \mathcal{E})$ *and any integer* $m \geq 1$,

$$\|\mu K^m - \nu K^m\|_{\mathrm{TV}} \, \leq \, \left(1 - \frac{1}{b^2}\right)^m \|\mu - \nu\|_{\mathrm{TV}}.$$

*In particular, the Dobrushin coefficient satisfies* $\beta(K) \leq 1 - \frac{1}{b^2}$.

*Proof.* Write the Radon–Nikodym derivative of $\tilde{\pi}$ w.r.t. $q$:

$$w(x) := \frac{d\tilde{\pi}}{dq}(x) = \frac{G(x)/Z_G}{R(x)/Z_R} = \frac{Z_R}{Z_G} \cdot \frac{G(x)}{R(x)}.$$

By the assumption, $\frac{G}{R} \leq b$ $q$-a.e. Also, since $\frac{G}{R} \in [1/b, b]$ $\pi_{\mathrm{ref}}$-a.e., we have $Z_G = \pi_{\mathrm{ref}}(G) \in [\frac{1}{b} Z_R, b Z_R]$, hence $\frac{Z_R}{Z_G} \leq b$. Therefore $w(x) \leq b^2$ $q$-a.e. Denote $M := \operatorname{ess\,sup}_q w \leq b^2$.

For independent MH, $\alpha(x, y) = 1 \wedge \frac{w(y)}{w(x)}$. Using $w(x) \leq M$ and $1 \wedge u \geq u/M$ for $u \geq 0$, we get

$$\alpha(x, y) = 1 \wedge \frac{w(y)}{w(x)} \, \geq \, \frac{w(y)}{M}.$$

Hence for any measurable $A$,

$$K(x, A) \, \geq \, \int_A \alpha(x, y) \, q(dy) \, \geq \, \frac{1}{M} \int_A w(y) \, q(dy) \, = \, \frac{1}{M} \tilde{\pi}(A) \, \geq \, \frac{1}{b^2} \tilde{\pi}(A),$$

which proves (i).

For (ii), (i) implies the usual Doeblin decomposition: $K(x, \cdot) = \frac{1}{b^2} \tilde{\pi}(\cdot) + (1 - \frac{1}{b^2}) K'(x, \cdot)$ for some Markov kernel $K'$. Thus for any signed measure $\Delta$ with $\Delta(\mathsf{E}) = 0$,

$$\|\Delta K\|_{\mathrm{TV}} \leq \left(1 - \frac{1}{b^2}\right) \|\Delta\|_{\mathrm{TV}},$$

and iterating yields (ii). □

*Proof of Theorem 6.1.* We consider the path space of importance resampling, where for each $h \in [H]$, the random variable

$$Y_h := ((S_1^{(h,1)}, S_1^{(h,2)}, \cdots, S_1^{(h,M)}, A_1^h), (S_2^{(h,1)}, S_2^{(h,2)}, \cdots, S_2^{(h,M)}, A_2^h), \cdots, (S_T^{(h,1)}, S_T^{(h,2)}, \cdots, S_T^{(h,M)}, A_T^h))$$

denote the $h$-th Metropolis-Hastings proposal, where $A_t^h \in [M]$ is the index of the resampled state among $M$ candidates. We also write $S_{1:t}^{[h]} := (S_1^{(h,A_1^h)}, S_2^{(h,A_2^h)}, \cdots, S_t^{(h,A_t^h)})$. Then the acceptance rate $\alpha(X_{h-1}, Y_h)$ is calculated, and $U_h \sim$ Unif$(0,1)$ is compared with the acceptance rate to decide whether or not we accept $Y_h$. If so $X_h = Y_h$, otherwise $X_h = X_{h-1}$.

We use $\bar{Z}_t^{[h]} := \frac{1}{M} \sum_{j=1}^M \hat{V}(s_0, (S_{1:t-1}^{[h]}, S_t^{(h,j)}))$ as the average over $M$ candidate reward given then prefix $S_{1:t}^{[h]}$.

To apply the Metropolis-Hastings algorithm, one typically calculate the probability of generating the entire $Y_h$ as

$$\mathbb{P}(Y_h) = \prod_{t=1}^T \left( \frac{\hat{V}(s_0, (S_{1:t-1}^{[h]}, S_t^{(h,A_t^h)}))}{M \cdot \bar{Z}_t^{[h]}} \prod_{j=1}^M \pi_{\text{ref}}(S_t^{(h,j)} | S_{1:t-1}^{[h]}) \right).$$

Correspondingly, the target distribution is chosen to be

$$\mathbb{P}(Y_h') \propto \hat{V}(S_0, S_{1:T}^{[h]}) \prod_{t=1}^T \left( \frac{1}{M} \prod_{j=1}^M \pi_{\text{ref}}(S_t^{(h,j)} | S_{1:t-1}^{[h]}) \right)$$

of which the marginal distribution of $S_{1:T}^{[h]}$ (or can be viewed as a deterministic transform of the random variable $Y_h \mapsto S_{1:T}^{[h]}$) is $\mathbb{P}(S_{1:T}^{[h]}) \propto \pi_{\text{ref}}(S_{1:T}^{[h]} | s_0) \cdot \hat{V}(S_0, S_{1:T}^{[h]})$, which is our desired $\tilde{\pi}_T$.

We now define the good set $\mathsf{G}$ at each $h \in [H]$ to be

$$Y_h \in \mathsf{G} \iff \forall t \in [T], \quad \left| \frac{\bar{Z}_t^{[h]}}{\mathbb{E}_{s_t \sim \pi_{\text{ref}}(\cdot | s_{1:t-1}^{[h]})}[\hat{V}(s_0, (s_{1:t-1}^{[h]}, s_t))]} - 1 \right| \leq \xi.$$

And we define the good event $\mathcal{E}$ as

$$\mathcal{E} := \bigcap_{h=2}^{H+1} \{Y_h \in \mathsf{G}\} \cap \{X_1 \in \mathsf{G}\}.$$

Now we claim that $X_1, X_2, \cdots, X_{H+1}$ is still a Markov chain conditioning on the good event $\mathcal{E}$. To show this, first define the filtration

$$\mathcal{F}_h := \sigma(X_1, (Y_2.U_2), (Y_3, U_3), \cdots, (Y_h, U_h)).$$

Since every $Y_h$ follows an i.i.d. distribution which we name $Q(\cdot)$, we consider the conditional law of $Y_h$ within the good set $\mathsf{G}$ as $Q(\cdot | \mathsf{G})$. It's easy to see that conditioning on the good event $\mathcal{E}$, the $\{Y_h\}_{h=1}^{H+1}$ are still i.i.d. distributed according to $Q_{\mathsf{G}} := Q(\cdot | \mathsf{G})$, and are independent of $U_1, U_2, \cdots$. We also denote the target distribution as $\mathbb{P}(Y_{H+1}') \sim \Pi(\cdot)$, of which the conditional version is $\Pi_{\mathsf{G}} := \Pi(\cdot | \mathsf{G})$. We define the conditional MH kernel

$$K_{\mathsf{G}}(x, dy) := Q_{\mathsf{G}}(dy)\alpha(x, y) + \left(1 - \int Q_{\mathsf{G}}(du)\alpha(x, u)\right) \delta_x(dy).$$

Our goal is to show that for any bounded measurable $f$,

$$\mathbb{E}[f(X_{h+1}) | \mathcal{E}, \mathcal{F}_h] = K_{\mathsf{G}}f(X_h) \quad \text{almost surely on } \mathcal{E}.$$

Recall that (by definition of MH update)

$$X_{h+1} = \begin{cases} Y_{h+1}, & U_{h+1} \leq \alpha(X_h, Y_{h+1}), \\ X_h, & U_{h+1} > \alpha(X_h, Y_{h+1}). \end{cases}$$

Equivalently,

$$f(X_{h+1}) = f(Y_{h+1})\,\mathbb{I}\{U_{h+1} \le \alpha(X_h, Y_{h+1})\} + f(X_h)\,\mathbb{I}\{U_{h+1} > \alpha(X_h, Y_{h+1})\}.$$

We now compute the conditional expectation given $\sigma(\mathcal{F}_h \cup \{\mathcal{E}\})$. On $\mathcal{E}$, we have $Y_{h+1} \in \mathsf{G}$ and (by the construction of $\mathcal{E}$) the proposal randomness at step $h+1$ is independent of $\mathcal{F}_h$ and has the conditional law $Q_{\mathsf{G}}$; moreover $U_{h+1} \sim \mathrm{Unif}(0,1)$ is independent of $(\mathcal{F}_h, Y_{h+1})$ (still under the conditioning on $\mathcal{E}$). Also note that $X_h$ is $\mathcal{F}_h$-measurable.

Therefore, on $\mathcal{E}$,

$$\begin{aligned}
\mathbb{E}\left[f(Y_{h+1})\,\mathbb{I}\{U_{h+1} \le \alpha(X_h, Y_{h+1})\} \mid \mathcal{E}, \mathcal{F}_h\right] &= \mathbb{E}\left[\mathbb{E}\left[f(Y_{h+1})\,\mathbb{I}\{U_{h+1} \le \alpha(X_h, Y_{h+1})\} \mid \mathcal{E}, \mathcal{F}_h, Y_{h+1}\right] \mid \mathcal{E}, \mathcal{F}_h\right] \\
&= \mathbb{E}\left[f(Y_{h+1})\,\mathbb{P}\left(U_{h+1} \le \alpha(X_h, Y_{h+1}) \mid \mathcal{E}, \mathcal{F}_h, Y_{h+1}\right) \mid \mathcal{E}, \mathcal{F}_h\right] \\
&= \mathbb{E}\left[f(Y_{h+1})\,\alpha(X_h, Y_{h+1}) \mid \mathcal{E}, \mathcal{F}_h\right] \\
&= \int Q_{\mathsf{G}}(dy)\,f(y)\,\alpha(X_h, y).
\end{aligned}$$

Similarly, on $\mathcal{E}$,

$$\begin{aligned}
\mathbb{E}\left[\mathbb{I}\{U_{h+1} > \alpha(X_h, Y_{h+1})\} \mid \mathcal{E}, \mathcal{F}_h\right] &= 1 - \mathbb{E}\left[\mathbb{I}\{U_{h+1} \le \alpha(X_h, Y_{h+1})\} \mid \mathcal{E}, \mathcal{F}_h\right] \\
&= 1 - \int Q_{\mathsf{G}}(dy)\,\alpha(X_h, y).
\end{aligned}$$

Since $f(X_h)$ is $\mathcal{F}_h$-measurable, combining the two displays yields that, on $\mathcal{E}$,

$$\begin{aligned}
\mathbb{E}[f(X_{h+1}) \mid \mathcal{E}, \mathcal{F}_h] &= \int Q_{\mathsf{G}}(dy)\,f(y)\,\alpha(X_h, y) + f(X_h)\left(1 - \int Q_{\mathsf{G}}(dy)\,\alpha(X_h, y)\right) \\
&= K_{\mathsf{G}} f(X_h).
\end{aligned}$$

This proves

$$\mathbb{E}[f(X_{h+1}) \mid \mathcal{E}, \mathcal{F}_h] = K_{\mathsf{G}} f(X_h) \qquad \text{a.s. on } \mathcal{E},$$

as desired.

Therefore, the conditional law satisfies

$$\mathrm{Law}(X_{H+1})|_{\mathcal{E}} = \mathrm{Law}(X_1)|_{\mathcal{E}} K_{\mathsf{G}}^H.$$

Also, we prove that $\Pi_{\mathsf{G}}$ is the stationary distribution of $K_{\mathsf{G}}$. Recall that $Q_{\mathsf{G}} := Q(\cdot \mid \mathsf{G})$ and $\Pi_{\mathsf{G}} := \Pi(\cdot \mid \mathsf{G})$ are both supported on $\mathsf{G}$. Since $\Pi \ll Q$ on the path space, we also have $\Pi_{\mathsf{G}} \ll Q_{\mathsf{G}}$ on $\mathsf{G}$. Define the Radon–Nikodym derivative

$$r(x) := \frac{d\Pi_{\mathsf{G}}}{dQ_{\mathsf{G}}}(x), \qquad x \in \mathsf{G}.$$

Note that $r$ equals $w := d\Pi/dQ$ up to a positive constant on $\mathsf{G}$, hence

$$\alpha(x, y) = 1 \wedge \frac{w(y)}{w(x)} = 1 \wedge \frac{r(y)}{r(x)}, \qquad x, y \in \mathsf{G}.$$

We first verify detailed balance for the off-diagonal part. For any $x, y \in \mathsf{G}$,

$$\begin{aligned}
\Pi_{\mathsf{G}}(dx)\,Q_{\mathsf{G}}(dy)\,\alpha(x, y) &= r(x)\,Q_{\mathsf{G}}(dx)\,Q_{\mathsf{G}}(dy)\left(1 \wedge \frac{r(y)}{r(x)}\right) \\
&= Q_{\mathsf{G}}(dx)\,Q_{\mathsf{G}}(dy)\,\min\{r(x), r(y)\} \\
&= r(y)\,Q_{\mathsf{G}}(dy)\,Q_{\mathsf{G}}(dx)\left(1 \wedge \frac{r(x)}{r(y)}\right) \\
&= \Pi_{\mathsf{G}}(dy)\,Q_{\mathsf{G}}(dx)\,\alpha(y, x).
\end{aligned}$$

Therefore, for any measurable $B \subseteq \mathsf{G}$,

$$(\Pi_{\mathsf{G}} K_{\mathsf{G}})(B) = \int_{\mathsf{G}} \Pi_{\mathsf{G}}(dx) \int_{\mathsf{G}} Q_{\mathsf{G}}(dy) \, \alpha(x,y) \, \mathbb{I}_B(y) + \int_{\mathsf{G}} \Pi_{\mathsf{G}}(dx) \Big( 1 - \int_{\mathsf{G}} Q_{\mathsf{G}}(du) \, \alpha(x,u) \Big) \mathbb{I}_B(x)$$

$$=: I_1 + I_2.$$

We simplify $I_1$ using the detailed balance identity above and Fubini-Tonelli's theorem:

$$I_1 = \int_{\mathsf{G}} \int_{\mathsf{G}} \Pi_{\mathsf{G}}(dx) \, Q_{\mathsf{G}}(dy) \, \alpha(x,y) \, \mathbb{I}_B(y) = \int_{\mathsf{G}} \int_{\mathsf{G}} \Pi_{\mathsf{G}}(dy) \, Q_{\mathsf{G}}(dx) \, \alpha(y,x) \, \mathbb{I}_B(y)$$

$$= \int_{\mathsf{G}} \Pi_{\mathsf{G}}(dy) \, \mathbb{I}_B(y) \int_{\mathsf{G}} Q_{\mathsf{G}}(dx) \, \alpha(y,x) = \int_{\mathsf{G}} \Pi_{\mathsf{G}}(dx) \, \mathbb{I}_B(x) \int_{\mathsf{G}} Q_{\mathsf{G}}(du) \, \alpha(x,u).$$

Plugging this into $I_2$ gives

$$(\Pi_{\mathsf{G}} K_{\mathsf{G}})(B) = \int_{\mathsf{G}} \Pi_{\mathsf{G}}(dx) \, \mathbb{I}_B(x) \int_{\mathsf{G}} Q_{\mathsf{G}}(du) \, \alpha(x,u) + \int_{\mathsf{G}} \Pi_{\mathsf{G}}(dx) \Big( 1 - \int_{\mathsf{G}} Q_{\mathsf{G}}(du) \, \alpha(x,u) \Big) \mathbb{I}_B(x)$$

$$= \int_{\mathsf{G}} \Pi_{\mathsf{G}}(dx) \, \mathbb{I}_B(x) = \Pi_{\mathsf{G}}(B).$$

Hence $\Pi_{\mathsf{G}} K_{\mathsf{G}} = \Pi_{\mathsf{G}}$, i.e., $\Pi_{\mathsf{G}}$ is a stationary distribution of $K_{\mathsf{G}}$.

Therefore a standard Dobrushin contraction statement (Lemma F.1) gives

$$\|\mathrm{Law}(X_{H+1})|_{\mathcal{E}} - \Pi_{\mathsf{G}}\|_{\mathrm{TV}} \le (1 - b^{-2})^H \|\mathrm{Law}(X_1)|_{\mathcal{E}} - \Pi_{\mathsf{G}}\|_{\mathrm{TV}},$$

where $b := \left( (1+\varepsilon) \dfrac{1+\xi}{1-\xi} \right)^{T-1}$. Thus

$$\|\mathrm{Law}(X_{H+1})|_{\mathcal{E}} - \Pi\|_{\mathrm{TV}} \le \|\mathrm{Law}(X_{H+1})|_{\mathcal{E}} - \Pi_{\mathsf{G}}\|_{\mathrm{TV}} + \|\Pi - \Pi_{\mathsf{G}}\|_{\mathrm{TV}}$$

$$\le (1 - b^{-2})^H \|\mathrm{Law}(X_1)|_{\mathcal{E}} - \Pi_{\mathsf{G}}\|_{\mathrm{TV}} + \|\Pi - \Pi_{\mathsf{G}}\|_{\mathrm{TV}}$$

$$\le 2(1 - b^{-2})^H + 2\mathbb{P}(Y \notin \mathsf{G}).$$

Notice that $S_{1:T}^{[H+1]}$ is obtained by a deterministic transformation (marginalization) $g : X_{H+1} \mapsto S_{1:T}^{[H+1]}$, and the marginalized version of $\Pi$ is exactly $\tilde{\pi}_T$, using the data processing inequality yields

$$\|\mathrm{Law}(S_{1:T}^{[H+1]})|_{\mathcal{E}} - \tilde{\pi}_T\|_{\mathrm{TV}} \le \|\mathrm{Law}(X_{H+1})|_{\mathcal{E}} - \Pi\|_{\mathrm{TV}}.$$

Thus in the regime where $\varepsilon = O(1/T)$ and $\xi = O(1/T)$, we have $b = O(1)$. Therefore let $(1 - b^{-2})^H \le \delta_{\mathrm{TV}}/4$ and $\mathbb{P}(Y \notin \mathsf{G}) \le \delta_{\mathrm{TV}}/4$ yields $H = O(\log(\delta_{\mathrm{TV}}^{-1}))$ and $M = O(C_{\mathrm{act}} T^2 \log(\delta^{-1}))$ (with $C_{\mathrm{act}}$ defined in Lemma B.5). Since $\mathbb{P}(\mathcal{E}) \ge 1 - HT\delta$, replacing $\delta$ with $\delta/(HT)$, and using that $C_{\mathrm{act}} \le L(1+\varepsilon)$ yields the total time complexity being $MTH = \tilde{O}(LT^3 \log(\delta^{-1}) \log(\delta_{\mathrm{TV}}^{-1}))$. $\qquad\square$

## G. Rejection-based Action-level sampling with Metropolis-Hastings correction

**Rejection-sampling MH.** As an alternative to Resampling-pool MH in Section 6, one can use rejection sampling to obtain (conditionally) exact samples from the desired local tilt. However, the acceptance probability of the outer MH step now depends on estimated normalizing constants, and the resulting estimation error enters the analysis in a fundamentally different way. As we can see, while the inner sampling step is exact on a good event, the overall complexity becomes worse in $\delta_{\mathrm{TV}}$.

**Lemma G.1** (Exact MH contraction for Algorithm 8). *Consider Algorithm 8 in the idealized setting where all conditional expectations appearing in the acceptance ratio are computed* exactly *(i.e. no normalization-factor estimation error). Suppose Assumption 3.2 holds with parameter $\varepsilon$. Let $K$ be the independent Metropolis–Hastings kernel on $\mathcal{S}_{1:T}$ used in the outer MH loop, targeting $\tilde{\pi}_T(ds_{1:T} \mid s_0) \propto \pi_{\mathrm{ref}}(ds_{1:T} \mid s_0)\hat{V}(s_0, s_{1:T})$ and proposing from the (naive-guided) trajectory distribution produced by one run of the inner loop. Then for all probability measures $\mu, \nu$ on $\mathcal{S}_{1:T}$ and all integers $m \geq 1$,*

$$\|\mu K^m - \nu K^m\|_{\mathrm{TV}} \leq c^m \|\mu - \nu\|_{\mathrm{TV}}, \qquad c \leq 1 - \frac{1}{(1+\varepsilon)^{2T}}.$$

*Proof.* Let $\pi_{\mathrm{ref}}(\cdot \mid s_0)$ be the base path measure on $\mathcal{S}_{1:T}$ induced by $\pi_{\mathrm{ref}}(\cdot \mid s_0, s_{1:t-1})$. For $t = 1, \ldots, T$, define the exact one-step normalizers

$$Z_t(s_0, s_{1:t-1}) := \mathbb{E}_{s_t \sim \pi_{\mathrm{ref}}(\cdot|s_0, s_{1:t-1})}\big[\hat{V}(s_0, (s_{1:t-1}, s_t))\big], \qquad s_{1:0} := \varnothing.$$

The proposal distribution on full trajectories generated by the inner loop is proportional to $\pi_{\mathrm{ref}}(s_{1:T} \mid s_0) R(s_{1:T})$ with

$$R(s_{1:T}) := \prod_{t=1}^{T} \frac{\hat{V}(s_0, s_{1:t})}{Z_t(s_0, s_{1:t-1})},$$

while the target is proportional to $\pi_{\mathrm{ref}}(s_{1:T} \mid s_0) G(s_{1:T})$ with

$$G(s_{1:T}) := \frac{\hat{V}(s_0, s_{1:T})}{Z_1(s_0)}.$$

Therefore

$$\frac{R(s_{1:T})}{G(s_{1:T})} = \prod_{t=1}^{T-1} \frac{\hat{V}(s_0, s_{1:t})}{Z_{t+1}(s_0, s_{1:t})}, \qquad \frac{G(s_{1:T})}{R(s_{1:T})} = \prod_{t=1}^{T-1} \frac{Z_{t+1}(s_0, s_{1:t})}{\hat{V}(s_0, s_{1:t})}.$$

By Assumption 3.2, for every prefix $s_{1:t}$,

$$\max\left\{ \frac{\hat{V}(s_0, s_{1:t})}{Z_{t+1}(s_0, s_{1:t})}, \frac{Z_{t+1}(s_0, s_{1:t})}{\hat{V}(s_0, s_{1:t})} \right\} \leq 1 + \varepsilon,$$

hence pointwise on $\mathcal{S}_{1:T}$,

$$\max\left\{ \frac{R}{G}, \frac{G}{R} \right\} \leq (1 + \varepsilon)^T.$$

Applying Lemma F.1 with $b = (1+\varepsilon)^T$ yields the TV contraction with $c \leq 1 - 1/b^2 = 1 - (1+\varepsilon)^{-2T}$. $\square$

**Theorem G.2** (Time complexity for Action-level sampling with Metropolis-Hastings). *Under Assumption 3.1 and 3.2, and consider the regime where bellman error $\varepsilon = O(1/T)$. Then for any $0 < \delta \lesssim \delta_{\mathrm{TV}}$, using time $\tilde{O}\left(\frac{LT^3 \log(\delta^{-1})}{\delta_{\mathrm{TV}}^2}\right)$, there exist a good event $\mathcal{E}$ such that $\mathbb{P}(\mathcal{E}) \geq 1 - \delta$, conditioning on which the sampling distribution of Algorithm 8 achieves a TV error less than $\delta_{\mathrm{TV}}$, i.e. $\|\tilde{\pi}_T - \hat{\pi}_T|_{\mathcal{E}}\|_{\mathrm{TV}} \leq \delta_{\mathrm{TV}}$.*

*Proofs of Theorem G.2.* In this proof, we denote $X_h \in \mathcal{S}_{1:T}$ the accepted proposal at the $h$-th step of Metropolis-Hastings. $Y_h \in \mathcal{S}_{1:T}$ is the proposal before rejection/acceptance. The output distribution on an event $\hat{\pi}_T|_{\mathcal{E}}$ is defined as $\mathrm{Law}(X_{H+1})|_{\mathcal{E}}$. Note that the same as in the proof of Proposition D.4, the time contributed by rejection sampling proposal is negligible comparing to estimating the normalizing constant, since the former only requires an estimation up to constant relative error, and the latter, as we will see again in this proof, requires estimation up to $O(\delta_{\mathrm{TV}}/T)$ relative error. We omit the former, and the rate stays unchanged.

Fix $f \in \mathrm{Osc}_1$. Define the exact MH semigroup

$$f_h := K^{H-h+1}f, \qquad h = 1, \dots, H+1,$$

so $f_1 = K^H f$ and $f_{H+1} = f$, and $f_h = K f_{h+1}$. Then $\mathrm{osc}(f_h) \leq 1$.

Let $X_1$ be the first proposal (accepted by default). For each $h = 1, \dots, H$, given current state $X_h$, draw an independent proposal $Y_{h+1}$ and compute the noisy acceptance probability $\hat{\alpha}_h = \hat{\alpha}_h(X_h, Y_{h+1})$; let $\alpha_h$ be the true acceptance probability. Then draw $U_h \sim \mathrm{Unif}[0,1]$ and set

$$X_{h+1} = \begin{cases} Y_{h+1}, & U_h \leq \hat{\alpha}_h, \\ X_h, & U_h > \hat{\alpha}_h. \end{cases}$$

Let $\mathcal{F}_h$ be the sigma-algebra containing all randomness revealed up to *just before* drawing $U_h$ (so it contains $X_h$, $Y_{h+1}$ and all MC randomness used to compute $\hat{\alpha}_h$, but not $U_h$ nor any future randomness).

Write the global good event as $\mathcal{E} = \mathcal{E}_{\leq h} \cap \mathcal{E}_{>h}$ where $\mathcal{E}_{>h} := \cap_{j=h+1}^{H+1} \mathcal{E}_j$. Consider that every normalizing factors is estimated within $1 + \xi$ relative accuracy. Then the accuracy of the normalizing factor is within $\xi_{\mathrm{acc}} = \left( \dfrac{1+\xi}{1-\xi} \right)^T - 1$.

Specifically, Lemma B.5 shows that the event $\mathcal{E}_h := \left\{ |\xi_{\mathrm{acc}}| \leq 16\sqrt{2}T \sqrt{\dfrac{C_{\mathrm{act}} \log(T\delta^{-1})}{N_m}} \right\}$ occurs with probability greater than $1 - \delta$ conditioning on $\mathcal{F}_{h-1}$ when $N_m \geq 4 C_{\mathrm{act}} T^2 \log(T\delta^{-1})$. Thus $\mathbb{P}(\mathcal{E}_h) \geq 1 - \delta$.

We have

$$\mathbb{E}\big[f_{h+1}(X_{h+1})\mathbb{I}_{\mathcal{E}}\big] = \mathbb{E}\big[f_{h+1}(X_{h+1})\big] - \mathbb{E}\big[f_{h+1}(X_{h+1})\mathbb{I}_{\mathcal{E}^c}\big].$$

For each $h = 1, \dots, H$, define the auxiliary "true-accept" next state

$$\widetilde{X}_{h+1} = \begin{cases} Y_{h+1}, & U_h \leq \alpha_h, \\ X_h, & U_h > \alpha_h. \end{cases}$$

Condition on $\mathcal{F}_h$ (so $U_h$ is the only remaining randomness). Then

$$\mathbb{E}_{U_h}\big[f_{h+1}(X_{h+1}) \mid \mathcal{F}_h\big] = \hat{\alpha}_h f_{h+1}(Y_{h+1}) + (1 - \hat{\alpha}_h)f_{h+1}(X_h),$$

and similarly

$$\mathbb{E}_{U_h}\big[f_{h+1}(\widetilde{X}_{h+1}) \mid \mathcal{F}_h\big] = \alpha_h f_{h+1}(Y_{h+1}) + (1 - \alpha_h)f_{h+1}(X_h),$$

and

$$\mathbb{E}[\mathbb{E}_{U_h}\big[f_{h+1}(\widetilde{X}_{h+1}) \mid \mathcal{F}_h\big]] = \mathbb{E}[K f_{h+1}(X_h)] = \mathbb{E}[f_h(X_h)].$$

We also have

$$\mathbb{E}[\mathbb{E}_{U_h}\big[f_{h+1}(X_{h+1}) \mid \mathcal{F}_h\big]] = \mathbb{E}[f_{h+1}(X_{h+1})].$$

Hence the exact identity

$$\mathbb{E}_{U_h}\big[f_{h+1}(X_{h+1}) \mid \mathcal{F}_h\big] - \mathbb{E}_{U_h}\big[f_{h+1}(\widetilde{X}_{h+1}) \mid \mathcal{F}_h\big] = (\hat{\alpha}_h - \alpha_h)\big(f_{h+1}(Y_{h+1}) - f_{h+1}(X_h)\big). \tag{44}$$

We are only interested in the difference under the good event $\mathcal{E}_{\leq h}$, in which $|\hat{\alpha}_h - \alpha_h|$ is small, so we also have

$$\left| \mathbb{E}\left[\mathbb{E}_{U_h}\big[f_{h+1}(X_{h+1}) \mid \mathcal{F}_h\big] - \mathbb{E}_{U_h}\big[f_{h+1}(\widetilde{X}_{h+1}) \mid \mathcal{F}_h\big]\right]\right|$$

$$\leq \left| \mathbb{E}\left[\mathbb{I}_{\mathcal{E}_h}\left(\mathbb{E}_{U_h}\big[f_{h+1}(X_{h+1}) \mid \mathcal{F}_h\big] - \mathbb{E}_{U_h}\big[f_{h+1}(\widetilde{X}_{h+1}) \mid \mathcal{F}_h\big]\right)\right]\right|$$

$$+ \left| \mathbb{E}\left[\mathbb{I}_{\mathcal{E}_h^c}\left(\mathbb{E}_{U_h}\big[f_{h+1}(X_{h+1}) \mid \mathcal{F}_h\big] - \mathbb{E}_{U_h}\big[f_{h+1}(\widetilde{X}_{h+1}) \mid \mathcal{F}_h\big]\right)\right]\right|$$

$$\leq \sup_{\mathcal{E}_h} |(\hat{\alpha}_h - \alpha_h)||\big(f_{h+1}(Y_{h+1}) - f_{h+1}(X_h)\big)|\mathbb{P}(\mathcal{E}_h) + \mathrm{osc}(f_{h+1})\mathbb{P}(\mathcal{E}_h^c).$$

Now we preform the telescoping

$$\mathbb{E}\big[f(X_{H+1})\big] - \mathbb{E}\big[K^H(f)(X_1)\big] = \sum_{h=1}^{H} \mathbb{E}\left[\mathbb{E}_{U_h}\big[f_{h+1}(X_{h+1}) \mid \mathcal{F}_h\big] - \mathbb{E}_{U_h}\big[f_{h+1}(\widetilde{X}_{h+1}) \mid \mathcal{F}_h\big]\right].$$

Thus

$$\mathbb{E}\big[f(X_{H+1})\mathbb{I}_{\mathcal{E}}\big]/\mathbb{P}(\mathcal{E}) - \mathbb{E}\big[K^H(f)(X_1)\big]$$
$$= (\mathbb{E}\big[f(X_{H+1})\big] - \mathbb{E}\big[K^H(f)(X_1)\big])/\mathbb{P}(\mathcal{E}) - (\mathbb{E}\big[f(X_{H+1})\mathbb{I}_{\mathcal{E}^c}\big] - (1 - \mathbb{P}(\mathcal{E}))\mathbb{E}\big[K^H(f)(X_1)\big])/\mathbb{P}(\mathcal{E})$$
$$= \frac{1}{\mathbb{P}(\mathcal{E})}\sum_{h=1}^{H}\mathbb{E}\left[\mathbb{E}_{U_h}\big[f_{h+1}(X_{h+1}) \mid \mathcal{F}_h\big] - \mathbb{E}_{U_h}\big[f_{h+1}(\widetilde{X}_{h+1}) \mid \mathcal{F}_h\big]\right] - \frac{\mathbb{P}(\mathcal{E}^c)}{\mathbb{P}(\mathcal{E})}(\mathbb{E}\big[f(X_{H+1})\mathbb{I}_{\mathcal{E}^c}\big]/\mathbb{P}(\mathcal{E}^c) - \mathbb{E}\big[K^H(f)(X_1)\big]).$$

so

$$\big|\mathbb{E}\big[f(X_{H+1})\mathbb{I}_{\mathcal{E}}\big]/\mathbb{P}(\mathcal{E}) - \mathbb{E}\big[K^H(f)(X_1)\big]\big| \leq \sum_{h=1}^{H}(2H\delta + \xi_{\mathrm{acc}})\mathrm{osc}(f_{h+1}) + 2H\delta\,\mathrm{osc}(f) \leq (2H(H+1)\delta + H\xi_{\mathrm{acc}}).$$

where we used the fact that $|\hat{\alpha}_h - \alpha_h| \leq \xi_{\mathrm{acc}}$ on the good event $\mathcal{E}_h$.

Therefore we have

$$\|\mathrm{Law}(X_{H+1})|_{\mathcal{E}} - \tilde{\pi}_T\|_{\mathrm{TV}} \leq \|\mathrm{Law}(X_{H+1})|_{\mathcal{E}} - \mathrm{Law}(X_1)K^H\|_{\mathrm{TV}} + \|\mathrm{Law}(X_1)K^H - \tilde{\pi}_T\|_{\mathrm{TV}}$$
$$\leq 2H(H+1)\delta + H\xi_{\mathrm{acc}} + c^H\|\mathrm{Law}(X_1) - \tilde{\pi}_T\|_{\mathrm{TV}}$$

where by Lemma G.1, the exact MH kernel satisfies the uniform TV contraction with $c \leq 1 - \dfrac{1}{(1+\varepsilon)^{2T}}$.

Recall that Theorem 4.3 shows that $\|\mathrm{Law}(X_1) - \tilde{\pi}_T\|_{\mathrm{TV}} \leq 2\varepsilon T$. We focus on the $\varepsilon = O(1/T)$ regime, specifically when $\varepsilon < \dfrac{1}{5T}$, thus $c^H \leq (4\varepsilon T)^H$, and

$$\|\mathrm{Law}(X_{H+1})|_{\mathcal{E}} - \tilde{\pi}_T\|_{\mathrm{TV}} \leq 2H(H+1)\delta + H\xi_{\mathrm{acc}} + (4\varepsilon T)^H \cdot 2\varepsilon T.$$

By picking $\dfrac{1}{2}\delta_{TV} = (4\varepsilon T)^H \cdot 2\varepsilon T$, we have $H = \dfrac{\log(\frac{\delta_{\mathrm{TV}}}{4\varepsilon T})}{\log(4\varepsilon T)} = O(\log(\delta_{\mathrm{TV}}^{-1}))$. Then choose $\delta \leq \dfrac{\delta_{\mathrm{TV}}}{8H(H+1)} = O(\delta_{\mathrm{TV}})$, and $\xi_{\mathrm{acc}} = \delta_{\mathrm{TV}}(4H)^{-1}$, i.e. $N_m = \tilde{O}(\dfrac{T^2 C_{\mathrm{act}}\log(\delta^{-1})}{\delta_{\mathrm{TV}}^2})$. Substituting $\delta \leftarrow \dfrac{\delta}{H}$ doesn't change the rate. Then within event $\mathcal{E}$, the time complexity is $O(N_m T H) = \tilde{O}(\dfrac{T^3 L \log(\delta^{-1})}{\delta_{\mathrm{TV}}^2})$. $\qquad\square$

*Remark* G.3. **Why it is worse than resampling-pool (Section 6).** In this construction, the MH acceptance probability depends on estimated normalizing constants produced by Monte Carlo. Consequently, the estimation error no longer only affects the (constant-order) contraction coefficient; instead it induces an additional *additive* error at each MH step, which accumulates *linearly* with the number of MH iterations. To make the final TV error below $\delta_{\mathrm{TV}}$, we must therefore estimate the relevant normalizers to relative error $O(\delta_{\mathrm{TV}}/T)$, leading to an unavoidable $\delta_{\mathrm{TV}}^{-2}$ overhead.

## H. New assumption: Global bellman error

**Assumption H.1** (Global Bellman error bound)**.** We assume for some $\varepsilon_g$, the following holds:

$$\max_{t\in[T]} \max_{s_{1:t}\in\mathcal{S}_{1:t}} \max\left\{ \frac{\hat{V}(s_0, s_{1:t})}{\mathbb{E}_{s_{t+1:T}\sim\pi_{\text{ref}}(\cdot|s_0,s_{1:t})}[\hat{V}(s_0, s_{1:T})]}, \right.$$
$$\left. \frac{\mathbb{E}_{s_{t+1:T}\sim\pi_{\text{ref}}(\cdot|s_0,s_{1:t})}[\hat{V}(s_0, s_{1:T})]}{\hat{V}(s_0, s_{1:t})} \right\} \leq 1 + \varepsilon_g$$

Under this assumption, we have a new complexity bound for SMC with naive proposal characterized by the global bellman error rather than our previous local bellman error. We also show later that the result hold similarly for any proposal $\pi_{\text{p}}$.

**Proposition H.2.** *Under the same condition as Theorem 5.1, with Assumption 3.2 replaced by Assumption H.1, the number of particles suffice to achieve a sampling TV error below $\delta_{\text{TV}}$ is $N \geq \dfrac{L^6 \, T \, (1+\varepsilon_g)^6}{2 \, \delta_{\text{TV}}}$. And the total time complexity is* $O\left(\dfrac{L^6(1+\varepsilon_g)^6 T^2}{\delta_{\text{TV}}}\right).$

*Proof of Proposition H.2.* We follow exactly the proof strategy of Theorem 5.1, starting from the local TV-bias bound (Theorem E.6):

$$\|\bar{\eta}^N_{T+1} - \eta_{T+1}\|_{\text{TV}} \leq \frac{1}{4N}\sum_{p=1}^{T+1}(q^2_{p,T+1}-1)\,\beta(D\Phi_{p,T+1}). \tag{45}$$

By Lemma E.1, we have $\beta(D\Phi_{p,T+1}) \leq 2q_{p,T+1}\beta(P_{p,T+1}) \leq 2q_{p,T+1}$, and hence

$$\|\bar{\eta}^N_{T+1} - \eta_{T+1}\|_{\text{TV}} \leq \frac{1}{2N}\sum_{p=1}^{T+1} q_{p,T+1}\,(q^2_{p,T+1}-1). \tag{46}$$

The last term $p = T + 1$ vanishes since $q_{T+1,T+1} = 1$.

**Step 1: A uniform bound on $q_{p,T+1}$ under global Bellman error.** We consider the naive proposal, where $M_t(s_{1:t-1}, ds_{1:t}) = \pi_{\text{ref}}(s_t \mid s_0, s_{1:t-1})$ and $G_t(s_{1:t}) = \hat{V}(s_0, s_{1:t})/\hat{V}(s_0, s_{1:t-1})$. Recall the FK semigroup interpretation

$$Q_{p,T+1}(1)(x) = \mathbb{E}\left[\prod_{t=p}^{T} G_t(S_{1:t}) \,\Big|\, S_{1:p} = x\right].$$

Because the $G_t$'s are telescoping ratios, the product cancels all cross terms:

$$\prod_{t=p}^{T} G_t(S_{1:t}) = \frac{\hat{V}(s_0, S_{1:T})}{\hat{V}(s_0, S_{1:p-1})}.$$

Therefore, for any prefix/state $x = s_{1:p} \in E_p$,

$$Q_{p,T+1}(1)(x) = \frac{\mathbb{E}\left[\hat{V}(s_0, S_{1:T}) \mid S_{1:p} = x\right]}{\hat{V}(s_0, s_{1:p-1})}. \tag{47}$$

Now apply Assumption H.1 at time $t = p$:

$$\frac{1}{1+\varepsilon_g}\,\hat{V}(s_0, s_{1:p}) \;\leq\; \mathbb{E}\left[\hat{V}(s_0, S_{1:T}) \mid S_{1:p} = s_{1:p}\right] \;\leq\; (1+\varepsilon_g)\,\hat{V}(s_0, s_{1:p}).$$

Plugging into (47) yields

$$\frac{1}{1+\varepsilon_g} \cdot \frac{\hat{V}(s_0, s_{1:p})}{\hat{V}(s_0, s_{1:p-1})} \;\leq\; Q_{p,T+1}(1)(s_{1:p}) \;\leq\; (1+\varepsilon_g) \cdot \frac{\hat{V}(s_0, s_{1:p})}{\hat{V}(s_0, s_{1:p-1})}.$$

By Assumption 3.1, $\hat{V}(s_0, s_{1:p})/\hat{V}(s_0, s_{1:p-1}) \in [1/L, \ L]$. Hence for all $p \leq T$ and all $x \in E_p$,

$$\frac{1}{L(1 + \varepsilon_g)} \ \leq \ Q_{p,T+1}(1)(x) \ \leq \ L(1 + \varepsilon_g).$$

Consequently,

$$q_{p,T+1} = \sup_{x,y \in E_p} \frac{Q_{p,T+1}(1)(x)}{Q_{p,T+1}(1)(y)} \ \leq \ L^2(1 + \varepsilon_g)^2, \qquad \forall p \leq T. \tag{48}$$

**Step 2: Plug into the local TV-bias bound.** Using $q(q^2 - 1) \leq q^3$ and (48), we obtain for all $p \leq T$:

$$q_{p,T+1} \left( q_{p,T+1}^2 - 1 \right) \ \leq \ q_{p,T+1}^3 \ \leq \ L^6 (1 + \varepsilon_g)^6.$$

Plugging into (46) gives

$$\|\bar{\eta}_{T+1}^N - \eta_{T+1}\|_{\mathrm{TV}} \leq \frac{1}{2N} \sum_{p=1}^{T} L^6(1 + \varepsilon_g)^6 = \frac{L^6 \, T \, (1 + \varepsilon_g)^6}{2N}.$$

Therefore, it suffices to take $N \geq \dfrac{L^6 \, T \, (1 + \varepsilon_g)^6}{2 \, \delta_{\mathrm{TV}}}$ to ensure $\|\bar{\eta}_{T+1}^N - \eta_{T+1}\|_{\mathrm{TV}} \leq \delta_{\mathrm{TV}}$.

Finally, since the naive-proposal SMC runs $T$ propagation steps (each costing $O(N)$) plus the terminal reweight/resampling step, the total time is $O(NT)$; substituting the above choice of $N$ yields $O\left( \dfrac{L^6(1 + \varepsilon_g)^6 T^2}{\delta_{\mathrm{TV}}} \right)$. $\qquad\square$

These bounds do not have $T$ on the power, thus there's no need to restrict $\varepsilon_g$ to the $O(1/T)$ regime to avoid exponential complexity. And we also claim that this does not violate our previous lower bound, since the lower bound were proved under the assumption of the local bellman error. This is exactly the case for (Rohatgi et al., 2025) as well, which adopts the exact same assumption as Assumption H.1.

Assumption H.1 and Assumption 3.2 can transit to one another in the sense that, if Assumption H.1 holds with $1 + \varepsilon_g$, then Assumption 3.2 holds with $(1 + \varepsilon) = (1 + \varepsilon_g)^2$. Conversely, if Assumption 3.2 holds with $(1 + \varepsilon)$, then Assumption H.1 holds with $(1 + \varepsilon_g) = (1 + \varepsilon)^T$.

A similar result holds for the particle complexity of any proposal $\pi_{\mathrm{p}}$, including the optimal proposal. We have the following result

**Proposition H.3** (Particles Complexity of arbitrary proposal). *Fix the horizon $T \geq 2$ and a target accuracy $\delta_{\mathrm{TV}} \in (0, 1)$. Under Assumptions 3.1 and H.1, let $\bar{\eta}_{T+1}^N := \mathbb{E}[\eta_{T+1}^N]$ be the output distribution of an SMC sampler (after the terminal resampling step $T+1$ with $M_{T+1} = \mathrm{Id}$). We further suppose*

$$\max_{t \in [T]} \max_{s_{1:t} \in \mathcal{S}_{1:t}} \max \left\{ \frac{\pi_{\mathrm{ref}}(s_t \mid s_{1:t-1})}{\pi_{\mathrm{p}}(s_t \mid s_{1:t-1})} \cdot \frac{\hat{V}(s_0, s_{1:t})}{\hat{V}(s_0, s_{1:t-1})}, \frac{\pi_{\mathrm{p}}(s_t \mid s_{1:t-1})}{\pi_{\mathrm{ref}}(s_t \mid s_{1:t-1})} \cdot \frac{\hat{V}(s_0, s_{1:t-1})}{\hat{V}(s_0, s_{1:t})} \right\} \leq L_{\mathrm{p}}.$$

*Then a sufficient condition that guarantees $\|\bar{\eta}_{T+1}^N - \eta_{T+1}\|_{\mathrm{TV}} \ \leq \ \delta_{\mathrm{TV}}$ is*

$$N \geq \frac{L_{\mathrm{p}}^6 T (1 + \varepsilon_g)^6}{2 \delta_{\mathrm{TV}}}.$$

*Specifically, if $\pi_{\mathrm{p}}$ is the optimal proposal, we also have*

$$N \ \geq \ \frac{\left( (1 + \varepsilon_g)^{12} - 1 \right) T (1 + \varepsilon_g)^6}{2 \, \delta_{\mathrm{TV}}}$$

*is a sufficient condition for $\|\bar{\eta}_{T+1}^N - \eta_{T+1}\|_{\mathrm{TV}} \ \leq \ \delta_{\mathrm{TV}}$.*

*Proof.* The proof was similar to that of Theorem 5.1, Theorem D.1 and Proposition H.2. $\qquad\square$

