# OpenReview forum: "On the Power of (Approximate) Reward Models for Inference-Time Scaling: Sequential Monte Carlo and Beyond"
_ICML.cc/2026/Conference — ICML 2026 regular_

### Official Review · Reviewer_qqaE · 2026-03-09

**Soundness:** 3
**Presentation:** 2
**Significance:** 3
**Originality:** 4
**Overall Recommendation:** 4
**Confidence:** 3

**Summary:**

Expanding reasoning has become an effective method for enhancing the reasoning capabilities of large language models. Among them, the Sequential Monte Carlo (SMC) framework is particularly important. It rewards the model for evaluating intermediate results and dynamically allocates computing resources. However, in practice, it is impossible to obtain a real reward model; only approximate models can be used. Regarding the question of why and when approximate reward models can effectively support reasoning expansion, this article theoretically answers this question. It determines the Bellman error of the approximate reward model as the core indicator and, combined with SMC, reduces the complexity of reasoning from exponential level to polynomial level, thereby achieving significant efficiency improvement.

**Compliance With Llm Reviewing Policy:**

Affirmed.

**Final Justification:**

Thanks to the author's detailed response, I still maintain my previous results.

**Key Questions For Authors:**

(1) In your paper, you model the reasoning process as a Feynman-Kac (FK) flow and conduct a unified analysis using the Sequential Monte Carlo (SMC) method. This is a very elegant and powerful perspective. However, for readers who are not familiar with particle filtering, understanding how the potential function and the Markov kernel in the FK model specifically correspond to the reward model in the language model and the sampling of the next token becomes somewhat abstract. Could you demonstrate this mapping process step by step using a specific and simplified example of language model reasoning?
(2) Regarding this wealth of theoretical knowledge, can it be explained through simple experiments, thereby further verifying the credibility and practicality of the theory?
(3) In Section 6 and Appendix G, two correction methods, "resampling pool MH" and "rejection sampling MH," are compared, and it is pointed out that they have fundamental differences in their theoretical reliance on the total variational error. In practice, for typical LLM inference tasks, which method do you expect to have an advantage?

**Limitations:**

The authors have not adequately discussed the limitations and potential negative societal impact of their work. The paper contains only a brief "Impact Statement" that dismisses the need for discussion, stating that the work is theoretical and has no immediate consequences beyond those already known for machine learning research. Please include the limitations of the model in the outlook, as well as the subsequent improvement plans.

**Strengths And Weaknesses:**

Soundness: The theoretical derivation of the paper is very profound and complex, extensively drawing on the classic theories of the Feynman-Kac model and particle filtering. Judging from the provided appendix content, the proof process is rigorous, and the logical chain between lemmas and theorems is complete. The author successfully mapped the language model inference problem into a mature probabilistic framework and conducted innovative analysis based on this. Using the Feynman-Kac model to describe the self-regressive generation process guided by the reward model is a very appropriate and ingenious choice. However, no experimental verification is provided throughout the paper. For a paper claiming to explain and guide practice, it has certain deficiencies.

Presentation: The overall structure of the paper is clear. The author endeavors to reinterpret the reasoning problem using the language of Feynman-Kac and SMC and clearly states the objective at the beginning of each section. Moreover, the idea of unifying the reasoning algorithm under the SMC framework is the core highlight of the paper and has been well elaborated. However, the writing of the paper is extremely obscure. Many key steps are buried in the lengthy proofs in the appendix, while the narrative in the main text is overly concise and lacks intuitive explanations and motivations for the derivations.

Significance: Expansion during reasoning is one of the most popular and important directions in current LLM research. Understanding the underlying principles, especially how the approximate reward model affects performance, is of great value for guiding algorithm design and resource allocation. Therefore, the problem studied in the paper is extremely important and relevant. The paper provides a very profound theoretical perspective, unifying a series of heuristic algorithms such as SMC under a rigorous mathematical framework, deepening the understanding of these algorithms in the theoretical community. However, the paper lacks experimental verification, resulting in the conclusion remaining on a theoretical level.

Originality: The language model inference problem is systematically and rigorously modeled as a Feynman-Kac flow, and analyzed using the SMC theory. This is a highly original contribution. Although SMC has previously been used for language generation, this paper is pioneering in its in-depth theoretical exploration of its relationship with the quality of the reward model, the lower bound, and computational complexity.

---

> ### Author Rebuttal · Authors · 2026-03-31
>
> We thank the reviewer for the encouraging feedback and for appreciating the Feynman--Kac perspective of the paper.
>
> > On making the FK/SMC mapping more concrete
>
> We thank the reviewer for this helpful suggestion. We agree that, without a concrete illustration, the FK notation can feel abstract for readers who are less familiar with SMC. In the revised paper, we added a simple three-step arithmetic reasoning example, together with a schematic reasoning-tree figure, to make this mapping explicit. In this example, the state at time $t$ is the current reasoning prefix, the Markov kernel $M_t$ samples the next reasoning step from the language model and appends it to the prefix, and the potential $G_t$ reweights the expanded prefix according to the incremental change in the reward/value model. Under the naive proposal, $G_t$ reduces to $\hat V(s_0,s_{1:t}) / \hat V(s_0,s_{1:t-1})$, so the cumulative weights telescope along a full trajectory and recover the reward-tilted target distribution. Concretely, in the example, $M_t$ corresponds to “generate the next reasoning sentence,” while $G_t$ corresponds to “reweight the resulting prefix based on how promising it now appears.”
>
> > On experimental illustration
>
> We also agree that the theory would be much more compelling with a simple experiment. Thus, we added a synthetic setup with a known ground-truth reward-to-go, in which we can directly control the Bellman error and compare the TV error of SMC as this error varies.
>
> >>**Setting.** We use a 0-1 state depth-$T$ Bernoulli tree with $\pi_{ref}={\rm Bernoulli}(1/2)$ and terminal reward
> $$
> \phi(s_{1:T})=\exp(\beta\sum_{t=1}^Ts_t).
> $$
> The reward-to-go is available in closed form, so we use it as ground truth and perturb it to control the Bellman error. One main table is below (entries: TV error):
> |N\ε|0.0|2.0|4.0|6.0|8.0|10.0|
> |---|---:|---:|---:|---:|---:|---:|
> |16|0.308|0.355|0.399|0.448|0.526|0.583|
> |32|0.229|0.263|0.314|0.355|0.406|0.456|
> |64|0.167|0.196|0.232|0.268|0.309|0.338|
> |128|0.120|0.141|0.166|0.195|0.226|0.246|
> |256|0.086|0.102|0.123|0.144|0.162|0.178|
> |512|0.062|0.072|0.087|0.103|0.114|0.128|
>
> This experiment helps make the paper’s main qualitative point concrete: the quality of the Bellman approximation is what primarily determines when SMC-based inference-time scaling works well.
>
> > On the comparison between the two correction methods
>
> Regarding the two correction methods in Section 6 / Appendix G, our current theoretical picture suggests that the **resampling-pool MH** construction is the more attractive one when exactness of the acceptance ratio matters, because it computes the proposal probability exactly on an augmented space and therefore retains geometric contraction in TV. By contrast, the rejection-based correction generally accumulates proposal / acceptance estimation error and can lose this favorable dependence. In particular, in this method the error is dominated by estimating normalizing factors, which only converges at the canonical Monte Carlo rate $1/\sqrt{N}$ rather than exponentially fast in $N$. We made this practical takeaway more explicit in the revision.
>
> > On presentation and intuition in the main text
>
> We also appreciate the reviewer’s comments on presentation and agree that the exposition was too terse in places, with too much of the intuition deferred to the appendix. In the revision, we made a more substantial effort to improve accessibility in the main text. In particular, we expanded the proof sketches around the main results, added higher-level intuition before the technical arguments, and more explicitly explained why the key quantities in the proofs (such as Bellman error, semigroup stability, and reweighting error accumulation) are the ones that matter for practice. Our goal was that readers can first understand the mechanism at a conceptual level—why approximate reward-to-go information changes the effective difficulty of the SMC problem, and where the complexity improvement comes from—before reading the full details of the proofs. We revised the main text accordingly to better connect the technical derivations with their practical interpretation.
>
> > On limitations
>
> Finally, we agree that the current discussion of limitations is too brief. We expanded it accordingly, with a detailed discussion on future research. On the limitations side, we now make clear that the current theory is presented in a worst-case form, while also noting that average-case relaxations are possible and would not alter the main proof idea. On the future-work side, we highlight several concrete directions, including the design of improved **McKean kernels and larger-scale empirical evaluations** of the algorithms considered here.
>
> Overall, we are grateful for the reviewer’s thoughtful comments. We hope the clarifications above, the revisions and additions we made, address the main concerns. We respectfully ask the reviewer to reconsider their score in light of these changes.

---

> > ### Author Rebuttal · Reviewer_qqaE · 2026-04-01
> >
> > The author further resolved my doubts through experimental data.

---

### Official Review · Reviewer_M9Tf · 2026-03-11

**Soundness:** 3
**Presentation:** 3
**Significance:** 3
**Originality:** 3
**Overall Recommendation:** 5
**Confidence:** 3

**Summary:**

The work aims to quantify the conditions under which process reward models can help achieve polynomial-time test-time scaling. Under a set of assumptions, the authors first derive algorithm-agnostic lower bounds on the time complexity required to approximate the reward-tilted distribution. They then specifically look at sequential monte carlo (SMC) methods and derive upper bounds on the particle complexity and runtime. Finally, a connection is drawn to chain-based methods.

**Compliance With Llm Reviewing Policy:**

Affirmed.

**Final Justification:**

My main concerns about the paper were about assumptions, limitations, and practical takeaways. I believe they have been adequately addressed to the extent that they can be done in a rebuttal window. This has changed my evaluation from a 4 to a 5, as I believe it is a novel theoretical contribution.

**Key Questions For Authors:**

1. Could the authors elaborate on the implications of assumptions 3.1 and 3.2?
2. How can the analysis presented in the paper be utilized? Can it help us in algorithm selection or reward model training?

**Limitations:**

The main paper lacks a section on limitations and future work.

**Strengths And Weaknesses:**

**Strengths:**
1. The lower bounds derived in Section 4 are algorithm-agnostic, and identify a transition from polynomial to exponential-time based on the Bellman error bound.
2. The paper increases our theoretical understanding of test-time alignment methods and appears sound in its derivations.


**Weaknesses:**
1. Assumption 3.1, which is central to the paper, also limits its applicability–it enforces strict positivity of the potential function, effectively requiring the aligned distribution to share support with the base model distribution. This is a stronger assumption than found in prevailing literature [1, 2, 3, 4], and does not admit hard constraints.
2. The paper is descriptive about its theoretical results, but does not clearly articulate practical takeaways.


[1] Syntactic and Semantic Control of Large Language Models via Sequential Monte Carlo \
[2] Taming Imperfect Process Verifiers: A Sampling Perspective on Backtracking \
[3] Fast Controlled Generation from Language Models with Adaptive Weighted Rejection Sampling \
[4] On the Query Complexity of Verifier-Assisted Language Generation

---

> ### Author Rebuttal · Authors · 2026-03-31
>
> We thank the reviewer for the positive assessment and for raising the important questions about the implications of Assumptions 3.1 and 3.2, as well as the practical use of the analysis.
>
> >On the implications and strengths of Assumptions 3.1 and 3.2
>
> We agree that the practical meaning of these assumptions should have been explained more clearly, and we have revised the discussion accordingly.
>
> For **Assumption 3.1**, the key intuition is that what is really needed is some form of **coverage**. If the base policy assigns low probability to certain regions, then sampling-based inference-time algorithms will have difficulty discovering what happens there. From this perspective, some coverage-type condition is unavoidable.
>
> Our current worst-case assumption is best viewed as the cleanest and simplest way to formalize this idea in an end-to-end theorem. The tradeoff is that it rules out **zero-support hard constraints**, while still allowing **low-support / highly concentrated** targets. In other words, the assumption does **not** prevent the aligned distribution from putting most of its mass on only a few answers.
>
> We agree that this worst-case form is stronger than necessary. It is mainly a convenient way to state a uniform upper and lower bound. A more natural weakening is an **average-case** notion of coverage, and would be much closer in spirit to action-level coverage assumptions such as $C_{\mathrm{act}}$ in [2], and our view is that this mainly changes the moment-control part of the proof rather than the main proof backbone. We have made this connection clearer in the revision.
>
> For **Assumption 3.2**, the intended interpretation is that the reward model is something to be **learned**, and the theory is identifying how accurate it must be as the horizon grows. The point of the theorem is to say that if one wants polynomial-time inference scaling, then the learned prefix-level reward-to-go must be accurate at roughly this level. As with Assumption 3.1, this condition is currently stated in a worst-case uniform form for clarity, but it can also be weakened to average-case or suitable moment conditions rather than worst-case control, but the essence remains intact.
>
> >On how the analysis can be utilized
>
> We agree that the practical takeaways should have been stated more sharply, and we have added a clearer discussion.
>
> First, the analysis gives guidance for **reward-model training**: the relevant quantity is not only terminal ranking quality, but also whether the model provides a sufficiently accurate **Bellman-consistent signal**.
>
> Second, it is useful for **algorithm selection**. The theory explains why terminal-only selection methods should generally retain exponential dependence, while methods that use accurate intermediate reward-to-go information can enter the polynomial regime.
>
> Third, the paper highlights a practical tradeoff between **particle count** and **proposal / kernel design**, and also clarifies when **one-particle guided methods with an additional correction mechanism** can elaborate different guarantees, potentially outperforming naive SMCs. This provides useful guidance to practical applications, including designing McKean kernels and using Metropolis Hastings correction instead of scaling up the number of particles.
>
> >On limitations and future work
>
> We agree with the reviewer that the main paper lacked a clear **limitations and future work** section, and we have added one in the revision. On the limitations side, we now state explicitly that the current theory is developed in a worst-case form, but will note that average-form weakening is possible and does not change the essence of the proof. On the future-work side, we highlight several concrete directions, including designing better **McKean kernels / proposal mechanisms**, carrying out **larger-scale empirical comparisons** among the algorithms discussed in the paper, and further improving the dependence of the complexity bounds through sharper analyses of proposal and resampling design.
>
> Overall, we appreciate the reviewer’s comments. We believe the revised discussion now makes both the role of the assumptions and the practical implications of the theory much clearer.
>
> [2] Taming Imperfect Process Verifiers: A Sampling Perspective on Backtracking

---

> > ### Author Rebuttal · Reviewer_M9Tf · 2026-04-02
> >
> > Thank you for addressing my concerns. I will raise my score to a 5.

---

### Official Review · Reviewer_7ePz · 2026-03-11

**Soundness:** 2
**Presentation:** 2
**Significance:** 2
**Originality:** 2
**Overall Recommendation:** 4
**Confidence:** 2

**Summary:**

This paper provides a theoretical analysis of inference-time scaling using Sequential Monte Carlo (SMC) methods with approximate reward models. The main result establishes that when the Bellman error of an approximate reward model is bounded by O(1/T) (where T is the reasoning horizon), SMC-based sampling can reduce the computational complexity from exponential in T to polynomial in T. The paper also proves matching (up to constants in the exponent) lower bounds showing that without such Bellman error control, exponential complexity is unavoidable.

**Compliance With Llm Reviewing Policy:**

Affirmed.

**Final Justification:**

The rebuttal addressed my doubts, but I am only convinced to raise the score to a weak accept

**Key Questions For Authors:**

1. Can you provide any evidence theoretical or empirical that existing reward model training procedures (RLHF, DPO, process reward models) produce models with Bellman error O(1/T)?
2. How does your framework handle the common practical setting where reward models are trained on outcome-level labels only (not process-level)?
3. Could you comment on how best-of-N sampling fits into your complexity landscape? It's the most common baseline in practice.

**Limitations:**

Yes

**Strengths And Weaknesses:**

Strengths:

1.  The formalization via Feynman-Kac flows and the identification of Bellman error as the governing quantity is elegant. The connection between the SMC literature and LLM inference-time computation is well-articulated.
2. The paper provides both complexity lower bounds (showing exponential scaling is inherent without good reward models) and upper bounds (showing polynomial scaling is achievable with Bellman error control). This gives a relatively complete picture of the complexity landscape.

Weaknesses:

1. The paper makes strong claims about what should happen in practice (e.g., polynomial complexity with bounded Bellman error), but provides zero empirical evidence. Even a simple synthetic experiment say, on a toy autoregressive model with a known ground-truth reward would dramatically strengthen the paper.
2. The paper essentially says if the reward model is good in the specific sense, then SMC works, but gives no guidance on whether real reward models are good in this sense. This is a critical missing piece.
3. Remark 3.4 acknowledges that the theory assumes exact policy evaluation at the sentence level, while practical LLM inference operates at the token level. At the token level, computing exact likelihoods under is straightforward, but reward models typically operate at coarser granularity. The paper doesn't reconcile this properly.

---

> ### Author Rebuttal · Authors · 2026-03-31
>
> We thank the reviewer for the thoughtful questions and for highlighting several important practical aspects.
>
> >On empirical evidence
>
> We agree that empirical support strengthens the paper. In response, **we have added a simple synthetic experiment** that evaluates how SMC TV accuracy changes as the Bellman error increases.
>
> >>**Setting.** We use a 0-1 state depth-$T$ Bernoulli tree with $\pi_{ref}={\rm Bernoulli}(1/2)$ and terminal reward
> $$
> \phi(s_{1:T})=\exp(\beta\sum_{t=1}^Ts_t).
> $$
> The reward-to-go is available in closed form, so we use it as ground truth and perturb it to control the Bellman error.
> One main table is below (entries: TV error):
> |N\ε|0.0|2.0|4.0|6.0|8.0|10.0|
> |---|---:|---:|---:|---:|---:|---:|
> |16|0.308|0.355|0.399|0.448|0.526|0.583|
> |32|0.229|0.263|0.314|0.355|0.406|0.456|
> |64|0.167|0.196|0.232|0.268|0.309|0.338|
> |128|0.120|0.141|0.166|0.195|0.226|0.246|
> |256|0.086|0.102|0.123|0.144|0.162|0.178|
> |512|0.062|0.072|0.087|0.103|0.114|0.128|
>
> This successfully illustrates the main qualitative message of the paper: **Bellman error is the key quantity governing the effectiveness of SMC-based inference-time scaling.**
>
> >On whether existing reward-model achieve Bellman error $O(1/T)$
>
> Our point is not that current RLHF/DPO/PRM systems are already known to satisfy a Bellman error bound of order $O(1/T)$. Rather, the theorem identifies a **necessary accuracy regime** for efficient inference-time scaling: without sufficiently small Bellman error, exponential dependence on the reasoning horizon cannot in general be avoided. At the same time, in realizable settings, RL / value-learning style objectives can in principle drive approximation error arbitrarily low given sufficient data, model capacity, and optimization, so the theorem should be read as identifying the **precision target** that training needs to achieve.
>
> This also gives a practical interpretation. The condition $ε=O(1/T)$ means that longer reasoning requires a proportionally more accurate reward model. Equivalently, a reward model with Bellman error $ε$ is only expected to support effective inference-time scaling up to horizons on the order of $1/ε$. Thus, in realistic systems, the main issue is often that the constant may be large, not that the condition is meaningless: a better reward model supports longer useful reasoning before error accumulation dominates. We emphasized this idea in our revision.
>
> We also emphasize that, as discussed in our paper, this $O(1/T)$ threshold is specific to our **local Bellman-error** analysis: there, polynomial dependence on $T$ requires $\varepsilon = O(1/T)$. Under the **global Bellman-error** condition, by contrast, the complexity is polynomial in both $T$ and $ε$ throughout the full regime.
>
> >On outcome-level labels
>
> We agree this practical setting deserves a more explicit discussion. Our framework does **not** require direct process-level supervision. In many realistic settings, training is closer to **RLVR (reinforcement learning with verifiable rewards)**, where supervision is only outcome-level—for example, whether a final math answer is correct, or whether code passes unit tests.
>
> The key question is then whether such outcome-level signals are sufficient to learn an approximate prefix-level reward-to-go. Our view is yes: value-learning/RL-style training is designed to propagate terminal supervision backward to intermediate states. In this sense, our theory identifies how accurate such a learned prefix-level verifier needs to be. We’ve made this discussion clearer in the revision.
>
> >On the sentence vs. token-level
>
> We agree this point can be explained more carefully, and we added it in our revision. Our sentence-level framework does **not** require exact sentence-level policy probabilities; **it only requires access to samples** from the sentence-level policy. In this sense, sentence-level inference is a more general setting, while token-level is a special case with more structure.
>
> Similarly, the fact that reward information may only be available at a coarser granularity does not fundamentally conflict with SMC. If reward information is available only every few tokens, or only at the sentence level, then resampling can simply be performed at that same granularity.
>
> >On best-of-N
>
> We agree that best-of-N is an important practical baseline and we discussed it explicitly in the new draft. First, best-of-N is not itself a posterior-sampling method; a soft analogue is terminal weighted resampling. From this perspective, **best-of-N is the hard version of terminal-only resampling**: no intermediate resampling, and selects at the end. As a result, it does not benefit from intermediate reward-to-go guidance, which is why our theory predicts that it generally retains the exponential dependence, whereas intermediate resampling enables the polynomial regime.
>
> Overall, we are grateful for these suggestions. Given these additions and revisions, we respectfully ask the reviewer to reconsider their score.

---

> > ### Author Rebuttal · Reviewer_7ePz · 2026-04-02
> >
> > Thanks to the authors for adding the additional experiments. Please update the final submission with these to further strengthen the paper. I will update my score accordingly.

---

### Official Review · Reviewer_y4P1 · 2026-03-13

**Soundness:** 3
**Presentation:** 1
**Significance:** 2
**Originality:** 3
**Overall Recommendation:** 4
**Confidence:** 2

**Summary:**

This paper studies inference-time scaling with approximate reward models through the lens of Sequential Monte Carlo (SMC). The authors analyze how errors in an approximate reward/value model affect the ability of SMC to efficiently sample from a reward-tilted distribution over responses. The main theoretical result shows that if the local Bellman error of the reward model scales as O(1/T), where T is the reasoning horizon, then SMC can avoid the exponential complexity associated with the growing response space. The paper also provides lower bounds suggesting that sufficiently small Bellman error is necessary to achieve sub-exponential scaling, along with analysis of several SMC variants and corrections.

**Compliance With Llm Reviewing Policy:**

Affirmed.

**Final Justification:**

As mentioned below, the authors helped explain the role of the assumptions and included a synthetic experiment showing the impact of the 1/T bound as I requested. I maintain some evaluations on modern PRMs would be useful and also keep my low confidence due to unfamiliarity with some of the theoretical machinery.

**Key Questions For Authors:**

- I'm curious if the authors believe there to be a significant gap between their worst-case bound and an average-case bound for eg. on-policy samples? I also wonder how accessible it seems to prove such a result.
- Do you have any evidence commonly used reward models fit this O(1/T) bound?
- Could the authors comment on how strong they believe Assumptions 3.1 and 3.2 to be?

**Limitations:**

Limitations not concretely discussed. See weaknesses for my thoughts.

**Strengths And Weaknesses:**

Note: Though this work is outside my expertise, I have done my best to review it.

# Strengths
- As approximate reward models are increasingly common, the paper has an important motivation in understanding when these approximate models are sufficient.
- The O(1/T) bound on Bellman error corresponding to overcoming the exponentially growing response space is an interesting results. I would be interested to know whether this bound is achieved reasonably by practical PRMs or not.
- The paper provides both upper and lower bounds relating Bellman error to inference complexity.

# Weaknesses
- Despite the broad motivation and title, the analysis focuses on Sequential Monte Carlo (SMC), which is a relatively narrow class of inference-scaling methods using reward models in practice. In my experience with recent work on reasoning and inference scaling for LLMs, SMC is not typically a first-choice approach. As a result, the scope suggested by the title may be broader than the methods analyzed, and narrowing the framing might better reflect the contribution.
- I found the writing somewhat difficult to follow. Even though I am familiar with Feynman–Kac models in practice, the introduction was challenging to read and the exposition felt quite terse. Improving accessibility—especially in the introduction and problem setup—would likely broaden the paper’s impact.
- It would help to better contextualize the strength of the assumptions. For example, Assumption 3.2 appears quite strong when applied uniformly over an exponentially large prefix tree of LLM responses. Some discussion of whether such assumptions are plausible for learned reward models would strengthen the paper.
- The paper would benefit from empirical support. For instance, it should be possible to construct a synthetic setup comparing SMC performance under a reward model that satisfies the Bellman error bound versus one that does not. More generally, relating the theoretical bounds to the capabilities of current state-of-the-art process reward models would significantly strengthen the work.

---

> ### Author Rebuttal · Authors · 2026-03-31
>
> We thank the reviewer for the thoughtful comments and for recognizing both the motivation and the technical contribution of the paper.
>
> >On scope, framing, presentation and accessibility
>
> We agree that the current title and framing are too broad. Our contribution is a theory of **SMC-based inference-time scaling** and its closely related one-particle MH-corrected variants. **We revised the title and introduction to make this scope explicit.**
>
>  **We revised the introduction and problem setup**, to better separate the high-level motivation from the FK/SMC machinery, and to add more details in how the inference problem is reformulated in the FK language. We specifically added a three-step arithmetic reasoning example, together with a reasoning-tree figure, to illustrate how the FK model fits in our problem. We apologize for possible confusion.
>
> >On the strength of Assumptions 3.1 and 3.2
>
> We agree that the current assumptions are stated in a uniformly strong form, and in the revised version we explain both why they are needed and how restrictive they are.
>
> At a high level, what we fundamentally and necessarily need is some form of **coverage**. If the base policy has low probability in some regions, then any method will have difficulty drawing samples from there. Conversely, when the base policy is aligned with the target, the importance weights do not fluctuate, the coverage is 1 and sampling becomes easiest. Thus some coverage-type condition is unavoidable.
>
> This is the role of **Assumption 3.1**. Our current version is a worst-case local fluctuation bound, which is the cleanest way to control the one-step importance weights fluctuation. However, **our theory does not fundamentally require the strongest worst-case form**: a more natural weakening is an **on-policy average-case** version of the same local stability requirement. In this sense, quantities such as $C_{act}$ in [1] can be viewed as weaker average-case counterparts. Similarly, for **Assumption 3.2**, the current uniform worst-case form is **not intrinsic to our proof strategy**; it is mainly the cleanest presentation.
>
> To explain it at a high level, TV distance is dual to $f:\\|f\\|\_\infty \le1$, so when the proof reduces to controlling
> $$
> \sup_{\\|f\\|\_\infty\le1}\mathbb{E}[gf],
> $$
> the sharp quantity is $\\|g\\|\_{L\_1}$. In the current paper, we upper bound this further by a stronger worst-case quantity such as $\\|g\\|\_\infty$ for **cleanness and convenience**. This is why the assumptions are currently phrased in a uniform worst-case form rather than in the weakest possible average-case form.
>
> Hence, **the gap to on-policy average-case versions is small**: the main proof skeleton and key techniques are the same, and the main changes are in moment-control details. For some statements, one may additionally need mild higher-moment control, and some other technical proving steps, but **the proof idea does not change**. We made this explicit in the revision and added an appendix discussion on weaker average-case variants. This also contrasts with [1]: there, weakening the assumption appears to change the type of conclusion more substantially, whereas in our SMC theory, the weakening can still preserve the core statement.
>
> >On empirical evidence and the $O(1/T)$ regime
>
> **We added a simple synthetic experiment**, and plotted the SMC TV accuracy against the Bellman error.
>
> >>**Setting.** We use a 0-1 state depth-$T$ Bernoulli tree with $\pi_{ref}={\rm Bernoulli}(1/2)$ and terminal reward
> $$
> φ(s_{1:T})=\exp(β\sum_{t=1}^Ts_t).
> $$
> The reward-to-go is available in closed form, so we use it as ground truth and perturb it to control the Bellman error.
> One main chart is as below (inner value being TV error):
> |N\ε|0.0|2.0|4.0|6.0|8.0|10.0|
> |---|---:|---:|---:|---:|---:|---:|
> |16|0.308|0.355|0.399|0.448|0.526|0.583|
> |32|0.229|0.263|0.314|0.355|0.406|0.456|
> |64|0.167|0.196|0.232|0.268|0.309|0.338|
> |128|0.120|0.141|0.166|0.195|0.226|0.246|
> |256|0.086|0.102|0.123|0.144|0.162|0.178|
> |512|0.062|0.072|0.087|0.103|0.114|0.128|
>
> This successfully illustrates the main qualitative message of the paper: **Bellman error is the core quantity governing SMC accuracy.**
>
> Regarding whether current reward models satisfy an $O(1/T)$ bound, our point is **not** that such a guarantee has already been empirically certified. Rather, the paper identifies **how accurate a reward model must be** as a function of horizon $T$ to avoid exponential. In this sense, the reward model is to be trained, and our theory identifies the target level of Bellman error of training. Equivalently, a reward model with $ε$ error level can efficiently support $O(1/ε)$ reasoning steps.
>
> Overall, we appreciate the reviewer’s suggestions, which help us improve the paper. Given these clarifications and revisions, we respectfully ask the reviewer to reconsider their score.
>
> [1] arxiv:2510.03149

---

> > ### Author Rebuttal · Reviewer_y4P1 · 2026-04-02
> >
> > I thank the authors for their rebuttal. I agree limiting the scope and title to SMC-based methods is necessary. I am also grateful for their authors explanations about the strength of Assumptions 3.1 and 3.2. Finally, the synthetic experiments are interesting. I still believe it would valuable to evaluate the error on modern PRMs to see how far they are from overcoming the exponential and whether this seems a practical endeavor. I will raise by score to 4, though I maintain my low confidence.

---

### Decision · Program_Chairs · 2026-04-30

**Decision:**

Accept (regular)

**Comment:**

The paper studies why and how approximate reward models (rather than perfect ones) are sufficient for improving LLM reasoning through inference-time scaling. It focuses on the Sequential Monte Carlo (SMC) framework. The authors identify Bellman error as the governing factor. They prove that if the Bellman error of an approximate reward model is bounded by O(1/T) (where T is the reasoning horizon), the computational complexity of finding a solution drops from exponential to polynomial.

Reviewers praise the paper for providing a "profound theoretical perspective" and a "technically solid" foundation for understanding why reward-guided search works. The addition of synthetic experiments during the rebuttal phase was crucial in addressing concerns about the theory's applicability. The reviewers however, also find the paper's assumptions are somewhat idealized. Hence, the recommendation is a weak accept.